# Benchmarking Structural Inference Methods for Interacting Dynamical Systems with Synthetic Data

**Aoran Wang** [1]* **Tsz Pan Tong** [1]* **Andrzej Mizera** [2] **Jun Pang** [1]

[1] University of Luxembourg  [2] IDEAS-NCBR & University of Warsaw

{aoran.wang, tszpan.tong, jun.pang}@uni.lu andrzej.mizera@ideas-ncbr.pl

## Abstract

Understanding complex dynamical systems begins with identifying their topological structures, which expose the organization of the systems. This requires robust structural inference methods that can deduce structure from observed behavior. However, existing methods are often domain-specific and lack a standardized, objective comparison framework. We address this gap by benchmarking 13 structural inference methods from various disciplines on simulations representing two types of dynamics and 11 interaction graph models, supplemented by a biological experimental dataset to mirror real-world application. We evaluated the methods for accuracy, scalability, robustness, and sensitivity to graph properties. Our findings indicate that deep learning methods excel with multi-dimensional data, while classical statistics and information theory based approaches are notably accurate and robust. Additionally, performance correlates positively with the graph's average shortest path length. This benchmark should aid researchers in selecting suitable methods for their specific needs and stimulate further methodological innovation. Project website: `https://structinfer.github.io/`.

## 1 Introduction

Dynamical systems pervade various domains, from gravitational interactions among galaxies to intricate chemical reactions. A common characteristic of these systems is their representation as interaction graphs, where nodes symbolize agents, edges depict interactions, and the adjacency matrix encapsulates the underlying structure. Examples of inherent interaction graphs are found in physical systems [61, 42, 111], multi-agent systems [15, 62], and biological systems [102, 85]. Understanding the structure of these interaction graphs is crucial as it enhances predictability and manipulability of dynamical systems, despite the complexity of the task [28].

Often, only observable node attributes within a specific timeframe are available, partially or fully obscuring the interaction graph's underlying structure amid dynamic complexities. This necessitates an approach to uncover the hidden structure of dynamical systems through observable features, leading to the concept of *structural inference*. Here, the compilation of observed features over time, termed a *trajectory*, is crucial for understanding dynamical systems. Unraveling the graph's structure simplifies interaction modeling, especially when the graph dimensions and interactions are known and time-independent.

Structural inference, rooted in statistics, has evolved significantly within the Bayesian network framework, prompting numerous algorithm proposals [70, 100, 101, 89, 25]. Notable advancements, such as in genome sequencing [93], have enabled the study of gene expression and regulatory mechanisms, fostering various structural inference methods for gene regulatory networks (GRNs) [71,

---

*Equal first contributions

38th Conference on Neural Information Processing Systems (NeurIPS 2024) Track on Datasets and Benchmarks.

33, 52, 46, 2, 72, 80]. Conversely, recent deep learning approaches focus primarily on general dynamical systems [58, 108, 112, 68, 19, 106].

Existing methods are often evaluated on distinct datasets and specific graph types, each tailored to different research domains with unique underlying assumptions. To address this fragmentation, we developed the Dataset for Structural Inference (DoSI), featuring a variety of interaction graphs and dynamical transition functions. We then established a unified and impartial benchmark to evaluate a broad range of techniques across diverse domains. This benchmark assesses established and cutting-edge methods using over 213,445 trajectories from both the meticulously curated DoSI and real-life biological datasets. These datasets include both one-dimensional and multi-dimensional trajectories, further enriched with varying levels of Gaussian noise to simulate real-world conditions.

This pioneering benchmark, requiring over 706,800 CPU hours and 263,400 GPU hours, allows us to rigorously evaluate the accuracy, scalability, robustness, and data efficiency of these methods. Our findings reveal that classical statistic methods are scalable and reliable across various datasets, information theory-based methods are notably robust, and deep learning methods excel in handling multi-dimensional features. This comprehensive evaluation offers valuable insights and sets the stage for future advancements in structural inference research. In summary, the contributions are:

- We developed the Dataset for Structural Inference, a versatile dataset featuring a range of interaction graphs and dynamical functions to facilitate broad applicability in structural inference research.
- The study introduces a unified and impartial benchmark that evaluates 13 structural inference methods using over 213,445 trajectories from synthetic and real-life datasets, encompassing both one-dimensional and multi-dimensional data.
- The benchmark provides comprehensive insights, revealing that classical statistical methods excel in scalability, information theory-based methods in robustness, and deep learning methods in handling complex multi-dimensional features.
- The findings from this extensive evaluation not only enhance our understanding of different structural inference approaches but also set the groundwork for future innovations to tackle dynamic and noisy systems.

## 2 Preliminaries

In this section, we delve into the intricacies of structural inference of dynamical systems. We conceptualize a dynamical system as a directed underlying interaction graph, wherein the system's agents translate to nodes, and the directed interactions among these agents manifest as edges in the graph. Denoted as $\mathcal{G} = (\mathcal{V}, \mathcal{E})$, the directed graph consists of $\mathcal{V}$, the feature set of $n$ nodes represented by $\{V_i, 1 \leq i \leq n\}$, and $\mathcal{E}$, the set of edges. The temporal evolution of nodes' features is encapsulated in trajectories: $\mathcal{V} = \{V^0, V^1, \ldots, V^T\}$, spanning $T + 1$ time steps, with $V^t$ signifying the feature set of all $n$ nodes at time step $t$: $V^t = \{V_0^t, V_1^t, \ldots, V_n^t\}$. The feature vector at time $t$ for node $i$, denoted as $V_i^t \in \mathbb{R}^n, 1 \leq t \leq T$, is n-dimensional.

In our assumptions, the nodes are observed in their entirety, and $\mathcal{E}$ remains immutable during the observation. From $\mathcal{E}$, we derive an asymmetric adjacency matrix denoted as $\mathbf{A} \in \mathbb{R}^{n \times n}$. Within $\mathbf{A}$, each element $\mathbf{a}_{ij} \in 0, 1$ indicates the presence ($\mathbf{a}_{ij} = 1$) or absence ($\mathbf{a}_{ij} = 0$) of an edge from node $i$ to node $j$. An alternative representation for the graph structure is an edge list, where each entry $[i, j]$ in the list signifies a directed edge originating from node $i$ and terminating at node $j$. Given the node features observed over a time interval in $\mathcal{V}$, the primary focus of this paper centers on the challenge of structural inference. This challenge involves the unsupervised reconstruction of either the asymmetric adjacency matrix $\mathbf{A}$ or the edge list that encapsulates the underlying interaction graph. It is important to note that this problem is distinct from link prediction tasks, where connections are at least partially observable [115, 41].

## 3 Methods for structural inference

### 3.1 Methods based on classical statistics

Statistical methods prioritize inference accuracy and uncertainty. Its results are interpreted conservatively, making it widely applicable across diverse scenarios:

- ⋆ **ppcor** [57]: ppcor method computes semi-partial correlations between pairs of nodes, quantifying the specific portion of variance attributed to the correlation between two nodes while accounting for the influence of other nodes. This computation draws on both Pearson and Spearman correlations.

⋆ **TIGRESS** [46]: Contrasting with other structural inference methods, which remove redundant edges from predicted edges, TIGRESS focuses on feature selection by iteratively adding more nodes to predict the target node using least angle regression and bootstrapping.

## 3.2 Methods based on information theory

Mutual information (MI) is a probabilistic measure of dependency described by the equation: $I(X;Y) = H(X) + H(Y) - H(X,Y)$, where $X, Y$ are random variables, $H(\cdot)$ and $H(\cdot, \cdot)$ are the entropy and joint entropy, respectively. MI possesses the ability to capture nonlinear interactions [31], rendering it widely used in various fields including neuroscience [83, 55], bioinformatics [116], and machine learning [10]. However, despite direct interactions, indirect interactions and data noise can introduce complexity and challenges. Different methods were proposed to tackle this problem:

⋆ **ARACNe** [71]: ARACNe is a popular method for GRN inference. The algorithm is initiated by calculating pairwise MI and subsequently employing the Data Processing Inequality principle to eliminate indirect interactions. This principle posits that the MI between two nodes connected by an indirect interaction should not surpass the MI of either node connected directly to a third node.

⋆ **CLR** [33]: Similar to ARACNe, CLR employs pairwise MI but differs in the interpretation of calculated MI. CLR relies on assuming a background noise distribution for MI and subsequently identifies interactions as MI outliers after both row- and column-wise standardization.

⋆ **PIDC** [17]: Partial Information Decomposition (PID) [110] undertakes the decomposition of MI into redundant, synergistic, and unique information. PIDC adopts the concept of PID to GRN inference and interprets aggregated unique information as the strength of gene interaction.

⋆ **Scribe** [86]: Scribe utilizes Restricted Directed Information [87] and its variants [88] to quantify causality within the structure by considering the influence of confounding factors.

## 3.3 Methods based on tree algorithms

The decision tree is a powerful supervised method that divides the feature space into subspaces and uses linear regressions within each. Despite its versatility across data types [76], decision trees can overfit, prompting strategies like boosting and bagging. Examples include AdaBoost [34], random forests [47], extremely randomized trees [38], XGBoost [20], and LightGBM [56]. Yet, applying tree-based methods directly to structural inference is constrained by the unsupervised task nature. GENIE3 [52], using random forests, addresses this, succeeding in modeling GRNs. GENIE3 models gene dynamics using other genes' behavior, revealing how supervised methods can aid structural inference.

⋆ **dynGENIE3** [50]: dynGENIE3 extends GENIE3 by concentrating on the temporal aspect, employing ordinary differential equations (ODEs) to model time series dynamics. In this approach, a random forest is employed for each gene to capture the derivatives within the time series.

⋆ **XGBGRN** [69]: XGBGRN aligns with the principles of dynGENIE3, though it diverges in its choice of algorithm. Specifically, XGBGRN leverages XGBoost, in place of random forests, to model the derivatives of the time series data.

## 3.4 Methods based on deep learning

Contemporary structural inference methods [58, 68, 19, 106] leverage the information bottleneck (IB) principle [99, 98, 94] and variational autoencoders (VAEs), a form of variational IB approximation [3]. As outlined in [106], these VAE-based methods solve: $\mathbf{Z} = \arg\min_{\mathbf{Z}} I(\mathbf{Z}; V^t, \mathbf{A}) - \mathfrak{u} \cdot I(\mathbf{Z}; V^{t+1})$, where $\mathbf{Z}$ is the latent feature space, $V^t$ represents node features at time $t$, $\mathbf{A}$ is the adjacency matrix, and $\mathfrak{u}$ is the Lagrangian multiplier. This approach extracts the dynamical system's structure through VAE sampling. Extensions to this framework [78, 120] incorporate architectural designs in graph neural networks and diffusion models to better suit data characteristics. Moreover, neural networks enable handling both one-dimensional and multi-dimensional features, unlike earlier non-deep learning methods focused on one-dimensional features. Prominent deep learning structural inference methods encompass:

⋆ **NRI** [58]: NRI stands as a pioneering method that employs a VAE for structure inference. Its encoder integrates node-to-edge and edge-to-node processes to collect node features and acquire edge features. In this context, NRI assumes a fixed fully connected $\mathbf{A}$ within the encoder.

- ★ **ACD** [68]: ACD introduces a probabilistic approach to amortized causal discovery to learn the causal graph from time series. This method also addresses latent confounding issues by predicting an additional variable and implementing structural bias.
- ★ **MPM** [19]: MPM, distinct from typical message-passing approaches, utilizes relational interaction in the encoder and spatio-temporal message-passing in the decoder. This alteration comprehensively captures relationships and enhances the grasp of dynamical rules.
- ★ **iSIDG** [106]: iSIDG diverges from other VAE-based methods by iteratively updating $\mathbf{A}$ based on direction information deduced from the adjacency matrix. Its goal centers on inferring the authentic interaction graph by removing indirect edges that contribute to confusion.
- ★ **RCSI** [107]: RCSI, a variant of iSIDG, incorporates reservoir computing units that concentrate on time series prediction, enabling the VAE to prioritize structure inference. This modification significantly reduced the number and length of trajectories required for training.

### 3.5 More related works

Besides the methods previously discussed, fNRI decomposes the inferred interaction graph into a multiplex graph, with each layer signifying a distinct interaction type [108]. MetaNRI employs modular meta-learning to implicitly encode time invariance and contextually infer relationships [4]. In the adjacent field of causal structural discovery, many methods necessitate interventional data or rely on strong assumptions that may not be suitable for our settings [121, 27, 16, 40, 114, 53, 113, 13, 21]. Recent approaches like LOCS [59] and Aether [60] offer structural inference techniques for hybrid dynamical systems, while Graph-Switching Dynamical Systems [64] and Amortized Equation Discovery [65] target systems with switching dynamics. Methods like REDSDS [5] and recurrent SLDS [63] also contribute to the growing pool of structural inference techniques by focusing on systems with latent switching behavior. As we are updating the benchmark with more recent papers, we will include these methods in the near future. While this paper does not exhaust all methods, such as [117, 23], we recommend that researchers use our datasets to benchmark their approaches.

**Other benchmarks for structural inference.** This study is, to our knowledge, the first to introduce a unified, objective, and reproducible benchmark for structural inference in interacting dynamical systems. Prior benchmarks have been domain-specific, addressing areas such as GRNs in single-cell data [9, 84, 119], gene co-expression networks [22, 77], map inference algorithms [14, 1, 18], chemical reaction networks [67, 11], and functional connectivity [24, 66]. Although benchmarks in causal discovery exist [6, 74], they operate under different assumptions. Notably, the closest related work [122] primarily focuses on time-series forecasting. Our benchmark distinguishes itself by offering a comprehensive, cross-domain framework that advances structural inference methodologies and enables meaningful comparisons across diverse approaches.

## 4 Datasets for benchmarking

While domain-specific datasets like Boolean models and miRNA-target genes datasets exist for structural inference [84, 22], they are often too specialized, limited in size, or challenging to interpret. This highlights a significant gap for a unified, interpretable dataset in the field. To address this, we developed the **D**ataset f**o**r **S**tructural **I**nference (DoSI), which involves 1) creating interaction graphs and 2) simulating dynamical systems, detailed in subsequent sections. Additionally, we incorporated a real-world biological dataset to not only demonstrate the practical applicability of structural inference methods but also to highlight the dataset's limitations.

### 4.1 Underlying interaction graphs of DoSI

Our primary goal is to use synthetic data to evaluate structural inference methods, taking into account the diversity in structure and characteristics of underlying interaction graphs. We referenced existing literature [8, 7, 32] to gather properties from 11 types of real-world graphs, including brain networks (BN), chemical reaction networks in the atmosphere (CRNA), food webs (FW), gene coexpression networks (GCN), gene regulatory networks (GRN), intercellular networks (IN), landscape networks (LN), man-made organic reaction networks (MMO), reaction networks inside living organisms (RNLO), social networks (SN), and vascular networks (VN). These graphs' properties—such as clustering coefficient $C$, average shortest path length $d$, the power-law exponent of the degree

distribution $\gamma$, average degree $\langle k \rangle$, density $\delta$, and if available, the power-law exponent of the in-/out-degree distribution $\gamma^{\text{in}}$ and $\gamma^{\text{out}}$ -are detailed in Table 1 in the Appendix.

This table shows significant variability in graph properties, underscoring the importance of mimicking this diversity in our synthetic graph generation to effectively evaluate structural inference methods. The size of these graphs, ranging from 15 to 250 nodes, also influences method performance. Tailored creation pipelines for different graph types, based on these properties and structural biases from literature [58, 19, 68, 106], are further discussed in Appendix B.1.

### 4.2 Dynamical systems

In DoSI, we use the generated graphs as interaction graphs to simulate dynamical systems, where node features evolve over time and are influenced by both the interaction graph and the dynamic function. The interaction graph determines which nodes interact, and the dynamic function quantifies these interactions' impact. We utilize two common simulations, "Springs" and "NetSims" [19, 58, 68, 106, 108], to generate trajectories. We detail the functionality of these simulations, modifications for our purposes, and the generation of trajectories with varying Gaussian noise levels. Additionally, we prepare an experimental biological dataset to evaluate the effects of noise and imperfections in data collection. The method of preparing this dataset through trajectory reconstruction is detailed as well. For further information on the dynamical simulations, please see Appendix B.2 and B.4.

**Springs simulation.** Inspired by prior work [58], we simulate the motion of spring-connected particles within a 2D box. Particles (nodes) are interconnected by springs (edges) adhering to Hooke's law. We use interaction graphs to set up these connections and generate trajectories with various initial conditions. The dynamics are governed by a second-order ODE, simplified here for clarity:

$$m_i \cdot x_i''(t) = \sum_{j \in \mathcal{N}_i} -k \cdot \big(x_i(t) - x_j(t)\big), \tag{1}$$

where each node's mass $m_i$ is assumed to be 1, and the spring constant $k$ is also set to 1. $\mathcal{N}_i$ refers to the set of neighboring nodes with directed connections to node $i$. We integrate this equation to compute $x_i'(t)$ and subsequently $x_i(t)$ for each time step. The sampled values of $x_i'(t)$ and $x_i(t)$ form the 4D node features. We produce trajectories with 49 time steps for training and validation, and 100 for testing, resulting in 8,000 training, 2,000 validation, and 2,000 test trajectories per graph.

**NetSims simulation.** This simulation models brain activity data using nodes that represent brain regions, and edges that define interactions based on prior interaction graphs [95]. The dynamics follow a first-order ODE:

$$x_i'(t) = \sigma \cdot \sum_{j \in \mathcal{N}_i} x_j(t) - \sigma \cdot x_i(t) + C \cdot u_i, \tag{2}$$

where $\sigma$ controls temporal smoothing and is set to $0.1$ [95], and $C$, the interaction weight, is zero to minimize noise. The 1D node features at each time step are formed using the sampled $x_i(t)$. We generate trajectories under conditions similar to those of the Springs simulation.

To this end, our benchmark includes two types of dynamical systems modeled by first-order ODEs (NetSims) and second-order ODEs (Springs), covering a broad spectrum of real-world phenomena from motion dynamics to single-cell behavior. Additionally, with the inclusion of 'Charged Particles' detailed in Appendix 6.4, we address systems influenced by quadratic dependencies like electrostatic and gravitational forces, further expanding the applicability of our benchmark. Each dynamical system chosen aims to represent a comprehensive category of real-world systems.

**Addition of Gaussian noise.** Furthermore, to assess the performance of the structural inference methods under noisy conditions, we add Gaussian noise at various levels to the generated trajectories. The node features with added noises $\tilde{v}_i^t$ can be summarized as: $\tilde{v}_i^t = v_i^t + \zeta \cdot 0.02 \cdot \Delta$, where $\zeta \sim \mathcal{N}(0, 1)$, $v_i^t$ is the original feature vector of node $i$ at time $t$, and $\Delta$ is the noise level. The noise levels range from 1 to 5 to all the original trajectories.

**EMT dataset.** To compare model performance between synthetic and real-world data, we applied benchmarking models on a single-cell RNA sequencing (scRNA-seq) dataset from an epithelial–mesenchymal transition (EMT) study, originally collected by Cook and Vanderhyden [26] and processed by Sha et al. [92]. This dataset includes 3,133 cells and 3,000 genes, sampled across 5

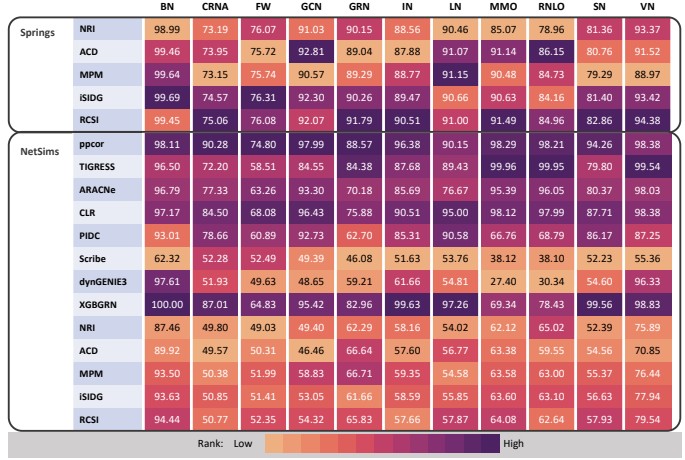

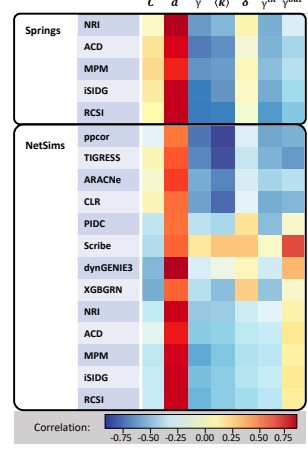

(a) Average AUROC values (in %) of investigated structural inference methods on noise-free trajectories, clustered by the type of interaction graphs and the type of simulations.

(b) Correlations between the AUROC results of structural inference methods and the properties of the graphs.

Figure 1: Results of investigated methods clustered by the type of interaction graphs and the correlations with the graph properties.

time points in 7 days. Using the STRING database [97], we identified the interaction network of the top 50 high-variance genes, serving as the ground-truth GRN. After removing isolated components, the network was reduced to 36 nodes with 103 edges. Trajectories were reconstructed using Waddington-OT [91] and interpolation methods, resulting in 577 trajectories of 22 time steps each. Detailed dataset construction information is available in Appendix B.4.

## 5 Benchmarking setup

To compare the structural inference methods in a unified, objective, and reproducible manner across different domains, we design three sets of experiments:

1. **Evaluation on original Trajectories**: This assesses methods using original, noise-free trajectories to understand how the properties of the underlying interaction graph affect inference results.
2. **Scalability analysis**: Following initial evaluations, this experiment tests the scalability of methods by analyzing their performance with varying computational resources and graph sizes.
3. **Evaluation on noisy trajectories**: Methods are tested against trajectories with different levels of Gaussian noise to determine their robustness.

Additionally, we explore the data efficiency of these methods by evaluating their performance on shorter trajectories, with results detailed in Appendix D.3. Note that methods based on classical statistics, information theory, and tree algorithms are evaluated only with one-dimensional trajectories due to compatibility issues with multi-dimensional features.

For performance metrics, we use the Area Under the Receiver Operating Characteristic curve (AUROC), which measures inference accuracy and the ability to distinguish true from false edges in the interaction graph. We ensure robust evaluation by averaging results from three runs of each method on labeled trajectory sets and performing an additional run on the set with the lowest AUROC value. AUROC is chosen over other metrics like accuracy, F1 score, and Hamming distance due to its effectiveness in handling imbalanced datasets and providing a comprehensive performance assessment at various thresholds (see Appendix D.4 for more on metric selection).

Additionally, we introduce the "Charged Particles" dataset, which simulates multi-dimensional trajectories governed by Coulomb force, tailored for deep learning models due to its complexity. This dataset contrasts with the Springs simulation's Hooke's law dynamics by featuring sophisticated ejection and entanglement dynamics (see Appendix 6.4 for details and benchmarking results). We also evaluate the performance of structural inference methods on the EMT dataset, a single-cell RNA sequencing dataset, with findings discussed in Appendix D.5.

# 6 Benchmarking results

## 6.1 Benchmarking over different interaction graphs

To evaluate the structural inference methods discussed in Sections 3.1 - 3.4, we conducted tests on trajectories generated using all 11 types of underlying interaction graphs described in Section 4.1. These tests included both types of simulations and were executed without any added noise. Despite using Tesla V100 GPU cards and facing computational limits, we successfully processed graphs up to 100 nodes, with a total of 706,816 CPU hours and 263,473 GPU hours.

Appendix C details the implementation of each method, including computational resources and hyperparameter optimization. Additionally, this section presents a clustering analysis of the AUROC results by interaction graph type and simulation, displaying average AUROC values in Fig. 1a and providing detailed data in Appendix D.1. Fig. 1b presents a heatmap showing correlations between the methods' average AUROC values and the properties of the interaction graphs, using terminologies from Section 4.1.

Deep learning methods like NRI and ACD exhibit superior performance on multi-dimensional data, which is especially evident when comparing the Springs and NetSims simulations. For example, NRI shows a 46.35% higher AUROC on gene coexpression networks, and ACD shows a 34.30% increase on landscape networks, as seen in Fig. 1a. These results highlight that multi-dimensional features provide a wealth of information, enhancing the effectiveness of these methods in structural inference by leveraging the complex interrelationships between different feature dimensions. Conversely, classical statistical methods such as ppcor and TIGRESS demonstrate remarkable consistency across various graph types, maintaining medium to high ranks across datasets. Their stable performance, illustrated in Fig. 1a, underscores their robustness and adaptability, making them reliable choices for scenarios where interaction graph structures do not match more complex or specialized models.

The correlation between the performance of structural inference methods and the properties of interaction graphs, as shown in Fig. 1b, reveals key insights. Generally, there is a positive correlation with the average shortest path length and a negative correlation with the average degree of the graphs. This indicates that methods perform better on sparser graphs with longer path lengths, where the simpler connections likely enhance method effectiveness. In contrast, denser graphs with high connectivity and shorter path lengths reduce performance, possibly due to increased complexity and noise, which can mask the underlying structures these methods seek to discern.

## 6.2 Benchmarking over scalability

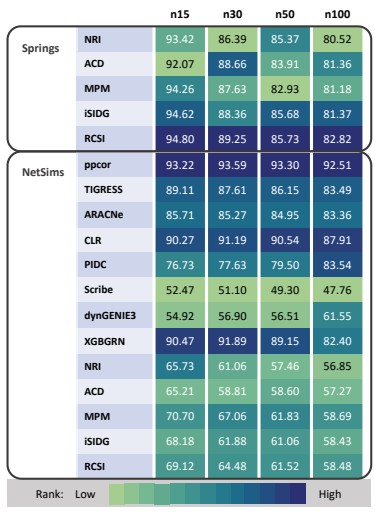

| | | n15 | n30 | n50 | n100 |
|---|---|---|---|---|---|
| Springs | NRI | 93.42 | 86.39 | 85.37 | 80.52 |
| | ACD | 92.07 | 88.66 | 83.91 | 81.36 |
| | MPM | 94.26 | 87.63 | 82.93 | 81.18 |
| | iSIDG | 94.62 | 88.36 | 85.68 | 81.37 |
| | RCSI | 94.80 | 89.25 | 85.73 | 82.82 |
| NetSims | ppcor | 93.22 | 93.59 | 93.30 | 92.51 |
| | TIGRESS | 89.11 | 87.61 | 86.15 | 83.49 |
| | ARACNe | 85.71 | 85.27 | 84.95 | 83.36 |
| | CLR | 90.27 | 91.19 | 90.54 | 87.91 |
| | PIDC | 76.73 | 77.63 | 79.50 | 83.54 |
| | Scribe | 52.47 | 51.10 | 49.30 | 47.76 |
| | dynGENIE3 | 54.92 | 56.90 | 56.51 | 61.55 |
| | XGBGRN | 90.47 | 91.89 | 89.15 | 82.40 |
| | NRI | 65.73 | 61.06 | 57.46 | 56.85 |
| | ACD | 65.21 | 58.81 | 58.60 | 57.27 |
| | MPM | 70.70 | 67.06 | 61.83 | 58.69 |
| | iSIDG | 68.18 | 61.88 | 61.06 | 58.43 |
| | RCSI | 69.12 | 64.48 | 61.52 | 58.48 |

Rank: Low ▬▬▬▬▬ High

Figure 2: Average AUROC values (in %) of structural inference methods on noise-free trajectories, clustered by the number of nodes in graphs and the type of simulations.

Using the raw results from Section 6.1, we conducted a clustering analysis based on the number of nodes in the interaction graphs to assess the scalability of the structural inference methods. The outcomes of this analysis are displayed in Fig. 2.

The performance of the majority of the methods tends to deteriorate as the dynamical systems increase in size, as demonstrated by a consistent trend in Fig. 2. Notably, PIDC and dynGENIE3 show improved inference results for larger systems, suggesting that these methods can effectively utilize the increased information available in larger graphs. This indicates that while larger systems provide more data, extracting and leveraging this information efficiently remains critical for enhancing method performance.

Deep learning methods show a significant sensitivity to graph size compared to classical statistical methods. The smallest decrease in AUROC among deep learning methods is 7.94%, in stark contrast to classical methods like ppcor, which only shows a 0.71% decrease when comparing graphs with 100 nodes to those with 15 nodes. This illustrates the scalability challenges for deep learning methods, despite their versatility in handling diverse feature types.

Moreover, classical statistical methods such as ppcor and TIGRESS prove to be highly scalable, maintaining stable performance across various graph sizes. Their consistent performance across different node counts, combined with their robustness across diverse interaction graphs, underscores their reliability among the evaluated structural inference methods.

## 6.3 Benchmarking over robustness

The robustness of structural inference methods is crucial for real-world applications, where data often contain noise. To evaluate this robustness, we generated noisy trajectories using NS_BN with varying levels of Gaussian noise. The differences in AUROC values between noisy and noise-free data, denoted as $\Delta$AUROC, along with their standard deviations, are summarized in Fig. 3 and detailed in Appendix D.3.

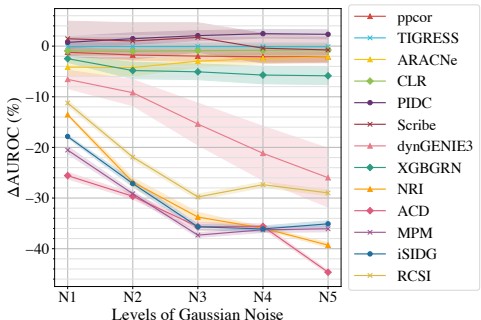

Figure 3: Performance drops (in %) on BN trajectories with different levels of added Gaussian noise.

Methods based on classical statistics and information theory, such as TIGRESS, CLR and PIDC, demonstrate resilience against various levels of Gaussian noise, maintaining performance levels despite noise presence. This suggests that their capacity to extract latent information through correlations or mutual information between node pairs effectively compensates for noise perturbations. These findings are promising for the future design of robust structural inference methods.

Conversely, tree-based and deep learning methods exhibit distinct sensitivities to Gaussian noise. Tree-based methods show variability in both the average AUROC and standard deviations, indicating a fluctuation in performance with increasing noise levels. Deep learning methods, while maintaining consistently low standard deviations, exhibit a decline in average performance under noisy conditions. This pattern suggests that deep learning methods may struggle to differentiate between noise effects and genuine data perturbations, leading to decreased performance as noise levels increase.

## 6.4 Benchmarking with Charged Particles

We observed that the two dynamic simulations do not encompass a prevalent type of real-world dynamical system characterized by quadratic dependencies. To address this gap, we introduce a third simulation of dynamical systems, grounded in the Coulomb force interactions among charged particles, and we have named it the "Charged Particles" simulation.

**Simulation of Charged Particles**. We simulate the movement of charged particles within a 2D enclosure, where nodes represent particles and edges symbolize the Coulomb forces acting between pairs of particles. Unlike the Springs and NetSims simulations, the Charged Particles simulation entails a unique approach: all nodes are interconnected, and none of the 11 types of generated underlying interaction graphs are employed. Consequently, every pair of nodes interacts, even if the interaction might be weak when the nodes are distant. These interactions involve either attraction or repulsion. Drawing inspiration from [58] and following a concept akin to the Springs simulation, our simulation involves $N$ particles (point masses) located within a 2D enclosure and subject to no external forces. The parameter $N$ is chosen from the set $15, 30, 50, 100$. The simulation accounts for elastic collisions with the boundary of the enclosure. The particles carry charges $q_i \in \pm q$, sampled uniformly at random. The inter-particle interactions are governed by Coulomb forces, defined as $F_{ij}(t) = C \cdot \text{sign}(q_i \cdot q_j) \cdot \frac{1}{\|x_i(t) - x_j(t)\|^2}$, with a constant $C$ set to 1. Here, $F_{ij}(t)$ denotes the force exerted on particle $i$ by particle $j$ at time $t$, and $x_i(t)$ represents the 2D location vector of particle $i$ at time $t$. So the adjacency matrix $\mathbf{A}$ in this simulation is formed as a matrix with each element $a_{ij}$ in it as either $+1$ or $-1$, where $a_{ij} = +1$ stands for repelling between node $i$ and $j$, while $a_{ij} = -1$ stands for attracting between node $i$ and $j$. The dynamics of the Charged Particles simulation are

encapsulated in an ODE characterized by quadratic dependencies on particle locations, expressed as:

$$m_i \cdot x_i''(t) = \sum_{j \in \mathcal{N}_i} C \cdot \mathrm{sign}(q_i \cdot q_j) \cdot \frac{1}{\|x_i(t) - x_j(t)\|^2}, \tag{3}$$

Here, $m_i$ represents the mass of node $i$, assumed to be 1 for simplicity. $\mathcal{N}_i$ refers to the set of neighboring nodes with connections to node $i$. Here it represents all nodes in the system. The equation is integrated to compute $x_i'(t)$, and subsequently, $x_i(t)$ is determined for each time step. These calculated values of $x_i'(t)$ and $x_i(t)$ collectively constitute the 4D node features at each time point. Initially, the positions are drawn from a Gaussian distribution $\mathcal{N}(0, 0.5)$, while the initial velocities, represented as 2D vectors, are randomly generated with a norm of 0.5. With these initial positions and velocities in the 2D plane, trajectories are simulated using the solutions to Eq. 3. The simulation employs leapfrog integration with a small time step size of 0.001 seconds, and the trajectories are sampled at intervals of 100 minor time steps. As a result, the feature representation of each node at each time step consists of a 4D vector encompassing 2D positions and 2D velocities.

The simulation's design ensures that the next value of a particle's feature depends on its present value and interactions with other particles. Utilizing a set of initial positions and velocities, we generate trajectories for the current interacting dynamical system, encapsulating the feature vectors of all particles within the designated time frame. Specifically, trajectories comprising 49 time points (obtained through integration over 4,900 minor time steps) are generated for training and validation purposes. For testing, trajectories with 100 time steps are generated, aligning with the requirements in [58, 106]. To ensure robustness, a total of 8,000 trajectories are generated for training, along with 2,000 for validation and 2,000 for testing. This process is repeated thrice, yielding three sets of trajectories with the same node count but distinct initializations.

**Implementation of Structural Inference Methods**. For methods reliant on deep learning, we maintain uniform settings akin to those utilized for the Springs simulation trajectories. Furthermore, we configure the parameter "edge_types" to a value of two, aligning with the requirement to infer the two distinct edges corresponding to $a_{ij} = \pm 1$. However, it's crucial to note that the remaining methods are tailored explicitly for structural inference tasks involving trajectories featuring one-dimensional attributes. Regrettably, their respective literature lacks both theoretical and practical guidelines pertaining to adapting these methods for trajectories characterized by multi-dimensional attributes. Additionally, these methods inherently lack the capability to deduce multiple edge types, thereby restricting their applicability in this context. Consequently, the deep learning structural inference methods were exclusively employed for the analysis of the Charged Particles dataset.

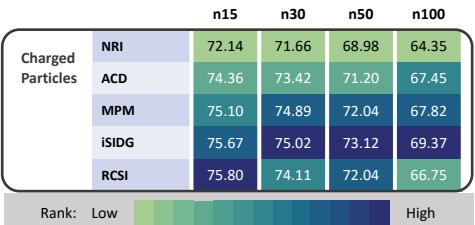

Figure 4: AUROC values (in %) of deep learning structural inference methods on Charged Particles trajectories.

**Results**. Figure 4 provides a comprehensive summary of the average AUROC values and standard deviations for each method across various node counts within the graph. A comparison of these results with those from the Springs dataset reveals that while all methods continue to successfully infer the structure of the underlying interaction graphs, their performance is relatively diminished in this case. The reason lies in the increased complexity of the task, as the methods are now required to infer two distinct edge types, which inherently poses a greater challenge. Moreover, it is noteworthy that the performance of all methods is influenced by the number of nodes present within the graph, corroborating with Section 6.2. The sensitivity to node count underscores the intricate interplay between the size of the graph and the efficacy of the methods. In light of the presented data, it becomes evident that the feasibility of deep learning methods in the structural inference of dynamical systems governed by quadratic dependencies on locations is empirically substantiated.

# 7 Conclusion

In this study, we benchmarked 13 structural inference methods using trajectories from two types of dynamical simulations and various underlying interaction graphs, assessing their performance in the

presence of noise, varying trajectory lengths, and real-world scenarios. Our findings highlight several key insights:

- **Leveraging correlations:** Methods like ppcor and TIGRESS, based on classical statistics, excel in stability and accuracy, effectively leveraging time-series correlations between nodes to enhance structural inference. These methods are robust against noisy and short trajectories, illustrating their efficacy in challenging data conditions.

- **Importance of dimensionality:** Deep learning methods outperform in multi-dimensional settings, underscoring the value of diverse, multi-dimensional data in capturing complex node dynamics and improving inference accuracy. In contrast, classical methods are preferable when only one-dimensional data is available.

- **Performance on sparse graphs:** All evaluated methods yield better results with trajectories from sparse and less connected graphs, suggesting potential for developing techniques to estimate graph properties without prior knowledge.

- **Leveraging mutual information against noise:** Information theory-based methods like PIDC and Scribe demonstrate robustness against Gaussian noise, leveraging mutual information metrics to mitigate noise effects and inform robust algorithm design.

Despite these insights, the study's limitations include reliance on static graph assumptions and a focus on a limited set of methods. For a detailed discussion of these limitations, see Appendix E.

**Updating Plan.** n the near future, we plan to update the benchmark by incorporating results from additional methods, including recurrent SLDS [63], LOCS [59], REDSDS [5], Aether [60], SDS [64] and AMORE [65]. We will also stay attentive to the latest advancements in structural inference and continually integrate new methods into the benchmark. We encourage researchers in the field to benchmark their methods using the DoSI dataset to further advance this area of research.

**Outlook.** The findings underscore the value of leveraging correlations and mutual information in structural inference. Future research could explore innovative methods that apply these principles across both one-dimensional and multi-dimensional feature trajectories, potentially using neural networks to learn feature representations and perform advanced correlation and mutual information analyses. These approaches could extend the scope of structural inference to more complex and dynamic systems, making them more applicable to real-world scenarios.

In addition, developing and evaluating structural inference methods for systems with evolving structures should be a key focus. Many real-world dynamical systems, such as biological networks, social systems, and technological infrastructures, exhibit dynamic topologies where nodes and edges change over time. Capturing these evolving structures is crucial for accurately modeling and understanding such systems.

Another important direction is bridging the gap between simulated and real-world data by incorporating partial observations and various types of noise. This would help address the challenges posed by limited real-world data and create more realistic simulations, ultimately enhancing the applicability and robustness of structural inference methods in practice.

As we continue to benchmark structural inference methods that meet our criteria, we encourage researchers to utilize the Dataset for Structural Inference (DoSI) to evaluate their methods or to contact us for benchmarking. We are open to new approaches and are eager to advance research in structural inference.

# Acknowledgements

The generation, collection, and storage of the dataset used in this work are under the project BSIMDS, which is supported by a collaboration project between the High-Performance Computing Team of the University of Luxembourg (ULHPC) and Amazon Web Services (AWS). The experiments presented in this paper were carried out using the HPC facilities of the University of Luxembourg [103] (see `hpc.uni.lu`). Besides that, authors Tsz Pan Tong and Jun Pang acknowledge financial support of the Institute for Advanced Studies of the University of Luxembourg through an Audacity Grant (AUDACITY-2021).

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

# Appendix of Benchmarking Structural Inference Methods for Interacting Dynamical Systems with Synthetic Data

## A  Dataset documentation

Here, we provide documentation for our dataset in the common datasheets format [37].

### A.1  Motivation

Q1  **For what purpose was the dataset created?** Was there a specific task in mind? Was there a specific gap that needed to be filled? Please provide a description.

We produced the dataset to evaluate the structural inference methods mentioned in this work. To the best of our knowledge, it is the first dataset that includes the trajectories based on eleven different types of underlying interacting graph structures. Furthermore, it is also the first dataset that provides trajectories of both one-dimensional and multi-dimensional features for structural inference. Comprehensive evaluation of the performance of structural inference methods originating from different research disciplines requires an objective and unified dataset containing both trajectories of different dimensions and trajectories based on different underlying interacting graphs. Our goal was to create a dataset that could be utilized for this purpose.

Q2  **Who created the dataset (for example, which team, research group) and on behalf of which entity (for example, company, institution, organization)?**

The dataset was a joint effort by three authors: Aoran Wang, Tsz Pan Tong, and Jun Pang. The authors are researchers affiliated with the Department of Computer Science at the University of Luxembourg. Jun Pang is also affiliated with the Institute for Advanced Studies (IAS) of the University of Luxembourg.

Q3  **Who funded the creation of the dataset?** If there is an associated grant, please provide the name of the grantor and the grant name and number.

The generation, collection, and storage of the dataset used in this work are under the project BSIMDS, which is supported by a collaboration project between the High-Performance Computing Team of the University of Luxembourg (ULHPC) and Amazon Web Services (AWS). The experiments presented in this paper were carried out using the HPC facilities of the University of Luxembourg [103] (see `hpc.uni.lu`). Besides that, authors Tsz Pan Tong and Jun Pang acknowledge financial support of the Institute for Advanced Studies of the University of Luxembourg through an Audacity Grant (AUDACITY-2021).

Q4  **Any other comments?**

No.

## A.2 Composition

Q5 **What do the instances that comprise the dataset represent (for example, documents, photos, people, countries)?** Are there multiple types of instances (for example, movies, users, and ratings; people and interactions between them; nodes and edges)? Please provide a description.

The instances represent time-series features of nodes (trajectories) in a period of time, and the corresponding ground-truth interaction graph. The instances are all `.npy` files. Each time-series feature of nodes was produced by the simulation of dynamical systems with the simulation code included in the GitHub repository.

Q6 **How many instances are there in total (of each type, if appropriate)?**

The dataset has a total of 20,858 `.npy` files.

Q7 **Does the dataset contain all possible instances or is it a sample (not necessarily random) of instances from a larger set?** If the dataset is a sample, then what is the larger set? Is the sample representative of the larger set (for example, geographic coverage)? Is the sample representative of the larger set (e.g., geographic coverage)? If so, please describe how this representativeness was validated/verified. If it is not representative of the larger set, please describe why not (e.g., to cover a more diverse range of instances, because instances were withheld or unavailable).

The dataset contains all possible instances.

Q8 **What data does each instance consist of?** "Raw" data (for example, unprocessed text or images) or features? In either case, please provide a description.

The instance consists of synthetic "Raw" data of node features in a time period and the underlying ground-truth interaction graphs. The instance also contains one set of processed EMT dataset and its suggested underlying GRN. All are in `.npy` format.

The composition of the whole dataset consists of twelve folders representing eleven types of underlying interacting graphs and the EMT dataset.

The folder for the EMT dataset, `emt36_grn_network`, contains the following files:

- Trajectory for training: `traj_EMT50cc36_wot_interpolated_by_pchip_22t.npy`
- Reference GRN: `edges_EMT36.npy`

The folders for eleven types of underlying interacting graphs are:

- `brain_networks`,
- `chemical_reaction_networks_in_atmosphere`,
- `food_webs`,
- `gene_coexpression_networks`,
- `gene_regulatory_networks`,
- `intercellular_networks`,
- `landscape_networks`,
- `man-made_organic_reaction_networks`,
- `reaction_networks_inside_living_organism`,
- `social_networks`,
- `vascular_networks`.

Each of these folders has a subfolder named either `directed` or `undirected`, which contains the trajectories for either directed graphs or undirected graphs based on the type of the graphs. Then in these subfolders, the data can be divided into two groups based on the type of dynamical simulations: Springs or NetSims. So every subfolder only contains the data generated by either simulation:

– Generated by Springs simulation. (The subfolder is named as `springs`.) For instance, for a graph of `K` nodes and noted as the `R`-th repetition of its group, the instances in the same subfolder which belong to this simulation are:
  * Trajectories for training:
    `loc_train_springsKrR.npy`, `vel_train_springsKrR.npy`,
  * Groundtruth graphs for training:
    `edges_train_springsKrR.npy`,
  * Trajectories for validation:
    `loc_valid_springsKrR.npy`, `vel_valid_springsKrR.npy`,
  * Groundtruth graphs for validation:
    `edges_valid_springsKrR.npy`,
  * Trajectories for test:
    `loc_test_springsKrR.npy`, `vel_test_springsKrR.npy`,
  * Groundtruth graphs for test:
    `edges_test_springsKrR.npy`.
– Generated by NetSims simulation. (The subfolder is named as `netsims`.) For instance, for a graph of `K` nodes and noted as the `R`-th repetition of its group, the instances in the same subfolder which belong to this simulation are:
  * Trajectories for training:
    `bold_train_netsimsKrR.npy`,
  * Groundtruth graphs for training:
    `edges_train_netsimsKrR.npy`,
  * Trajectories for training:
    `bold_valid_netsimsKrR.npy`,
  * Groundtruth graphs for training:
    `edges_valid_netsimsKrR.npy`,
  * Trajectories for test:
    `bold_test_netsimsKrR.npy`,
  * Groundtruth graphs for test:
    `edges_test_netsimsKrR.npy`.

For trajectories with `L` level of Gaussian noise, they are marked with additional subscripts `_nL` at the end of its corresponding noise-free trajectories (before `.npy`).

Q9 **Is there a label or target associated with each instance?** If so, please provide a description.

Each instance has a corresponding ground-truth interaction graph that is used to generate the set of trajectories.

Q10 **Is any information missing from individual instances?** If so, please provide a description.

No.

Q11 **Are relationships between individual instances made explicit (for example, users' movie ratings, social network links)?** If so, please describe how these relationships are made explicit.

We divided the files into groups on the basis of the type of underlying interacting graph, and subsequently on the dynamic functions of the trajectories generation.

Q12 **Are there recommended data splits (for example, training, development/validation, testing)?** If so, please provide a description of these splits, explaining the rationale behind them.

We have already split the data into training sets, validation sets, and testing sets with ratios of 8: 2: 2 based on the counts of trajectories. All of them are open to audiences.

Q13 **Are there any errors, sources of noise, or redundancies in the dataset?** If so, please provide a description.

Yes, besides the generated raw trajectories, we also provided noisy trajectories. The noisy trajectories are the raw ones added with Gaussian noises of different levels. For example, the files with `xx_n5.npy` are the noisy trajectories obtained from `xx.npy` with 5 levels of additive Gaussian noise. The noises were only added to the trajectories, not the ground-truth interaction graphs.

Q14 **Is the dataset self-contained, or does it link to or otherwise rely on external resources (for example, websites, tweets, other datasets)?** If it links to or relies on external resources, a) are there guarantees that they will exist, and remain constant, over time; b) are there official archival versions of the complete dataset (i.e., including the external resources as they existed at the time the dataset was created); c) are there any restrictions (e.g., licenses, fees) associated with any of the external resources that might apply to a future user? Please provide descriptions of all external resources and any restrictions associated with them, as well as links or other access points, as appropriate.

Yes, the dataset is self-contained.

Q15 **Does the dataset contain data that might be considered confidential (for example, data that is protected by legal privilege or by doctor-patient confidentiality, data that includes the content of individuals' non-public communications)?** If so, please provide a description.

We allow free distribution of the dataset.

Q16 **Does the dataset contain data that, if viewed directly, might be offensive, insulting, threatening, or might otherwise cause anxiety?** If so, please describe why.

No.

## A.3 Collection process

Q17 **How was the data associated with each instance acquired?** Was the data directly observable (for example, raw text, movie ratings), reported by subjects (for example, survey responses), or indirectly inferred/derived from other data (for example, part-of-speech tags, model-based guesses for age or language)? If the data was reported by subjects or indirectly inferred/derived from other data, was the data validated/verified? If so, please describe how.

We first generated ground-truth interaction graphs following the sampled ranges of properties of eleven types of real-world graphs, which include: brain networks, chemical reaction networks in the atmosphere, food webs, gene coexpression networks, gene regulatory networks, intercellular networks, landscape networks, man-made organic reaction networks, reaction networks inside living organism, social networks, and vascular networks. Among these, the graphs from gene coexpression networks and landscape networks are undirected, while the rest are directed. We generated the graphs with different counts of nodes: 15, 30, 50, 100, 150, 200, and 250. And we generated graphs of each size with 3 repetitions while ensuring that the three were not identical. In total, we generated 231 ground truth interacting graphs.

Then we ran simulations based on the generated ground truth interaction graphs. There were two types of dynamic simulations, "Springs" and "NetSims". Every ground truth interaction graph joined the simulation and in total, we obtained 462 sets of trajectories.

After that, we created another set of trajectories with the addition of Gaussian noises. The Gaussian noises were added to the generated trajectories with 5 different amplifying levels. In total, we generated 2310 sets of trajectories with Gaussian noises.

Q18 **What mechanisms or procedures were used to collect the data (for example, hardware apparatuses or sensors, manual human curation, software programs, software APIs)?** How were these mechanisms or procedures validated?

The whole data generation process was run on Amazon EC2 C7g.2xlarge instances, which are powered by AWS Graviton3 processors, and are provided to the University of Luxembourg as a part of the collaboration. We first ran a Python script for graph generation, over 32 vCPUs of C7g.2xlarge instances, and with 128 GB RAM. Then we fed the generated graphs to the Python script for dynamic simulations with the same hardware settings. The generated graphs were validated by manual inspection and post-processed with the computation of statistics on the degrees, connectivity, number of self-loops, clustering coefficients, and average shortest paths.

Q19 **If the dataset is a sample from a larger set, what was the sampling strategy (for example, deterministic, probabilistic with specific sampling probabilities)?**

No. The dataset is not a sample from a larger set.

Q20 **Who was involved in the data collection process (for example, students, crowdworkers, contractors) and how were they compensated (for example, how much were crowd-workers paid)?**

No crowdworkers were used in the curation of the dataset. One of the authors of this paper, Aoran Wang, was involved in the data collection process.

Q21 **Over what timeframe was the data collected? Does this timeframe match the creation timeframe of the data associated with the instances (for example, recent crawl of old news articles)?** If not, please describe the timeframe in which the data associated with the instances was created.

The data was collected in the period from December 15, 2022 to March 3, 2023.

Q22 **Were any ethical review processes conducted (for example, by an institutional review board)?** If so, please provide a description of these review processes, including the outcomes, as well as a link or other access point to any supporting documentation.

No, such processes were unnecessary in our case.

Q23 **Does the dataset relate to people?** If not, you may skip the remaining questions in this section.

No.

## A.4 Preprocessing/Cleaning/Labeling

Q24 **Was any preprocessing/cleaning/labeling of the data done (for example, discretization or bucketing, tokenization, part-of-speech tagging, SIFT feature extraction, removal of instances, processing of missing values)?** If so, please provide a description. If not, you may skip the remainder of the questions in this section.

We conducted preprocessing only on the EMT dataset, and the details of the preprocessing can be found in Appendix B.4.

Q25 **Was the "raw" data saved in addition to the preprocessed/cleaned/labeled data (for example, to support unanticipated future uses)?** If so, please provide a link or other access point to the "raw" file.

The "raw" data was not saved together with the preprocessed data. The "raw" data is available in `https://www.ncbi.nlm.nih.gov/geo/query/acc.cgi?acc=GSE147405`, and the EMT data processed by Sha et al. [92] is available in `https://github.com/yutongo/TIGON/tree/19d6648195a47b4d2a2d5025b440d37cf0ac9a17/EMT_data`. Our EMT dataset was built based on the data provided by Sha et al., and the postprocessing script is available in our code repository.

Q26 **Is the software that was used to preprocess/clean/label the data available?** If so, please provide a link or other access point.

Starting with the EMT dataset provided by Sha et al., the following software are needed to reproduce our EMT dataset:

- Anndata [104]: `https://github.com/scverse/anndata`
- NetworkX [43]: `https://networkx.org/`
- NumPy [45]: `https://numpy.org/`
- pandas [109]: `https://pandas.pydata.org/`
- Requests: `https://requests.readthedocs.io/en/latest/`
- SciPy [105]: `https://scipy.org/`
- wot [91]: `https://broadinstitute.github.io/wot/`

Q27 **Any other comments?**

No.

## A.5 Uses

Q28 **Has the dataset been used for any tasks already?**

No. The dataset has not been used for any tasks yet.

Q29 **Is there a repository that links to any or all papers or systems that use the dataset?** If so, please provide a link or other access point.

No. The dataset has not been used for any tasks yet.

Q30 **What (other) tasks could the dataset be used for?**

The dataset could be used for time-series prediction and possibly the task of graph completeness.

Q31 **Is there anything about the composition of the dataset or the way it was collected and preprocessed/cleaned/labeled that might impact future uses?** For example, is there anything that a future user might need to know to avoid uses that could result in unfair treatment of individuals or groups (e.g., stereotyping, quality of service issues) or other undesirable harms (e.g., financial harms, legal risks) If so, please provide a description. Is there anything a future user could do to mitigate these undesirable harms?

We do not think the composition of the dataset or the way it was collected or preprocessed/cleaned/labeled could impact future uses.

Q32 **Are there tasks for which the dataset should not be used?** If so, please provide a description.

Due to the known biases of the dataset, under no circumstance should any methods be put into production using the dataset as is. It is neither safe nor responsible. As it stands, the dataset should be solely used for research purposes in its uncurated state. Likewise, this dataset should not be used to aid in military or surveillance tasks.

Q33 **Any other comments?**

No.

## A.6 Distribution

Q34 **Will the dataset be distributed to third parties outside of the entity (e.g., company, institution, organization) on behalf of which the dataset was created?** If so, please provide a description.

Yes, the dataset will be open-source.

Q35 **How will the dataset be distributed (e.g., tarball on website, API, GitHub)?** Does the dataset have a digital object identifier (DOI)?

The data will be available through the website of this benchmark (`https://structinfer.github.io/download/`). For the large subsets such as the ones with more than 100 nodes, please contact the authors and the authors will provide a link to download them.

Q36 **When will the dataset be distributed?**

May 31, 2023 and onward.

Q37 **Will the dataset be distributed under a copyright or other intellectual property (IP) license, and/or under applicable terms of use (ToU)?** If so, please describe this license and/or ToU, and provide a link or other access point to, or otherwise reproduce, any relevant licensing terms or ToU, as well as any fees associated with these restrictions.

CC-BY-4.0

Q38 **Have any third parties imposed IP-based or other restrictions on the data associated with the instances?** If so, please describe these restrictions, and provide a link or other access point to, or otherwise reproduce, any relevant licensing terms, as well as any fees associated with these restrictions.

No.

Q39 **Do any export controls or other regulatory restrictions apply to the dataset or to individual instances?** If so, please describe these restrictions, and provide a link or other access point to, or otherwise reproduce, any supporting documentation.

No.

Q40 **Any other comments?**

We managed to upload the part of DoSI that are essential for the reproduction of the results in this benchmark paper. However, as the total size of the DoSI exceeds 7.8 TB, we are communicating with our grant provider on the publishing of the remaining dataset. At the moment, the website contains the subsets with no more than 100 nodes. For the subsets with more than 100 nodes, please contact authors and the authors will provide a link to access the data.

## A.7 Maintenance

Q41 **Who will be supporting/hosting/maintaining the dataset?**

The research group SaToSS at the University of Luxembourg hosts the dataset. The authors of this paper will also support and maintain the dataset.

Q42 **How can the owner/curator/manager of the dataset be contacted (e.g., email address)?**

Contact the authors at `aoran.wang@uni.lu`, or `tszpan.tong@uni.lu`.

Q43 **Is there an erratum?** If so, please provide a link or other access point.

There is no erratum for our initial release. Errata will be documented as future releases on the dataset website.

Q44 **Will the dataset be updated (e.g., to correct labeling errors, add new instances, delete instances)?** If so, please describe how often, by whom, and how updates will be communicated to users (e.g., mailing list, GitHub)?

We are planning to extend the dataset to ensure benchmark results with the highest statistical credibility. Such updates will be rare, as they involve subjective evaluation — a time-consuming task that requires extensive preparation. Also, we understand the problems that consumers can face during updates. But after updates become public, they will receive notification primarily through the mailing list, and all the new information will be on the benchmark website.

Q45 **If the dataset relates to people, are there applicable limits on the retention of the data associated with the instances (e.g., were individuals in question told that their data would be retained for a fixed period of time and then deleted)?** If so, please describe these limits and explain how they will be enforced.

No, the dataset does not relate to people.

Q46 **Will older versions of the dataset continue to be supported/hosted/maintained?** If so, please describe how. If not, please describe how its obsolescence will be communicated to users.

We will continue to support the older versions as long as we have enough funds.

Q47 **If others want to extend/augment/build on/contribute to the dataset, is there a mechanism for them to do so?** If so, please provide a description. Will these contributions be validated/verified? If so, please describe how. If not, why not? Is there a process for communicating/distributing these contributions to other users? If so, please provide a description.

We encourage everyone to share their ideas on extending our dataset to cover more compression cases and provide more reliable results. Our method of subjective quality evaluation, however, is set; we recommend researchers contact us by `aoran.wang@uni.lu`, or `tszpan.tong@uni.lu` to coordinate subjective quality evaluation.

Q48 **Any other comments?**

No.

## B   Further details of datasets

In this section, we provide more details about the datasets used in this work. We first describe the generation of DoSI dataset, and then provide details on EMT dataset. The generation of DoSI dataset consists of two steps: (1) generating underlying interaction graphs based on the properties of real-world graphs (Appendix B.1 and (2) simulating the dynamical systems (Appendix B.2). Besides, we provide the quality evaluation of DoSI in Appendix B.3.

## B.1 Underlying interaction graphs

The sampled properties of each type of graph are summarised in Table 1. Some values are missing because they were not reported in the literature [8, 7, 32]. In the next paragraphs, we briefly describe

Table 1: Sampled properties of 11 types of real-world graphs.

| Graphs | Properties | | | | | |
|--------|------------|------------|------------|------------|---------------|------------|
| | $C$ | $d$ | $\gamma$ | $\langle k \rangle$ | $\delta$ | $\gamma^{\text{out}}$ |
| BN | - | - | - | $[1.8, 2.0]$ | $[0.002, 0.25]$ | - |
| CRNA | $[0.25, 0.62]$ | $[1.5, 2.8]$ | - | - | $[0.02, 0.32]$ | - |
| FW | - | $[1.5, 2.5]$ | - | - | - | - |
| GCN | $[0.05, 0.45]$ | $[2.5, 5.2]$ | $[1.2, 2.4]$ | - | - | - |
| GRN | $[0.08, 0.25]$ | $[1.7, 4.0]$ | - | - | - | - |
| IN | - | $[2.0, 3.4]$ | - | - | - | - |
| LN | $[0.6, 0.8]$ | - | - | - | - | - |
| MMO | - | - | - | $[2.0, 3.6]$ | - | $[1.5, 2.6]$ |
| RNLO | - | - | - | $[2.1, 3.0]$ | - | - |
| SN | $[0.09, 0.20]$ | $[2.0, 4.2]$ | - | - | $[0.095, 0.15]$ | - |
| VN | - | - | $[3.7, 3.8]$ | $[1.5, 2.2]$ | - | - |

the generation of the underlying interaction graphs and the corresponding implementation. Each paragraph will discuss the generation and implementation of each type of underlying interaction graph, respectively. We use $N$ to denote the number of nodes.

**Brain Networks (BN).** BNs are the networks that represent the connectivity of brain regions, which can be determined by anatomical tracts or by functional associations [32]. In addition to the collected properties presented in Table 1, the structure of brain networks also shows a remarkable hierarchical structure. Therefore, we generate the directed BN graphs of the total number of nodes equal to $N$ by first creating a set of growing networks, each with 5 nodes. Then, we randomly connect the growing networks to obtain a connected graph. The pipeline is implemented with the Python Package NetworkX [43]. Specifically, we use the `gn_graph` function for growing network creation and the `k_edge_augmentation` function for connecting growing networks. Since there are many hyperparameters in the pipeline, we create a search space for these parameters and record the first three graphs whose properties are within the range of the ones in Table 1.

**Chemical Reaction Networks in the Atmosphere (CRNA).** A CRNA models the complex network of reaction transformations in the atmosphere of planets. There is a link from chemical $i$ to chemical $j$ if the former is a reactant and the latter is a product in at least one chemical reaction. CRNAs exhibit both small-worldness and randomness [32]. In this work, the directed CRNA graphs are generated by using the directed Erdös-Rényi graph generator of NetworkX [43]: `erdos_renyi_graph`. The argument `n` of the function is set to the total number of nodes $N$, and the argument `p` is set to a value from the search space $[0.05, 0.75]$. During the search, we record the first three graphs whose properties are in the ranges shown in Table 1.

**Food Webs (FW).** FWs are networks that describe the 'networks of feeding interactions among diverse co-occurring species in a particular habitat' [96]. It is widely accepted that 'empirical food webs' display exponential or uniform degree distributions. Therefore, in this work, the directed FW graphs are generated using a two-step procedure. We first sample the in-/out-degree sequences from an exponential function (`random.exponential` from Python library NumPy [45]) with different scales. The scales are computed by dividing $N$ by a hyperparameter from a search space. Then the in-/out-degree sequences are given to the directed configuration model generator of NetworkX: `directed_configuration_model`. During the search, we record the first three graphs whose properties are within the range shown in Table 1.

**Gene Coexpression Networks (GCN).** Two genes that have similar expression profiles are likely to have similar functions. Gene coexpression networks are built by calculating a similarity score for each pair of genes. The nodes of the networks represent the genes, and two genes are linked if

their similarity is above a certain threshold. GCNs are characterized by both 'small-worldness' and 'scale-freeness' features [32]. The undirected GCN graphs in this work are generated with three steps. We first sample the sequence of node degrees from `utils.powerlaw_sequence` of NetworkX. The argument `exponent` is a hyperparameter with value from the search space $[1.4, 2.8]$. Then, the sequence is given to `configuration_model` of NetworkX to generate a graph. However, the generated graph might have multiple disconnected components. We use the `k_edge_augmentation` function of NetworkX to connect the components with the argument k as another hyperparameter from the search space $[1, 10]$. During the search, we record the first three graphs whose properties are within the ranges shown in Table 1.

**Gene Regulatory Networks (GRN).**    GRN is another type of gene network in which the connections are between transcription factors and the genes that they regulate. In this work, the directed GRN graphs are generated by an open-source Python package: `https://github.com/zhivkoplias/network_generation_algo`. The package implements a graph generation algorithm with boosted feed-forward loop motif, which is known to be important for network dynamics. We change the `final_size` argument in the script to match $N$. Since there is a random process incorporated in the generation process, we run the script several times until we obtain three graphs whose properties fall in the ranges shown in Table 1.

**Intercellular Networks (IN).**    IN was studied to describe the topological organization produced by the spatial relationship among cells in different tissues. In this work, we follow the setup-principle of probabilistic cell graphs [32], where a link between two cells is established with a probability function of the Euclidean distance between them. In this work, we follow a simplified process. The directed IN graphs are generated by calling the directed Erdős-Rényi graph generator of NetworkX: `erdos_renyi_graph`. The argument n of the function is set to the total count of nodes $N$, and the argument p is set to a value from the search space $[0.05, 0.75]$. During the search, we record the first three graphs for each whose properties fall in the ranges shown in Table 1.

**Landscape Networks (LN).**    LNs are used to model the interconnectivity among the spatial pattern of scattered habitat patches in the landscape. They are similar to the random geometric networks [32]. In this work, the undirected LN graphs are generated using the `geographical_threshold_graph` function of NetworkX. The argument `theta` is set to a value computed with the multiplication of $N$ and a hyperparameter, which is selected from the search space $[0.5, 2.0]$. During the search, we record the first three graphs whose properties are within the range shown in Table 1.

**Man-made Organic Reaction Networks (MMO).**    A chemical reaction transforms one or more reactants into one or more products. A chemical $i$ is linked to a chemical $j$ if they are a reactant and a product, respectively, in any chemical reaction. It is observed that the in-degree and out-degree of the molecules follow power-law distributions [32]. We use the `scale_free_graph` generator of NetworkX to generate directed MMO graphs based on this property. We set `alpha`, `beta`, `delta_in`, and `delta_out` as hyperparameters with search spaces of $[0.01, 0.97]$, $[0.01, 0.98]$, $[0.01, 0.4]$, and $[0, 0.15]$, respectively. We calculate `gamma` by $1 - $ `alpha` $- $ `beta`. We convert the raw graphs to directed graphs using the `DiGraph` function. We select the first three directed graphs that match the properties in Table 1.

**Reaction Networks inside Living Organisms (RNLO).**    The RNLO graphs and MMO graphs have many similar properties, because they are both chemical reaction networks. We generate the directed RNLO graphs using the same pipeline as MMO, but with different property ranges presented in Table 1.

**Social Networks (SN).**    An SN is conceptualized as a graph, that is, a set of vertices (or nodes, units, points) representing social actors and a set of lines representing one or more social relations among them [29]. We use the `gnp_random_graph` generator of NetworkX to generate directed SN graphs. We set p as a hyperparameter with a search space of $[0.01, 0.99]$. We select the first three directed graphs that match the properties in Table 1.

**Vascular Networks (VN).**    A VN is a graph where nodes represent the junctions of channels and edges represent the connections between them. VNs have power-law degree distributions [32]. We

generate directed VN graphs by first creating a tree with a power-law degree distribution using the `random_powerlaw_tree` generator of NetworkX, where we set `gamma` as a hyperparameter with a search space of $[1.5, 4.9]$. We then convert the trees to directed graphs using the `DiGraph` function. We select the first three directed graphs that match the properties in Table 1.

We summarize the properties of the underlying interaction graphs mentioned in this work in Tables 2 - 12. The graphs are aligned in accordance with the type of graphs they belong to. The names of the graphs are represented as "number of nodes in the graph" + "repetition number". For example, the second repetition of the graph with 15 nodes is represented as `15r2`. In the tables, # Nodes denotes the number of nodes in the graph, # Edges denotes the number of edges in the graph, $C$ is the average clustering coefficient, $d$ is the average shortest path length, $\gamma$ is the power-law exponent of the degree distribution, $\langle k \rangle$ is the average node degree, $\delta$ is the density, and $\gamma^{in}$ and $\gamma^{out}$ are the power-law exponents of the in-degree/out-degree distributions, respectively. Among these metrics, # Nodes, # Edges, $C$, $d$, and $\delta$ are calculated by built-in functions of NetworkX. $\langle k \rangle$ is calculated by averaging over all node degrees in the graph, which is obtained by calling `.degree` with NetworkX. The power-law exponents are calculated by fitting the corresponding degree sequences with `powerlaw.Fit` of Python package powerlaw, then by outputting the `.powerlaw.alpha` variables of the obtained distributions. It is worth mentioning that some exponents are missing, where `powerlaw` could not find a suitable powerlaw function to fit or where the sampled degree sequence is too short for fitting. As shown in Tables 2 - 12, the properties of the graphs vary significantly, and the investigation of to which extent the different underlying graphs influence the performance of structural inference methods is worth studying.

Table 2: Properties of underlying interaction graphs of brain networks.

| Name | Properties | | | | | | | | |
|------|---------|---------|-----|-------|-------|-----------------|-------|-------------|--------------|
|      | # Nodes | # Edges | $C$ | $d$   | $\gamma$ | $\langle k \rangle$ | $\delta$ | $\gamma^{in}$ | $\gamma^{out}$ |
| 15r1 | 15 | 14 | 0.0 | 2.88 | 3.43 | 1.87 | 0.07 | 2.09 | - |
| 15r2 | 15 | 14 | 0.0 | 2.99 | 3.32 | 1.87 | 0.07 | 6.87 | - |
| 15r3 | 15 | 14 | 0.0 | 2.95 | 3.43 | 1.87 | 0.07 | 6.87 | - |
| 30r1 | 30 | 29 | 0.0 | 4.51 | 4.43 | 1.93 | 0.03 | 3.04 | 41.40 |
| 30r2 | 30 | 29 | 0.0 | 4.09 | 3.35 | 1.93 | 0.03 | 14.44 | 41.40 |
| 30r3 | 30 | 29 | 0.0 | 4.27 | 3.16 | 1.93 | 0.03 | 2.81 | 41.40 |
| 50r1 | 50 | 49 | 0.0 | 5.90 | 5.96 | 1.96 | 0.02 | 4.91 | 34.90 |
| 50r2 | 50 | 49 | 0.0 | 5.71 | 3.19 | 1.96 | 0.02 | 3.73 | 34.90 |
| 50r3 | 50 | 49 | 0.0 | 6.01 | 3.01 | 1.96 | 0.02 | 5.05 | 34.90 |
| 100r1 | 100 | 99 | 0.0 | 9.13 | 3.07 | 1.98 | 0.01 | 2.60 | 35.26 |
| 100r2 | 100 | 99 | 0.0 | 9.05 | 3.11 | 1.98 | 0.01 | 2.72 | 35.26 |
| 100r3 | 100 | 99 | 0.0 | 9.11 | 3.07 | 1.98 | 0.01 | 16.68 | 35.26 |
| 150r1 | 150 | 149 | 0.0 | 12.56 | 3.04 | 1.99 | 0.007 | 4.55 | 26.43 |
| 150r2 | 150 | 149 | 0.0 | 12.75 | 13.63 | 1.99 | 0.007 | 11.44 | 26.43 |
| 150r3 | 150 | 149 | 0.0 | 12.76 | 8.77 | 1.99 | 0.007 | 5.56 | 26.43 |
| 200r1 | 200 | 199 | 0.0 | 16.07 | 11.74 | 1.99 | 0.005 | 5.29 | 28.27 |
| 200r2 | 200 | 199 | 0.0 | 15.96 | 2.99 | 1.99 | 0.005 | 8.15 | 28.27 |
| 200r3 | 200 | 199 | 0.0 | 15.98 | 2.96 | 1.99 | 0.005 | 8.15 | 28.27 |
| 250r1 | 250 | 249 | 0.0 | 19.37 | 24.77 | 1.992 | 0.004 | 21.91 | 29.49 |
| 250r2 | 250 | 249 | 0.0 | 19.21 | 3.06 | 1.992 | 0.004 | 19.82 | 29.49 |
| 250r3 | 250 | 249 | 0.0 | 19.31 | 2.99 | 1.992 | 0.004 | 7.39 | 29.49 |

The corresponding code for graph generation can be found at `https://github.com/wang422003/Benchmarking-Structural-Inference-Methods-for-Interacting-Dynamical-Systems/tree/main/src/graphs`. The corresponding scripts for the generation of each graph type are summarized in Table 13.

Table 3: Properties of underlying interaction graphs of chemical reaction networks in the atmosphere.

| Name | Properties | | | | | | | | |
|---|---|---|---|---|---|---|---|---|---|
| | # Nodes | # Edges | $C$ | $d$ | $\gamma$ | $\langle k \rangle$ | $\delta$ | $\gamma^{in}$ | $\gamma^{out}$ |
| 15r1 | 15 | 40 | 0.26 | 2.78 | 3.02 | 5.33 | 0.19 | 4.57 | 2.29 |
| 15r2 | 15 | 46 | 0.25 | 2.22 | 12.00 | 6.13 | 0.22 | 6.84 | 5.17 |
| 15r3 | 15 | 54 | 0.26 | 2.00 | 4.82 | 7.2 | 0.26 | 4.76 | 28.42 |
| 30r1 | 30 | 208 | 0.26 | 1.90 | 5.77 | 13.87 | 0.24 | 7.29 | 9.57 |
| 30r2 | 30 | 205 | 0.26 | 1.93 | 5.82 | 13.67 | 0.24 | 4.70 | 4.87 |
| 30r3 | 30 | 203 | 0.25 | 1.97 | 7.52 | 13.53 | 0.23 | 11.27 | 4.24 |
| 50r1 | 50 | 591 | 0.25 | 1.80 | 15.90 | 23.64 | 0.24 | 10.63 | 9.77 |
| 50r2 | 50 | 611 | 0.25 | 1.79 | 11.16 | 24.44 | 0.25 | 7.67 | 11.19 |
| 50r3 | 50 | 605 | 0.25 | 1.79 | 11.07 | 24.2 | 0.25 | 8.52 | 17.02 |
| 100r1 | 100 | 2,510 | 0.26 | 1.75 | 14.90 | 50.2 | 0.25 | 10.54 | 14.50 |
| 100r2 | 100 | 2,485 | 0.25 | 1.75 | 33.90 | 49.7 | 0.25 | 7.21 | 195.98 |
| 100r3 | 100 | 2,527 | 0.25 | 1.75 | 16.34 | 50.54 | 0.26 | 21.52 | 11.60 |
| 150r1 | 150 | 5,729 | 0.26 | 1.74 | 22.89 | 76.39 | 0.26 | 19.25 | 19.53 |
| 150r2 | 150 | 5,804 | 0.26 | 1.74 | 15.83 | 77.39 | 0.26 | 28.46 | 12.55 |
| 150r3 | 150 | 5,614 | 0.25 | 1.75 | 17.12 | 74.85 | 0.26 | 17.11 | 20.10 |
| 200r1 | 200 | 10,108 | 0.25 | 1.75 | 38.65 | 101.08 | 0.25 | 17.15 | 21.61 |
| 200r2 | 200 | 10,337 | 0.26 | 1.74 | 66.57 | 103.37 | 0.26 | 31.54 | 284.48 |
| 200r3 | 200 | 10,254 | 0.26 | 1.74 | 26.23 | 102.54 | 0.26 | 17.98 | 48.32 |
| 250r1 | 250 | 15,944 | 0.26 | 1.74 | 40.62 | 127.55 | 0.26 | 22.35 | 26.00 |
| 250r2 | 250 | 16,119 | 0.26 | 1.74 | 28.17 | 128.95 | 0.26 | 19.57 | 22.78 |
| 250r3 | 250 | 15,938 | 0.26 | 1.74 | 56.66 | 127.50 | 0.26 | 24.15 | 20.49 |

Table 4: Properties of underlying interaction graphs of food webs.

| Name | Properties | | | | | | | | |
|---|---|---|---|---|---|---|---|---|---|
| | # Nodes | # Edges | $C$ | $d$ | $\gamma$ | $\langle k \rangle$ | $\delta$ | $\gamma^{in}$ | $\gamma^{out}$ |
| 15r1 | 15 | 96 | 0.51 | 1.69 | 3.73 | 20.13 | 0.46 | 29.47 | 48.46 |
| 15r2 | 15 | 100 | 0.49 | 1.72 | 7.02 | 20.4 | 0.48 | 20.61 | 1.87 |
| 15r3 | 15 | 88 | 0.58 | 1.73 | 23.76 | 17.6 | 0.42 | 2.40 | 20.93 |
| 30r1 | 30 | 404 | 0.51 | 1.57 | 12.22 | 41.27 | 0.46 | 98.48 | 117.97 |
| 30r2 | 30 | 309 | 1.42 | 1.86 | 12.24 | 33.27 | 0.36 | 16.15 | 13.28 |
| 30r3 | 30 | 329 | 0.47 | 1.86 | 3.25 | 36.53 | 0.38 | 10.47 | 3.93 |
| 50r1 | 50 | 1,084 | 0.51 | 1.61 | 4.18 | 68.6 | 0.44 | 19.66 | 18.47 |
| 50r2 | 50 | 1,005 | 0.52 | 1.73 | 23.17 | 65.24 | 0.41 | 19.41 | 29.10 |
| 50r3 | 50 | 1,081 | 0.50 | 1.60 | 32.83 | 68.2 | 0.44 | 85.54 | 34.72 |
| 100r1 | 100 | 4,161 | 0.50 | 1.63 | 61.69 | 134.96 | 0.42 | 58.86 | 35.41 |
| 100r2 | 100 | 3,978 | 0.50 | 1.67 | 29.08 | 127.26 | 0.40 | 35.08 | 25.24 |
| 100r3 | 100 | 4,092 | 0.48 | 1.62 | 3.92 | 129.96 | 0.41 | 64.89 | 53.87 |
| 150r1 | 150 | 8,658 | 0.48 | 1.67 | 51.93 | 181.85 | 0.39 | 35.03 | 21.72 |
| 150r2 | 150 | 9,355 | 0.4 | 1.60 | 24.08 | 196.67 | 0.42 | 102.89 | 55.11 |
| 150r3 | 150 | 8,924 | 0.48 | 1.65 | 89.06 | 188.59 | 0.40 | 78.48 | 138.23 |
| 200r1 | 200 | 16,885 | 0.50 | 1.60 | 73.29 | 268.35 | 0.42 | 45.44 | 57.85 |
| 200r2 | 200 | 17,279 | 0.51 | 1.58 | 80.59 | 274.72 | 0.43 | 65.31 | 55.84 |
| 200r3 | 200 | 15,069 | 0.49 | 1.65 | 43.84 | 243.89 | 0.38 | 86.15 | 40.40 |
| 250r1 | 250 | 23,669 | 0.46 | 1.63 | 86.40 | 300.02 | 0.38 | 58.10 | 22.70 |
| 250r2 | 250 | 25,569 | 0.49 | 1.62 | 91.49 | 327.32 | 0.41 | 89.91 | 55.57 |
| 250r3 | 250 | 24,596 | 0.48 | 1.63 | 3.59 | 315.344 | 0.40 | 56.17 | 54.21 |

Table 5: Properties of underlying interaction graphs of gene coexpression networks.

| Name | Properties | | | | | | |
|------|---------|---------|------|------|----------|------------------|------|
|      | # Nodes | # Edges | $C$ | $d$ | $\gamma$ | $\langle k \rangle$ | $\delta$ |
| 15r1 | 15 | 22 | 0.069 | 2.55 | 2.18 | 2.93 | 0.21 |
| 15r2 | 15 | 22 | 0.069 | 2.55 | 2.18 | 2.93 | 0.21 |
| 15r3 | 15 | 23 | 0.24 | 2.50 | 2.28 | 3.07 | 0.22 |
| 30r1 | 30 | 60 | 0.21 | 2.66 | 1.98 | 4.0 | 0.14 |
| 30r2 | 30 | 53 | 0.16 | 2.58 | 2.19 | 3.53 | 0.12 |
| 30r3 | 30 | 54 | 0.15 | 2.77 | 2.12 | 3.6 | 0.12 |
| 50r1 | 50 | 128 | 0.26 | 2.56 | 2.20 | 5.12 | 0.10 |
| 50r2 | 50 | 160 | 0.28 | 2.52 | 1.77 | 6.4 | 0.13 |
| 50r3 | 50 | 112 | 0.30 | 2.55 | 2.05 | 4.48 | 0.09 |
| 100r1 | 100 | 342 | 0.28 | 2.63 | 1.78 | 6.84 | 0.07 |
| 100r2 | 100 | 364 | 0.33 | 2.61 | 1.80 | 7.28 | 0.074 |
| 100r3 | 100 | 364 | 0.34 | 2.61 | 1.80 | 7.28 | 0.073 |
| 150r1 | 150 | 729 | 0.35 | 2.51 | 2.01 | 9.72 | 0.065 |
| 150r2 | 150 | 729 | 0.34 | 2.51 | 2.01 | 9.72 | 0.065 |
| 150r3 | 150 | 670 | 0.33 | 2.70 | 1.84 | 8.93 | 0.06 |
| 200r1 | 200 | 1,018 | 0.34 | 2.55 | 1.79 | 10.18 | 0.05 |
| 200r2 | 200 | 1,018 | 0.34 | 2.55 | 1.78 | 10.18 | 0.05 |
| 200r3 | 200 | 1,041 | 0.34 | 2.54 | 2.04 | 10.42 | 0.05 |
| 250r1 | 250 | 1,596 | 0.35 | 2.54 | 1.80 | 12.77 | 0.05 |
| 250r2 | 250 | 1,596 | 0.35 | 2.54 | 1.80 | 12.77 | 0.05 |
| 250r3 | 250 | 1,627 | 0.35 | 2.53 | 1.90 | 13.02 | 0.05 |

Table 6: Properties of underlying interaction graphs of gene regulatory networks.

| Name | Properties | | | | | | | | |
|------|---------|---------|------|------|----------|---------------------|----------|----------------|-----------------|
|      | # Nodes | # Edges | $C$ | $d$ | $\gamma$ | $\langle k \rangle$ | $\delta$ | $\gamma^{in}$ | $\gamma^{out}$ |
| 15r1 | 15 | 32 | 0.26 | 2.02 | 3.43 | 4.27 | 0.15 | 5.38 | 6.10 |
| 15r2 | 15 | 32 | 0.26 | 2.02 | 3.64 | 4.27 | 0.15 | 14.44 | 4.04 |
| 15r3 | 15 | 38 | 0.25 | 1.78 | 5.47 | 5.07 | 0.18 | 3.36 | 2.74 |
| 30r1 | 30 | 84 | 0.17 | 2.11 | 3.38 | 5.6 | 0.10 | 5.58 | 3.19 |
| 30r2 | 30 | 76 | 0.21 | 2.23 | 4.00 | 5.07 | 0.09 | 14.04 | 2.84 |
| 30r3 | 30 | 80 | 0.20 | 2.20 | 4.51 | 5.33 | 0.09 | 5.71 | 3.26 |
| 50r1 | 50 | 132 | 0.13 | 2.49 | 4.03 | 5.28 | 0.05 | 7.60 | 3.12 |
| 50r2 | 50 | 136 | 0.13 | 2.60 | 3.61 | 5.44 | 0.06 | 4.61 | 3.65 |
| 50r3 | 50 | 133 | 0.17 | 2.37 | 4.14 | 5.32 | 0.05 | 9.13 | 3.09 |
| 100r1 | 100 | 273 | 0.17 | 2.38 | 3.67 | 5.46 | 0.03 | 3.88 | 3.31 |
| 100r2 | 100 | 273 | 0.19 | 2.45 | 3.37 | 5.46 | 0.03 | 3.20 | 3.96 |
| 100r3 | 100 | 267 | 0.13 | 2.60 | 3.67 | 5.34 | 0.03 | 3.13 | 3.00 |
| 150r1 | 150 | 421 | 0.13 | 2.54 | 3.83 | 5.61 | 0.02 | 3.36 | 3.15 |
| 150r2 | 150 | 394 | 0.09 | 3.74 | 3.54 | 5.25 | 0.18 | 4.65 | 3.15 |
| 150r3 | 150 | 407 | 0.12 | 2.57 | 3.45 | 5.43 | 0.018 | 4.29 | 2.92 |
| 200r1 | 200 | 538 | 0.19 | 2.54 | 3.31 | 5.38 | 0.01 | 6.68 | 2.66 |
| 200r2 | 200 | 561 | 0.16 | 2.57 | 3.58 | 5.61 | 0.014 | 5.50 | 3.26 |
| 200r3 | 200 | 548 | 0.088 | 2.80 | 3.52 | 5.48 | 0.014 | 7.78 | 2.79 |
| 250r1 | 250 | 698 | 0.20 | 2.52 | 2.87 | 5.58 | 0.011 | 6.22 | 3.22 |
| 250r2 | 250 | 687 | 0.16 | 2.59 | 3.57 | 5.50 | 0.011 | 5.19 | 3.17 |
| 250r3 | 250 | 693 | 0.18 | 2.54 | 3.56 | 5.54 | 0.011 | 4.30 | 3.39 |

Table 7: Properties of underlying interaction graphs of intercellular networks.

| Name | Properties | | | | | | | | |
|---|---|---|---|---|---|---|---|---|---|
| | # Nodes | # Edges | $C$ | $d$ | $\gamma$ | $\langle k \rangle$ | $\delta$ | $\gamma^{in}$ | $\gamma^{out}$ |
| 15r1 | 15 | 24 | 0.066 | 2.57 | 15.32 | 3.2 | 0.11 | 13.33 | 6.05 |
| 15r2 | 15 | 27 | 0.15 | 2.12 | 11.73 | 3.6 | 0.13 | 4.45 | 2.35 |
| 15r3 | 15 | 25 | 0.13 | 2.21 | 5.27 | 3.33 | 0.12 | 6.32 | 8.40 |
| 30r1 | 30 | 42 | 0.0 | 3.22 | 12.10 | 2.8 | 0.048 | 11.92 | 8.53 |
| 30r2 | 30 | 52 | 0.050 | 2.74 | 8.79 | 3.47 | 0.060 | 14.90 | 8.37 |
| 30r3 | 30 | 106 | 0.12 | 2.59 | 19.01 | 7.07 | 12.18 | 5.23 | 5.59 |
| 50r1 | 50 | 122 | 0.041 | 2.59 | 27.89 | 4.88 | 0.050 | 10.54 | 6.20 |
| 50r2 | 50 | 136 | 0.045 | 2.50 | 11.00 | 5.44 | 0.056 | 39.39 | 8.59 |
| 50r3 | 50 | 110 | 0.062 | 2.73 | 22.02 | 4.4 | 0.045 | 13.92 | 41.33 |
| 100r1 | 100 | 511 | 0.057 | 2.99 | 9.97 | 10.22 | 0.052 | 25.92 | 7.46 |
| 100r2 | 100 | 500 | 0.049 | 2.25 | 8.89 | 10.0 | 0.051 | 9.92 | 13.84 |
| 100r3 | 100 | 464 | 0.050 | 2.31 | 10.73 | 9.28 | 0.047 | 16.17 | 22.80 |
| 150r1 | 150 | 1,173 | 0.052 | 2.08 | 10.90 | 15.64 | 0.052 | 49.92 | 7.30 |
| 150r2 | 150 | 1,103 | 0.049 | 2.72 | 7.65 | 14.71 | 0.049 | 6.82 | 9.19 |
| 150r3 | 150 | 1,117 | 0.048 | 2.69 | 12.75 | 14.89 | 0.050 | 20.21 | 8.80 |
| 200r1 | 200 | 2,023 | 0.052 | 2.54 | 18.64 | 20.23 | 0.051 | 7.19 | 7.68 |
| 200r2 | 200 | 1,924 | 0.048 | 2.58 | 23.10 | 19.24 | 0.048 | 14.98 | 10.42 |
| 200r3 | 200 | 2,004 | 0.052 | 2.55 | 21.82 | 20.04 | 0.050 | 10.32 | 5.11 |
| 250r1 | 250 | 3,121 | 0.049 | 2.46 | 9.03 | 24.97 | 0.050 | 10.09 | 13.40 |
| 250r2 | 250 | 3,144 | 0.050 | 2.46 | 18.56 | 25.15 | 0.051 | 10.79 | 8.13 |
| 250r3 | 250 | 3,101 | 0.050 | 2.47 | 8.19 | 24.81 | 0.050 | 39.61 | 7.07 |

Table 8: Properties of underlying interaction graphs of landscape networks.

| Name | Properties | | | | | |
|---|---|---|---|---|---|---|
| | # Nodes | # Edges | $C$ | $d$ | $\gamma$ | $\langle k \rangle$ | $\delta$ |
| 15r1 | 15 | 46 | 0.72 | 1.71 | 4.72 | 6.0 | 0.44 |
| 15r2 | 15 | 61 | 0.85 | 1.42 | 5.60 | 8.13 | 0.58 |
| 15r3 | 15 | 52 | 0.78 | 1.54 | 5.28 | 6.93 | 0.50 |
| 30r1 | 30 | 103 | 0.69 | 2.27 | 4.95 | 6.8 | 0.24 |
| 30r2 | 30 | 144 | 0.72 | 1.83 | 28.31 | 9.6 | 0.33 |
| 30r3 | 30 | 136 | 0.76 | 1.79 | 4.57 | 9.07 | 0.31 |
| 50r1 | 50 | 254 | 0.71 | 2.12 | 4.01 | 10.16 | 0.21 |
| 50r2 | 50 | 251 | 0.71 | 2.29 | 5.10 | 10.04 | 0.20 |
| 50r3 | 50 | 222 | 0.74 | 2.21 | 5.96 | 8.88 | 0.18 |
| 100r1 | 100 | 542 | 0.70 | 3.03 | 4.52 | 10.82 | 0.11 |
| 100r2 | 100 | 453 | 0.68 | 3.39 | 4.03 | 9.06 | 0.092 |
| 100r3 | 100 | 423 | 0.72 | 3.74 | 8.02 | 8.46 | 0.085 |
| 150r1 | 150 | 784 | 0.67 | 3.71 | 5.23 | 10.43 | 0.070 |
| 150r2 | 150 | 824 | 0.69 | 3.53 | 5.06 | 10.99 | 0.074 |
| 150r3 | 150 | 806 | 0.67 | 3.41 | 5.80 | 10.75 | 0.072 |
| 200r1 | 200 | 1,162 | 0.71 | 4.04 | 6.04 | 11.61 | 0.058 |
| 200r2 | 200 | 1,025 | 0.68 | 4.22 | 4.17 | 10.24 | 0.052 |
| 200r3 | 200 | 1,019 | 0.69 | 4.33 | 5.23 | 10.19 | 0.051 |
| 250r1 | 250 | 1,492 | 0.67 | 4.03 | 3.74 | 11.92 | 0.048 |
| 250r2 | 250 | 1,217 | 0.66 | 4.69 | 5.62 | 9.73 | 0.039 |
| 250r3 | 250 | 1,409 | 0.67 | 4.59 | 4.80 | 11.26 | 0.045 |

Table 9: Properties of underlying interaction graphs of man-made organic reaction networks.

| Name | Properties | | | | | | | | |
|------|---------|---------|---------|------|-------|---------------|--------|--------------|---------------|
|      | # Nodes | # Edges | $C$ | $d$ | $\gamma$ | $\langle k \rangle$ | $\delta$ | $\gamma^{in}$ | $\gamma^{out}$ |
| 15r1 | 15 | 15 | 0.067 | 1.86 | 4.73 | 2.0 | 0.071 | - | 2.17 |
| 15r2 | 15 | 15 | 0.045 | 1.96 | 4.44 | 2.0 | 0.071 | - | 1.94 |
| 15r3 | 15 | 15 | 0.044 | 1.96 | 4.44 | 2.0 | 0.070 | - | 1.94 |
| 30r1 | 30 | 30 | 0.002 | 2.43 | 5.36 | 2.0 | 0.034 | - | 2.27 |
| 30r2 | 30 | 30 | 0.0028 | 2.28 | 5.62 | 2.0 | 0.034 | - | 2.59 |
| 30r3 | 30 | 30 | 0.017 | 2.22 | 6.08 | 2.0 | 0.034 | - | 1.60 |
| 50r1 | 50 | 50 | 0.0035 | 2.29 | 7.92 | 2.0 | 0.020 | - | 1.65 |
| 50r2 | 50 | 50 | 0.00037 | 2.50 | 6.99 | 2.0 | 0.020 | - | 2.58 |
| 50r3 | 50 | 50 | 0.0018 | 2.36 | 7.60 | 2.0 | 0.020 | - | 1.77 |
| 100r1 | 100 | 100 | 9.37 | 2.40 | 11.26 | 2.0 | 0.010 | - | 2.57 |
| 100r2 | 100 | 100 | 7.29 | 2.48 | 10.96 | 2.0 | 0.010 | - | 2.35 |
| 100r3 | 100 | 100 | 0.00026 | 2.36 | 11.71 | 2.0 | 0.010 | - | 1.80 |
| 150r1 | 150 | 150 | 5.36 | 2.63 | 12.68 | 2.0 | 0.0067 | - | 1.74 |
| 150r2 | 150 | 150 | 2.81 | 2.71 | 11.47 | 2.0 | 0.0067 | - | 2.14 |
| 150r3 | 150 | 150 | 0.00016 | 2.58 | 11.52 | 2.0 | 0.0067 | - | 1.58 |
| 200r1 | 200 | 200 | 3.59e-5 | 2.46 | 15.75 | 2.0 | 0.0050 | - | 1.97 |
| 200r2 | 200 | 200 | 0.0025 | 2.30 | 14.31 | 2.0 | 0.0050 | - | 1.66 |
| 200r3 | 200 | 200 | 2.44e-5 | 2.43 | 13.67 | 2.0 | 0.0050 | - | 1.63 |
| 250r1 | 250 | 250 | 7.24e-5 | 3.00 | 9.33 | 2.0 | 0.0040 | - | 1.75 |
| 250r2 | 250 | 250 | 8.52e-6 | 2.58 | 12.95 | 2.0 | 0.0040 | - | 1.88 |
| 250r3 | 250 | 250 | 2.95e-6 | 2.89 | 10.30 | 2.0 | 0.0040 | - | 1.79 |

Table 10: Properties of underlying interaction graphs of reaction networks inside living organisms.

| Name | Properties | | | | | | | | |
|------|---------|---------|---------|------|-------|---------------|--------|--------------|---------------|
|      | # Nodes | # Edges | $C$ | $d$ | $\gamma$ | $\langle k \rangle$ | $\delta$ | $\gamma^{in}$ | $\gamma^{out}$ |
| 15r1 | 15 | 16 | 0.049 | 2.21 | 3.47 | 2.13 | 0.076 | 22.64 | 3.01 |
| 15r2 | 15 | 15 | 0.0071 | 2.30 | 3.81 | 2.0 | 0.071 | - | 5.77 |
| 15r3 | 15 | 15 | 0.037 | 2.11 | 4.19 | 2.0 | 0.071 | - | 1.81 |
| 30r1 | 30 | 30 | 0.0073 | 2.15 | 6.06 | 2.0 | 0.034 | - | 1.94 |
| 30r2 | 30 | 30 | 0.0032 | 2.36 | 5.52 | 2.0 | 0.034 | - | 2.04 |
| 30r3 | 30 | 30 | 0.0021 | 2.39 | 5.41 | 2.0 | 0.034 | - | 2.33 |
| 50r1 | 50 | 50 | 0.00058 | 2.45 | 7.16 | 2.0 | 0.020 | 71.69 | 2.22 |
| 50r2 | 50 | 50 | 0.010 | 2.28 | 8.25 | 2.0 | 0.020 | - | 2.78 |
| 50r3 | 50 | 50 | 0.00056 | 2.57 | 6.00 | 2.0 | 0.020 | 35.62 | 2.66 |
| 100r1 | 100 | 102 | 0.0117 | 2.53 | 8.49 | 2.04 | 0.010 | 36.35 | 3.47 |
| 100r2 | 100 | 100 | 0.00013 | 2.35 | 11.50 | 2.0 | 0.010 | - | 2.39 |
| 100r3 | 100 | 100 | 0.00012 | 2.51 | 10.47 | 2.0 | 0.010 | 143.83 | 1.41 |
| 150r1 | 150 | 150 | 3.57e-5 | 2.56 | 12.44 | 2.0 | 0.0067 | 107.76 | 1.43 |
| 150r2 | 150 | 151 | 0.0039 | 2.51 | 13.42 | 2.01 | 0.0068 | 108.48 | 1.53 |
| 150r3 | 150 | 152 | 0.0067 | 2.75 | 9.22 | 2.03 | 0.0068 | 30.88 | 5.21 |
| 200r1 | 200 | 201 | 0.0025 | 2.75 | 12.70 | 2.01 | 0.0051 | 72.05 | 2.92 |
| 200r2 | 200 | 202 | 1.57e-5 | 2.99 | 13.04 | 2.02 | 0.0051 | 96.70 | 2.57 |
| 200r3 | 200 | 202 | 0.0050 | 2.45 | 12.32 | 2.02 | 0.0051 | 41.19 | 1.63 |
| 250r1 | 250 | 254 | 0.0080 | 2.63 | 15.92 | 2.03 | 0.0041 | 72.85 | 19.98 |
| 250r2 | 250 | 251 | 0.0020 | 2.58 | 17.02 | 2.01 | 0.0040 | 120.26 | 2.42 |
| 250r3 | 250 | 250 | 9.37e-6 | 2.55 | 18.98 | 2.0 | 0.0040 | 360.23 | 1.39 |

Table 11: Properties of underlying interaction graphs of social networks.

| Name | # Nodes | # Edges | $C$ | $d$ | $\gamma$ | $\langle k \rangle$ | $\delta$ | $\gamma^{in}$ | $\gamma^{out}$ |
|---|---|---|---|---|---|---|---|---|---|
| 15r1 | 15 | 28 | 0.14 | 3.14 | 14.44 | 3.73 | 0.13 | 14.56 | 14.56 |
| 15r2 | 15 | 28 | 0.15 | 3.49 | 11.08 | 3.73 | 0.13 | 10.10 | 2.98 |
| 15r3 | 15 | 27 | 0.18 | 4.17 | 6.89 | 3.6 | 0.13 | 3.52 | 3.09 |
| 30r1 | 30 | 107 | 0.13 | 2.68 | 7.52 | 7.13 | 0.12 | 4.30 | 3.97 |
| 30r2 | 30 | 106 | 0.14 | 2.68 | 14.16 | 7.07 | 0.12 | 16.69 | 18.35 |
| 30r3 | 30 | 116 | 0.14 | 2.50 | 6.44 | 7.73 | 0.13 | 70.79 | 33.44 |
| 50r1 | 50 | 249 | 0.13 | 2.62 | 1.89 | 9.96 | 0.10 | 5.53 | 4.15 |
| 50r2 | 50 | 269 | 0.14 | 2.49 | 4.92 | 10.76 | 0.11 | 5.17 | 9.11 |
| 50r3 | 50 | 294 | 0.13 | 2.36 | 7.58 | 11.76 | 0.12 | 11.28 | 12.44 |
| 100r1 | 100 | 1,260 | 0.13 | 2.05 | 9.86 | 25.2 | 0.13 | 7.73 | 117.97 |
| 100r2 | 100 | 1,273 | 0.13 | 2.05 | 17.86 | 25.46 | 0.13 | 14.01 | 7.19 |
| 100r3 | 100 | 1,236 | 0.13 | 2.06 | 12.84 | 24.72 | 0.12 | 8.94 | 10.31 |
| 150r1 | 150 | 2,133 | 0.09 | 2.14 | 13.05 | 28.44 | 0.095 | 8.31 | 22.29 |
| 150r2 | 150 | 2,136 | 0.097 | 2.13 | 12.03 | 28.48 | 0.096 | 25.24 | 15.80 |
| 150r3 | 150 | 2,148 | 0.094 | 2.13 | 12.35 | 28.64 | 0.096 | 24.48 | 16.70 |
| 200r1 | 200 | 3,788 | 0.096 | 2.06 | 20.18 | 37.88 | 0.096 | 6.71 | 10.94 |
| 200r2 | 200 | 3,803 | 0.094 | 2.05 | 10.99 | 38.03 | 0.096 | 20.36 | 18.15 |
| 200r3 | 200 | 3,793 | 0.094 | 2.06 | 17.29 | 37.93 | 0.095 | 11.07 | 23.29 |
| 250r1 | 250 | 5,921 | 0.097 | 2.00 | 24.09 | 47.37 | 0.095 | 17.50 | 16.24 |
| 250r2 | 250 | 5,937 | 0.095 | 2.00 | 23.11 | 47.50 | 0.095 | 28.61 | 13.97 |
| 250r3 | 250 | 5,942 | 0.096 | 2.00 | 17.19 | 47.54 | 0.095 | 11.40 | 12.24 |

Table 12: Properties of underlying interaction graphs of vascular networks.

| Name | # Nodes | # Edges | $C$ | $d$ | $\gamma$ | $\langle k \rangle$ | $\delta$ | $\gamma^{in}$ | $\gamma^{out}$ |
|---|---|---|---|---|---|---|---|---|---|
| 15r1 | 15 | 14 | 0.0 | 3.2 | 3.43 | 1.87 | 0.067 | - | 3.34 |
| 15r2 | 15 | 14 | 0.0 | 3.28 | 3.28 | 1.87 | 0.067 | - | 4.64 |
| 15r3 | 15 | 14 | 0.0 | 3.56 | 4.47 | 1.87 | 0.067 | - | 3.52 |
| 30r1 | 30 | 29 | 0.0 | 6.58 | 5.08 | 1.93 | 0.033 | - | 3.75 |
| 30r2 | 30 | 29 | 0.0 | 7.37 | 6.39 | 1.93 | 0.033 | - | 4.61 |
| 30r3 | 30 | 29 | 0.0 | 5.40 | 3.83 | 1.93 | 0.033 | - | 2.99 |
| 50r1 | 50 | 49 | 0.0 | 11.09 | 5.94 | 1.96 | 0.02 | - | 4.31 |
| 50r2 | 50 | 49 | 0.0 | 7.90 | 3.15 | 1.96 | 0.02 | - | 3.82 |
| 50r3 | 50 | 49 | 0.0 | 11.04 | 5.76 | 1.96 | 0.02 | - | 4.15 |
| 100r1 | 100 | 99 | 0.0 | 18.32 | 4.65 | 1.98 | 0.01 | - | 3.77 |
| 100r2 | 100 | 99 | 0.0 | 15.84 | 5.02 | 1.98 | 0.01 | - | 4.28 |
| 100r3 | 100 | 99 | 0.0 | 17.33 | 4.73 | 1.98 | 0.01 | - | 4.25 |
| 150r1 | 150 | 149 | 0.0 | 25.47 | 5.45 | 1.99 | 0.0067 | - | 4.45 |
| 150r2 | 150 | 149 | 0.0 | 24.43 | 4.45 | 1.99 | 0.0067 | - | 3.19 |
| 150r3 | 150 | 149 | 0.0 | 25.22 | 4.46 | 1.99 | 0.0067 | - | 3.97 |
| 200r1 | 200 | 199 | 0.0 | 29.01 | 4.66 | 1.99 | 0.005 | - | 4.21 |
| 200r2 | 200 | 199 | 0.0 | 29.51 | 4.18 | 1.99 | 0.005 | - | 3.44 |
| 200r3 | 200 | 199 | 0.0 | 36.12 | 4.92 | 1.99 | 0.005 | - | 4.23 |
| 250r1 | 250 | 249 | 0.0 | 40.51 | 5.11 | 1.992 | 0.004 | - | 4.24 |
| 250r2 | 250 | 249 | 0.0 | 40.60 | 4.56 | 1.992 | 0.004 | - | 3.37 |
| 250r3 | 250 | 249 | 0.0 | 41.30 | 3.91 | 1.992 | 0.004 | - | 3.41 |

Table 13: The scripts for the graph generation.

| Graph | Script |
|-------|--------|
| BN | generate_brain_networks_hierarchical.py |
| CRNA | generate_chemical_reactions_in_atmosphere.py |
| FW | generate_food_webs.py |
| GCN | generate_gene_coexpression_networks.py |
| GRN | /network_generation_algo/src/test.py |
| IN | generate_intercellular_networks.py |
| LN | generate_landscape_networks.py |
| MMO | generate_man_made_organic_reaction_networks.py |
| RNLO | generate_reaction_networks_inside_living_organism.py |
| SN | generate_social_networks_latest.py |
| VN | generate_vascular_networks.py |

## B.2 Dynamical system simulations

The corresponding code for the simulations of interacting dynamical systems can be found at `https://github.com/wang422003/Benchmarking-Structural-Inference-Methods-for-Interacting-Dynamical-Systems/tree/main/src/simulations`. The corresponding scripts for every simulation are summarized in Table 13.

Table 14: The scripts for the simulation of interacting dynamical systems.

| Simulation | Script |
|------------|--------|
| Springs & NetSims | generate_trajectories.py |
| Springs & NetSims w. Noise | generate_noisy_trajectories.py |

The details on the simulations of "Springs" and "NetSims" are presented in the following paragraphs.

**Springs simulation.** We simulate the motion of spring-connected particles in a 2D box using the springs simulation, where the nodes are represented as particles, and the edges correspond to springs following Hooke's law for force calculations. Inspired by [58], we simulate $N$ particles (point masses) within a 2D box in the absence of external forces. Elastic collisions with the box are accounted for. The interaction graphs obtained from the previous section are employed to determine the spring connections. The particles are interconnected through springs with forces governed by Hooke's law, given by $F_{ij}(t) = -k(x_i(t) - x_j(t))$, where $F_{ij}(t)$ represents the force exerted on particle $i$ by particle $j$ at time $t$, $k$ is the spring constant, and $x_i(t)$ is the 2D location vector of particle $i$ at time $t$. The dynamic function of the Springs simulation is characterized by a second-order ODE which can be represented as follows:

$$m_i \cdot x_i''(t) = \sum_{j \in \mathcal{N}_i} -k \cdot \big(x_i(t) - x_j(t)\big), \tag{4}$$

Here, $m_i$ represents the mass of node $i$, assumed to be 1 for simplicity. The spring constant, denoted as $k$, is fixed at 1. $\mathcal{N}_i$ refers to the set of neighboring nodes with directed connections to node $i$. We integrate this equation to compute $x_i'(t)$ and subsequently $x_i(t)$ for each time step. The sampled values of $x_i'(t)$ and $x_i(t)$ form the 4D node features at each time step. The initial locations are sampled from a Gaussian distribution $\mathcal{N}(0, 0.5)$, and the initial velocities, also 2D vectors, are randomly generated with a norm of 0.5. Starting from these initial locations and velocities in two dimensions, we simulate the trajectories by solving Newton's equations of motion. The simulation is performed using leapfrog integration with a minor time step size of 0.001 seconds, and the trajectories are sampled every 100 minor time steps. Consequently, the feature representation of each node at every minor time step in this case is a 4D vector comprising 2D locations and 2D velocities.

We implement the simulation in such a way that the next value of a feature of each particle depends on the current value of the feature and the interactions with other particles. This design allows us to accommodate theoretically asymmetric interaction graphs, as the spring force is disentangled for each individual particle. Given a set of initial locations and velocities, we generate trajectories for the current interacting dynamical system, encompassing all feature vectors of the particles within the specified time period. Specifically, we generate trajectories comprising 49 time points (obtained with integration over 4,900 minor time steps) for training and validation purposes, while trajectories with 100 time steps are generated for testing to align with the requirements in [58, 106]. For each interaction graph, we generate a total of 8,000 trajectories for training, 2,000 trajectories for validation, and 2,000 trajectories for testing.

**NetSims simulation.** The NetSim dataset, described in [95], simulates blood-oxygen-level-dependent (BOLD) imaging data across different regions within the human brain. It has been extensively utilized in structural inference experiments as documented in [68, 106]. In [68], NetSims were initially adopted as the dataset for structural inference experiments. In this simulation, each node corresponds to a spatial region of interest derived from brain atlases or functional tasks. The node feature represents the 1D neural signal at each time step. To enhance the diversity and complexity of the data, we generate additional NetSims following the procedure outlined in [95]. The dynamics of the NetSims are modeled using dynamic causal modeling [35], and follow a first-order ODE model for the 1D BOLD signal of each node $i$ at time step $t$:

$$x_i'(t) = \sigma \cdot \sum_{j \in \mathcal{N}_i} x_j(t) - \sigma \cdot x_i(t) + C \cdot u_i, \tag{5}$$

where $\sigma$ governs the within-node temporal smoothing and neural lag between nodes, and is set to $0.1$ based on [95]. $C$ represents weights controlling the interaction of external inputs with the network and is set to zero here to minimize noise from external inputs $u_i$ [95]. The off-diagonal terms in **A** determine the interactions between nodes, while the diagonal elements are set to $-1$ to model within-node temporal decay. The 1D node features at each time step are formed using the sampled $x_i(t)$.

The initial features are sampled from a Gaussian distribution $\mathcal{N}(0, 0.5)$. For each initial feature, we generate a trajectory. The trajectory collection settings used in this study are consistent with those employed in the "Springs" simulation.

## B.3    Quality evaluation of DoSI

In order to assess the quality of the proposed DoSI dataset, we follow the metrics mentioned in [39], and with adaption to the data proposed in DoSI, as the original were designed for images data or text data. We have the following metrics: (1) file completeness, (2) trajectory completeness, (3) adjacency matrix completeness, and (4) label accuracy.

**File Completeness.** This metric aims at finding possible missing files in the dataset The measurement of this metric is originally calculated by:

$$X_{FC} = \frac{\sum_{i=1}^{N1} a_i + \sum_{i=1}^{N2} b_i}{2 \times \max(N1, N2)}, \tag{6}$$

where $N1$ is the number of trajectory files, $N2$ is the number of adjacency matrix files. Besides, we traverse the data folder, and for the $i$th data record, check whether the annotation file corresponding to the data record exists in the annotation folder, if it exists, $a_i = 0$, otherwise, $a_i = 1$. We traverse the annotation folder, and for the $i$th annotation file, check whether the data record file corresponding to the annotation file exists in the data folder. If it exists, $b_i = 0$, otherwise, $b_i = 1$. As we put the trajectory file (data record) and adjacency matrix file (annotation) of each subdataset in the same folder, we simplify the measurement metric in Eq. 6 to:

$$X_{FC} = \frac{\sum_{i=1}^{N} a_i}{2 \times N}, \tag{7}$$

where $N$ is the number of trajectory files. $a_i$: Traverse the folder, and for the $i$th trajectory, check whether the adjacency matrix file corresponding to the trajectory exists in the annotation folder. If it exists, $a_i = 0$, otherwise, $a_i = 1$.

**Trajectory Completeness.** We check the missing values in each trajectory in the proposed dataset, and report the metrics as:

$$X_{TC} = \frac{\sum_{t=1}^{T} c_t}{T}, \tag{8}$$

where $T$ refers to the expected length of the trajectory, $c_i$: Traverse from the time $t = 1$ for each trajectory till $t = T$, and for the $t$th time step, check whether the feature exists or not. If it exists, $c_t = 0$, otherwise, $c_t = 1$.

**Adjacency Matrix Completeness.** We check the missing values in each adjacency matrix in the proposed dataset, and report the metrics as:

$$X_{AC} = \frac{\sum_{i=1}^{N} \sum_{j=1}^{N} d_{ij}}{N^2}, \tag{9}$$

where $N$ refers to the number of the nodes, $d_{ij}$: check whether the value representing the connectivity from node $i$ to $j$ exists or not in the adjacency matrix. If it exists, $d_{ij} = 0$, otherwise, $d_{ij} = 1$.

**Label Accuracy.** We check the obtained adjacency matrix in the data folder and the corresponding adjacency matrix fed to the dynamics simulation, and report the value as: $X_{LC}$, it equals $0$ if there is no mismatch, or $1$, otherwise.

We report all results of all datasets in Tables 15-17. As shown in the tables, all of the proposed datasets are free from errors on the four metrics.

Table 15: Quality evaluation of the BN, CRNA, FW and GCN datasets proposed in DoSI.

Table 16: Quality evaluation of the GRN, IN, LN and MMO datasets proposed in DoSI.

Table 17: Quality evaluation of the RNLO, SN and VN datasets proposed in DoSI.

**B.4 More details on EMT dataset**

The EMT dataset contains observations of the TGFB1-induced epithelial-mesenchymal transition from the A549 cancer cell line. This dataset was collected by Cook and Vanderhyden [26] and was processed by Sha et al. [92]. The raw sequencing files [26] can be retrieved from `https://www.ncbi.nlm.nih.gov/geo/query/acc.cgi?acc=GSE147405` under the license of CC BY 4.0, while the processed gene expression matrixes [92] can be retrieved from `https://github.com/yutongo/TIGON` under the MIT License. This scRNA-seq dataset contains gene expression levels of 3,133 cells and 3,000 genes. The number of cells sampled at day $t = 0, \frac{1}{3}, 1, 3, 7$ are 577, 885, 788, 754, and 129, respectively. We query the interaction network of the top 50 high-variance genes in the EMT dataset on the STRING database [97] and use this interaction network as the ground-truth GRN for evaluation. Isolated components are removed in the ground-truth GRN, resulting in a network with 36 nodes and 103 undirected edges.

We reconstructed the cellular trajectories from the distributions of gene expression across time. We used Waddington-OT [91] to build the transition matrixes between cells in consecutive time steps using optimal transport (OT). For each time step $t_i$, denote the number of genes as $g$, the number of cells and gene expression matrix sampled at $t_i$ as $c_i$ and $X^{t_i} \in \mathbb{R}^{g \times c_i}$, respectively. For each pair of gene expression matrix $X^{t_i}, X^{t_{i+1}}$ at time $i, i+1$, Waddington-OT first computes the pairwise cell Euclidean distance matrix $M \in \mathbb{R}^{c_i \times c_{i+1}}$ as the cost matrix. Waddington-OT then solves the following unbalanced entropy-regularized OT optimization problem:

$$\gamma_{t_i \to t_{i+1}} = \underset{\gamma \in \mathbb{R}^{c_i \times c_{i+1}}}{\arg \min} \quad \sum_{p,q} \gamma_{p,q} M_{p,q} + \epsilon \sum_{p,q} \gamma_{p,q} \log \gamma_{p,q} \tag{10}$$

$$\text{subject to} \quad \sum_{p} \gamma_{p,q} = \frac{1}{c_i}$$

$$\sum_{q} \gamma_{p,q} = \frac{1}{c_{i+1}}$$

$$\gamma_{p,q} \geq 0,$$

where $p, q$ are the auxiliary cell indexes for time $t_i, t_{i+1}$, $\gamma_{t_i \to t_{i+1}}$ is the optimal transport plan between $X^{t_i}$ and $X^{t_{i+1}}$, and $\epsilon$ is the entropic regularization term. We view the optimal transport plan as a transition matrix between cells in $X^{t_i}$ and $X^{t_{i+1}}$, and reconstruct the cell trajectory $V_p$ by iteratively finding the next most probable cell at the next time step through the transition matrix $\gamma_{t_i \to t_{i+1}}$. Finally, we convert the irregular cell trajectories at 5-time points into regular time series with 22-time points using piecewise cubic hermite interpolating polynomial [36], where the time difference between consecutive interpolated time steps is determined by the greatest common divisor of the sample times.

# C Further implementation details of structural inference methods

In this section, we demonstrate the implementation details of the structural inference methods in this work. For every method, we show the implementation, computational resources, and if possible, the choice of hyperparameters.

The TIGRESS method, information-theory-based methods, and tree-based methods assumed an input of normalized 1D gene expression level, so we performed an extra hyperparameter search of the normalization method on top of the original method implementation. Among "NetSims" and "Springs" simulations, only the former gives 1D feature, so all methods are tested only on "NetSims" dataset. For each trajectory, we denote $v_i^t$ as the scalar neural signal for node $i$ at time $t$. The normalization methods included:

- `None`: no normalization,
- `Symlog`: symmetrically shifted logarithm transform with equation $f(v_i^t) = sign(v_i^t)log(1 + |v_i^t|)$,
- `Unitary`: L2 normalization on the node dimension, and
- `Z-score`: standardization using standard deviation on the node dimension.

## C.1 ppcor

**Implementation.** We use the official implementation of ppcor from the R package `ppcor` [57] with a customized wrapper. Our wrapper will parse multiple arguments to select a set of targeted trajectories for inference, transform trajectories into a suitable format, feed each trajectory into the ppcor algorithm, and store the output into designated directories. Our implementation can be found at `https://github.com/wang422003/Benchmarking-Structural-Inference-Methods-for-Interacting-Dynamical-Systems/tree/main/src/models/ppcor`. The method is implemented by `ppcor` [57] in R with the help of `NumPy` [45] Python package to store generated trajectories, `reticulate` from `https://github.com/rstudio/reticulate` to load Python variables into the R environment, `stringr` from `https://stringr.tidyverse.org` for string operation, and `optparse` from `https://github.com/trevorld/r-optparse` to produce Python-style argument parser.

**Computational resources.** We infer networks on Amazon EC2 C7g.2xlarge instances equipped with 64 vCPUs powered by AWS Graviton3 processors and 128 GB RAM. Each inference took one vCPU to run.

**Hyperparameters.** The hyperparameters that are being considered during implementation are (1) the normalization method, (2) the correlation statistics, and (3) the function to compute partial or semi-partial correlation. The corresponding search spaces are:

- the normalization method: `None`, `Symlog`, `Unitary`, `Z-score`,
- the correlation statistics: `pearson`, `spearman`,
- the function to compute partial or semi-partial correlation: `spcor`, `pcor`.

We search for the values of these hyperparameters on the NetSims simulation trajectories of CRNA graph of 15 nodes, and we find the best hyperparameters to be: (1) the normalization method: `None`, (2) the MI estimation method: `spearman`, and (3) the function to compute partial or semi-partial correlation: `pcor`. Due to computational requirements, we do not perform the hyperparameter search on every trajectory but use this set of choices for all of the experiments. We argue that there might be other possible values, but the effect on the structural inference results is minor.

## C.2 TIGRESS

**Implementation.** We use the official implementation of TIGRESS by the author at `https://github.com/jpvert/tigress` with a customized wrapper. Our wrapper will parse multiple arguments to select a set of targeted trajectories for inference, transform trajectories into a suitable format, feed each trajectory into the TIGRESS algorithm, and store the output in designated directories. Our implementation can be found at `https://github.com/wang422003/Benchmarking-Structural-Inference-Methods-for-Interacting-Dynamical-Systems/tree/main/src/models/TIGRESS`. The method is implemented in R with the help of `NumPy` [45] Python package to store generated trajectories, `reticulate` from `https://github.com/rstudio/reticulate` to load Python variables into the R environment, `stringr` from `https://stringr.tidyverse.org` for string operation, and `optparse` from `https://github.com/trevorld/r-optparse` to produce Python-style argument parser.

**Computational resources.** We infer networks on our clusters with 128 AMD Epyc ROME 7H12 @ 2.6 GHz CPUs and 256 GB RAM. Each inference took the whole cluster to run.

**Hyperparameters.** The hyperparameters that are being considered during implementation are (1) the normalization method, (2) the noise level in stability selection, (3) the number of steps in least angle regression (LARS), (4) the number of random subsampling in stability selection, (5) the scoring method in stability selection, and (6) the Boolean to perform node-level standardization. The corresponding search spaces are:

- the normalization method: `None`, `Symlog`, `Unitary`, `Z-score`,
- the noise level in stability selection: 0.1, 0.2, 0.5,
- the number of steps in LARS: 3, 5, 8, 10,
- the number of random subsampling in stability selection: 50, 100, 200, 500,

- the scoring method in stability selection: `area`, `max`,
- the Boolean to perform node-level standardization: `True`, `False`.

We search for the values of these hyperparameters on the NetSims simulation trajectories of CRNA graph of 15 nodes, and we find the best hyperparameters to be: (1) the normalization method: `Symlog`, (2) the noise level in stability selection: 0.5, (3) the number of steps in LARS: 5, (4) the number of random subsampling in stability selection: 500, (5) the scoring method in stability selection: `area`, and (6) the Boolean to perform node-level standardization: `True`. Due to computational requirements, we do not perform the hyperparameter search on every trajectory but use this set of choices for all of the experiments. We argue that there might be other possible values, but the effect on the structural inference results is minor.

## C.3 ARACNe

**Implementation.** We use the implementation of ARACNe by the Bioconductor [49] package `minet` [75] with a customized wrapper. Our wrapper will parse multiple arguments to select a set of targeted trajectories for inference, transform trajectories into a suitable format, feed each trajectory into the ARACNe algorithm, and store the output into designated directories. Our implementation can be found at `https://github.com/wang422003/Benchmarking-Structural-Inference-Methods-for-Interacting-Dynamical-Systems/tree/main/src/models/ARACNE`. The method is implemented by `minet` [75] in R with the help of NumPy [45] Python package to store generated trajectories, `reticulate` from `https://github.com/rstudio/reticulate` to load Python variables into the R environment, `stringr` from `https://stringr.tidyverse.org` for string operation, and `optparse` from `https://github.com/trevorld/r-optparse` to produce Python-style argument parser.

**Computational resources.** We infer networks on Amazon EC2 C7g.2xlarge instances equipped with 64 vCPUs powered by AWS Graviton3 processors and 128 GB RAM. Each inference took one vCPU to run.

**Hyperparameters.** The hyperparameters that are being considered during implementation are (1) the normalization method, (2) the MI estimation method, (3) the discretization method, and (4) the MI threshold for edge removal. The corresponding search spaces are:

- the normalization method: `None`, `Symlog`, `Unitary`, `Z-score`,
- the MI estimation method: `mi.empirical`, `mi.mm`, `mi.shrink`, `mi.sg`, `pearson`, `spearman`,
- the discretization method: `equalfreq`, `equalwidth`, `globalequalwidth`,
- the MI threshold for edge removal: 0, 0.01, 0.02, 0.05, 0.1, 0.2, 0.5, 1, 2, 5, 10.

We search for the values of these hyperparameters on the NetSims simulation trajectories of CRNA graph of 15 nodes, and we find the best hyperparameters to be: (1) the normalization method: `Symlog`, (2) the MI estimation method: `spearman`, (3) the discretization method: `equalfreq`, and (4) the MI threshold for edge removal: 0.1. Due to computational requirements, we do not perform the hyperparameter search on every trajectory but use this set of choices for all of the experiments. We argue that there might be other possible values, but the effect on the structural inference results is minor.

## C.4 CLR

**Implementation.** We use the implementation of CLR by the Bioconductor [49] package `minet` [75] with a customized wrapper. Our wrapper will parse multiple arguments to select a set of targeted trajectories for inference, transform trajectories into a suitable format, feed each trajectory into the CLR algorithm, and store the output into designated directories. Our implementation can be found at `https://github.com/wang422003/Benchmarking-Structural-Inference-Methods-for-Interacting-Dynamical-Systems/tree/main/src/models/CLR`. The method is implemented by `minet` [75] in R with the help of NumPy [45] Python package to store generated trajectories, `reticulate` from `https://github.com/rstudio/reticulate` to load Python variables into the R environment,

`stringr` from `https://stringr.tidyverse.org` for string operation, and `optparse` from `https://github.com/trevorld/r-optparse` to produce Python-style argument parser.

**Computational resources.** We infer networks on Amazon EC2 C7g.2xlarge instances equipped with 64 vCPUs powered by AWS Graviton3 processors and 128 GB RAM. Each inference took one vCPU to run.

**Hyperparameters.** The hyperparameters that are being considered during implementation are (1) the normalization method, (2) the MI estimation method, (3) the discretization method, and (4) the Boolean to skip the diagonal entries. The corresponding search spaces are:

- the normalization method: `None`, `Symlog`, `Unitary`, `Z-score`,
- the MI estimation method: `mi.empirical`, `mi.mm`, `mi.shrink`, `mi.sg`, `pearson`, `spearman`,
- the discretization method: `equalfreq`, `equalwidth`, `globalequalwidth`,
- the Boolean to skip the diagonal entries: `True`, `False`.

We search for the values of these hyperparameters on the NetSims simulation trajectories of CRNA graph of 15 nodes, and we find the best hyperparameters to be: (1) the normalization method: `Symlog`, (2) the MI estimation method: `spearman`, (3) the discretization method: `equalfreq`, and (4) the Boolean to skip the diagonal entries: `False`. Due to computational requirements, we do not perform the hyperparameter search on every trajectory but use this set of choices for all of the experiments. We argue that there might be other possible values, but the effect on the structural inference results is minor.

## C.5   PIDC

**Implementation.**   We use the official implementation of PIDC by the author at `https://github.com/Tchanders/NetworkInference.jl` with a customized wrapper. Our wrapper will parse multiple arguments to select a set of targeted trajectories for inference, transform trajectories into a suitable format, feed each trajectory into the PIDC algorithm, and store the output into designated directories. Our implementation can be found at `https://github.com/wang422003/Benchmarking-Structural-Inference-Methods-for-Interacting-Dynamical-Systems/tree/main/src/models/PIDC`.   The method is implemented in Julia [12] with the help of NumPy [45] Python package to store generated trajectories, `ArgParse.jl` from `https://github.com/carlobaldassi/ArgParse.jl` to parse command line arguments, `CSV.jl` from `https://github.com/JuliaData/CSV.jl` to save and load `.csv` files, `DataFrames.jl` from `https://github.com/JuliaData/DataFrames.jl` to manipulate data array, and `NPZ.jl` from `https://github.com/fhs/NPZ.jl` to load `.npy` into the Julia environment.

**Computational resources.** We infer networks on our clusters with 128 AMD Epyc ROME 7H12 @ 2.6 GHz CPUs and 256 GB RAM. Each inference took one CPU to run.

**Hyperparameters.** The hyperparameters that are being considered during implementation are (1) the normalization method, (2) the discretizing method, (3) the probability distribution estimator, and (4) the number of bins in discretization. The corresponding search spaces are:

- the normalization method: `None`, `Symlog`, `Unitary`, `Z-score`,
- the discretizing method: `uniform_width`, `uniform_count`,
- the probability distribution estimator: `maximum_likelihood`, `miller_madow`, `dirichlet`, `shrinkage`,
- the number of bins in discretization: 4, 5, 10, 20, 100, 200, 500, 1000, $\sqrt{\#\text{Nodes}}$.

We search for the values of these hyperparameters on the NetSims simulation trajectories of CRNA graph of 15 nodes, and we find the best hyperparameters to be: (1) the normalization method: `Symlog`, (2) the discretizing method: `uniform_count`, (3) the probability distribution estimator: `maximum_likelihood`, and (4) the number of bins in discretization: $\sqrt{\#\text{Nodes}}$. Due to computational requirements, we do not perform the hyperparameter search on every trajectory but use this set of choices for all of the experiments. We argue that there might be other possible values, but the effect on the structural inference results is minor.

## C.6 Scribe

**Implementation.** We optimize the official implementation of Scribe by the author at `https://github.com/aristoteleo/Scribe-py` with a customized wrapper. Our wrapper will parse multiple arguments to select a set of targeted trajectories for inference, transform trajectories into a suitable format, feed each trajectory into the Scribe algorithm, and store the output into designated directories. Our implementation has customized `causal_network.py` and `information_estimators.py` scripts so as to modify the hyperparameters directly from command line arguments. We have also optimized the parallel support and computation efficiency and kept minimal functionality for benchmarking purposes, at the same time maintaining its general mechanism. Our implementation can be found at `https://github.com/wang422003/Benchmarking-Structural-Inference-Methods-for-Interacting-Dynamical-Systems/tree/main/src/models/scribe`. The method is implemented in Python with the help of NumPy [45] package to store generated trajectories and `tqdm` from `https://github.com/tqdm/tqdm` to create progress bars.

**Computational resources.** We infer networks on our clusters with 128 AMD Epyc ROME 7H12 @ 2.6 GHz CPUs and 256 GB RAM. Each inference took the whole cluster to run.

**Hyperparameters.** The hyperparameters that are being considered during implementation are (1) the normalization method, (2) the MI estimator, (3) the number of nearest neighbors used in entropy estimation, (4) the number of conditional variables under consideration in MI estimation (only valid when the MI estimator is `crdi` or `ucrdi`), and (5) the Boolean for applying differentiation. The corresponding search spaces are:

- the normalization method: `None`, `Symlog`, `Unitary`, `Z-score`,
- the MI estimator: `rdi`, `urdi`, `crdi`, `ucrdi`,
- the number of nearest neighbors used in entropy estimation: 2, 3, 4, 5,
- the number of conditional variables under consideration in MI estimation: 1, 2, 3, 4, 5,
- the Boolean for applying differentiation: `True`, `False`.

We search for the values of these hyperparameters on the NetSims simulation trajectories of CRNA graph of 15 nodes, and we find the best hyperparameters to be: (1) the normalization method: `Unitary`, (2) the MI estimator: `urdi`, (3) the number of nearest neighbors used in entropy estimation: 2, and (4) the Boolean for applying differentiation: `False`. Due to computational requirements, we do not perform the hyperparameter search on every trajectory but use this set of choices for all of the experiments. We argue that there might be other possible values, but the effect on the structural inference results is minor.

## C.7 dynGENIE3

**Implementation.** We optimize the official Python implementation of dynGENIE3 by the author at `https://github.com/vahuynh/dynGENIE3` with a customized wrapper. Our wrapper will parse multiple arguments to select a set of targeted trajectories for inference, transform trajectories into a suitable format, feed each trajectory into the dynGENIE3 algorithm, and store the output in designated directories. Following the principle of maintaining dynGENIE's general mechanism, we have modified the `dynGENIE3.py` script so as to tune the hyperparameters directly from command line arguments, increase computation efficiency on big datasets, enable calculation of self-influence, and retain minimal functionality for benchmarking purposes. Our implementation can be found at `https://github.com/wang422003/Benchmarking-Structural-Inference-Methods-for-Interacting-Dynamical-Systems/tree/main/src/models/dynGENIE3`. The method is implemented in Python with the help of NumPy [45] package to store generated trajectories.

**Computational resources.** We infer networks on our clusters with 128 AMD Epyc ROME 7H12 @ 2.6 GHz CPUs and 256 GB RAM. Each inference took the whole cluster to run.

**Hyperparameters.** The hyperparameters that are being considered during implementation are (1) the normalization method, (2) the number of trees in random forest regression, and (3) the maximum depth allowed in random forest regression. The corresponding search spaces are:

- the normalization method: `None`, `Symlog`, `Unitary`, `Z-score`,
- the number of trees in random forest regression: 100, 200, 300, 400, 500, 600, 700, 800, 900, 1000,
- the maximum depth allowed in random forest regression: 10, 20, 30, 40, 50, 60, 70, 80, 90, 100, `unlimited`.

We search for the values of these hyperparameters on the NetSims simulation trajectories of CRNA graph of 15 nodes, and we find the best hyperparameters to be: (1) the normalization method: `Z-score`, (2) the number of trees in random forest regression: 700, and (3) the maximum depth allowed in random forest regression: 90. Due to computational requirements, we do not perform the hyperparameter search on every trajectory but use this set of choices for all of the experiments. We argue that there might be other possible values, but the effect on the structural inference results is minor.

## C.8 XGBGRN

**Implementation.** We use the official implementation of XGBGRN by the author at `https://github.com/lab319/GRNs_nonlinear_ODEs` with a customized wrapper. Our wrapper will parse multiple arguments to select a set of targeted trajectories for inference, transform trajectories into a suitable format, feed each trajectory into the XGBGRN algorithm, and store the output in designated directories. Our implementation can be found at `https://github.com/wang422003/Benchmarking-Structural-Inference-Methods-for-Interacting-Dynamical-Systems/tree/main/src/models/GRNs_nonlinear_ODEs`. The method is implemented in Python with the help of `NumPy` [45] package to store generated trajectories.

**Computational resources.** We infer networks on our clusters with 128 AMD Epyc ROME 7H12 @ 2.6 GHz CPUs and 256 GB RAM. Each inference took the whole cluster to run.

**Hyperparameters.** The hyperparameters that are being considered during implementation are (1) the normalization method, (2) the number of estimators, (3) the maximum depth allowed, (4) the subsample ratio during training, (5) the learning rate, and (6) the L1 regularization strength on weights. The corresponding search spaces are:

- the normalization method: `None`, `Symlog`, `Unitary`, `Z-score`,
- the number of estimators: 100, 200, 500, 1000,
- the maximum depth allowed: 3, 5, 6, 8, 10, `unlimited`,
- the subsample ratio during training: 0.6, 0.8, 1.0,
- the learning rate: 0.01, 0.02, 0.05, 0.1,
- the L1 regularization strength on weights: 0, 0.01, 0.02, 0.05.

We search for the values of these hyperparameters on the NetSims simulation trajectories of CRNA graph of 15 nodes, and we find the best hyperparameters to be: (1) the normalization method: `Unitary`, (2) the number of estimators: 100, (3) the maximum depth allowed: 3, (4) the subsample ratio during training: 0.6, (5) the learning rate: 0.1, and (6) the L1 regularization strength on weights: 0.02. Due to computational requirements, we do not perform the hyperparameter search on every trajectory but use this set of choices for all of the experiments. We argue that there might be other possible values, but the effect on the structural inference results is minor.

## C.9 NRI

**Implementation.** We use the official implementation code by the author from `https://github.com/ethanfetaya/NRI` with customized data loaders for our chosen datasets. We choose the `MLPEncoder` and `MLPDecoder` as the blocks for VAE. We add our metric evaluation in the "test" function, after the calculation of accuracy in the original code. Besides that, we add multiple arguments to select the target trajectories for training, but these arguments do not affect the general mechanism of NRI. Our implementation can be found at `https://github.com/wang422003/Benchmarking-Structural-Inference-Methods-for-Interacting-Dynamical-Systems/tree/main/src/models/NRI`. The method is implemented with PyTorch [81] with the help of

Table 18: Batch sizes of the training of different methods in accordance with the number of nodes in the trajectories.

| Methods | Number of Nodes | | | |
|---|---|---|---|---|
| | 15 | 30 | 50 | 100 |
| NRI | 64 | 16 | 16 | 8 |
| ACD | 64 | 16 | 16 | 8 |
| MPM | 32 | 16 | 16 | 8 |
| iSIDG | 64 | 16 | 16 | 8 |
| RCSI | 64 | 16 | 16 | 8 |
| GDP | 8192 | 2048 | 512 | 128 |

Scikit-Learn [82] to calculate metrics. The AUROC values are calculated between the ground truth adjacency matrix and the `prob` variable in the algorithm.

**Computational resources.** We train NRI with two different GPU cards depending on the number of nodes in the trajectories. For the trajectories with less than 50 nodes, we train NRI on a single NVIDIA Tesla V100 SXM2 16G GPU card, with 768 GB RAM, and with a single Xeon Gold 6132 @ 2.6GHz CPU. For the trajectories with equal or more than 50 nodes, we train NRI on a single NVIDIA Tesla V100 SXM2 32G GPU card, with 768 GB RAM, and with a single Xeon Gold 6132 @ 2.6GHz CPU. We show the batch sizes for training NRI in Table 18. The learning rate we use is identical to the default in NRI [58], i.e., 0.0005.

**Hyperparameters.** The hyperparameters that are being considered during implementation are (1) the number of units of the hidden layers in the encoder, (2) the number of units of the hidden layers in the decoder, (3) the dropout rates in the encoder, and (4) the dropout rates in the decoder, while the rest are set the same as the default. These hyperparameters can be set from the arguments of `arg_parser`. The corresponding search spaces are:

- the number of units of the hidden layers in the encoder: $\{128, 256, 512\}$,
- the number of units of the hidden layers in the decoder: $\{128, 256, 512\}$,
- the dropout rates in the encoder: $\{0.0, 0.3, 0.5, 0.6, 0.7, 0.8\}$,
- the dropout rates in the decoder: $\{0.0, 0.3, 0.5, 0.6, 0.7, 0.8\}$.

We search for the values of these hyperparameters based on 5 runs of NRI on the springs simulation trajectories of CRNA graphs of 15 nodes, and we find the best hyperparameters to be: (1) the number of units of the hidden layers in the encoder: 256, (2) the number of units of the hidden layers in the decoder: 256, (3) the dropout rates in the encoder: 0.5, and (4) the dropout rates in the decoder: 0.0. Due to computational requirements, we do not perform the hyperparameter search on every trajectory but use this set of choices for all of the experiments. We argue that there might be other possible values, but the effect on the structural inference results is minor.

## C.10 ACD

**Implementation.** We use the official implementation code by the author (`https://github.com/loeweX/AmortizedCausalDiscovery`) with customized data loaders for our chosen datasets. Same as default, we choose the `MLPEncoder` and `MLPDecoder` as the blocks for ACD. We implement the metric-calculation pipeline in the `forward_pass_and_eval()` function. Besides that, we add multiple arguments to select the target trajectories for training, but these arguments do not affect the general mechanism of ACD. Our implementation can be found at `https://github.com/wang422003/Benchmarking-Structural-Inference-Methods-for-Interacting-Dynamical-Systems/tree/main/src/models/ACD`. The method is implemented with PyTorch [81] with the help of Scikit-Learn [82] to calculate metrics. The AUROC values are calculated between the ground truth adjacency matrix and the `prob` variable in the algorithm.

**Computational resources.** We train ACD with two different GPU cards depending on the number of nodes in the trajectories. For the trajectories with less than 50 nodes, we train ACD on a single

NVIDIA Tesla V100 SXM2 16G GPU card, with 768 GB RAM, and with a single Xeon Gold 6132 @ 2.6GHz CPU. For the trajectories with equal or more than 50 nodes, we train ACD on a single NVIDIA Tesla V100 SXM2 32G GPU card, with 768 GB RAM, and with a single Xeon Gold 6132 @ 2.6GHz CPU. We show the batch sizes for training ACD in Table 18. The learning rate we use is identical to the default in ACD [68], i.e., 0.0005.

**Hyperparameters.** The hyperparameters that are being considered during implementation are (1) the number of units of the hidden layers in the encoder, (2) the number of units of the hidden layers in the decoder, (3) the dropout rates in the encoder, and (4) the dropout rates in the decoder, while the rest are set the same as the default. These hyperparameters can be set from the arguments of `arg_parser`. The corresponding search spaces are:

- the number of units of the hidden layers in the encoder: $\{128, 256, 512\}$,

- the number of units of the hidden layers in the decoder: $\{128, 256, 512\}$,

- the dropout rates in the encoder: $\{0.0, 0.3, 0.5, 0.6, 0.7, 0.8\}$,

- the dropout rates in the decoder: $\{0.0, 0.3, 0.5, 0.6, 0.7, 0.8\}$.

We search for the values of these hyperparameters based on 5 runs of ACD on the springs simulation trajectories of CRNA graphs of 15 nodes, and we find the best hyperparameters to be: (1) the number of units of the hidden layers in the encoder: 256, (2) the number of units of the hidden layers in the decoder: 256, (3) the dropout rates in the encoder: 0.5, and (4) the dropout rates in the decoder: 0.5. Due to computational requirements, we do not perform the hyperparameter search on every trajectory but use this set of choices for all of the experiments. We argue that there might be other possible values, but the effect on the structural inference results is minor.

## C.11 MPM

**Implementation.** We use the official implementation code by the author at `https://github.com/hilbert9221/NRI-MPM` with customized data loaders for our chosen datasets. Same as default, we choose the `RNNENC` and `RNNDEC` as the blocks for MPM. We add our metric evaluation for AUROC in the `evaluate` function of class `XNRIDECIns` in the original code. Besides that, we add multiple arguments to select the target trajectories for training, but these arguments do not affect the general mechanism of MPM. Our implementation can be found at `https://github.com/wang422003/Benchmarking-Structural-Inference-Methods-for-Interacting-Dynamical-Systems/tree/main/src/models/MPM`. The method is implemented with PyTorch [81] with the help of Scikit-Learn [82] to calculate metrics. The AUROC values are calculated between the ground truth adjacency matrix and the `prob` variable in `XNRIIns.test()`.

**Computational resources.** We train MPM with two different GPU cards depending on the number of nodes in the trajectories. For the trajectories with less than 50 nodes, we train MPM on a single NVIDIA Tesla V100 SXM2 16G GPU card, with 768 GB RAM, and with a single Xeon Gold 6132 @ 2.6GHz CPU. For the trajectories with equal or more than 50 nodes, we train MPM on a single NVIDIA Tesla V100 SXM2 32G GPU card, with 768 GB RAM, and with a single Xeon Gold 6132 @ 2.6GHz CPU. We show the batch size for training MPM in Table 18. Because the number of parameters in MPM is larger than those in other deep learning methods, the batch-size of MPM for graphs of 15 nodes is smaller than of other methods. The learning rate we use is identical to the default in MPM [19], i.e., 0.0005.

**Hyperparameters.** The hyperparameters that are being considered during implementation are (1) the number of units of the hidden layers in the encoder, (2) the number of units of the hidden layers in the decoder, (3) the dropout rates in the encoder, and (4) the dropout rates in the decoder, while the rest are set the same as the default. These hyperparameters can be set from the arguments of `config`. The corresponding search spaces are:

- the number of units of the hidden layers in the encoder: $\{128, 256, 512\}$,

- the number of units of the hidden layers in the decoder: $\{128, 256, 512\}$,

- the dropout rates in the encoder: $\{0.0, 0.3, 0.5, 0.6, 0.7, 0.8\}$,

- the dropout rates in the decoder: $\{0.0, 0.3, 0.5, 0.6, 0.7, 0.8\}$.

We search for the values of these hyperparameters based on 5 runs of MPM on the springs simulation trajectories of CRNA graphs of 15 nodes, and we find the best hyperparameters to be: (1) the number of units of the hidden layers in the encoder: 256, (2) the number of units of the hidden layers in the decoder: 256, (3) the dropout rates in the encoder: 0.0, and (4) the dropout rates in the decoder: 0.0. Due to computational requirements, we do not perform the hyperparameter search on every trajectory but use this set of choices for all of the experiments. We argue that there might be other possible values, but the effect on the structural inference results is minor.

### C.12   iSIDG

**Implementation.** We use the official implementation sent by the authors. Same as default, we choose the `GINEncoder` and `MLPDecoder` as the blocks for iSIDG. The original code contains evaluation pipelines to calculate AUROC values. Besides that, we add multiple arguments to select the target trajectories for training, but these arguments do not affect the general mechanism of iSIDG. Our implementation can be found at `https://github.com/wang422003/Benchmarking-Structural-Inference-Methods-for-Interacting-Dynamical-Systems/tree/main/src/models/iSIDG`. The method is implemented with PyTorch [81] with the help of Scikit-Learn [82] to calculate metrics. The AUROC values are calculated between the ground truth adjacency matrix and the `prob` variable in the algorithm.

**Computational resources.** We train iSIDG with two different GPU cards depending on the number of nodes in the trajectories. For the trajectories with less than 50 nodes, we train iSIDG on a single NVIDIA Tesla V100 SXM2 16G GPU card, with 768 GB RAM, and with a single Xeon Gold 6132 @ 2.6GHz CPU. For the trajectories with equal or more than 50 nodes, we train iSIDG on a single NVIDIA Tesla V100 SXM2 32G GPU card, with 768 GB RAM, and with a single Xeon Gold 6132 @ 2.6GHz CPU. We show the batch size for training iSIDG in Table 18. The learning rate we use is identical to the default in iSIDG [106], i.e., 0.0005.

**Hyperparameters.** The hyperparameters that are being considered during implementation are (1) the number of units of the hidden layers in the encoder, (2) the number of units of the hidden layers in the decoder, (3) the dropout rates in the encoder, (4) the dropout rates in the decoder, (5) the weight for KL-divergence in the loss, (6) the weight for smoothness in the loss, (7) the weight for connectiveness in the loss, and (8) the weight for sparsity in the loss, while the rest are set the same as the default. These hyperparameters can be set from the arguments of `arg_parser`. The corresponding search spaces are:

- the number of units of the hidden layers in the encoder: $\{128, 256, 512\}$,
- the number of units of the hidden layers in the decoder: $\{128, 256, 512\}$,
- the dropout rates in the encoder: $\{0.0, 0.3, 0.5, 0.6, 0.7, 0.8\}$,
- the dropout rates in the decoder: $\{0.0, 0.3, 0.5, 0.6, 0.7, 0.8\}$,
- the weight for KL-divergence: $\{100, 200, 300, 400, 500\}$,
- the weight for smoothness: $\{20, 30, 40, 50, 60, 70\}$,
- the weight for connectiveness: $\{10, 20, 30, 40, 50\}$,
- the weight for sparsity: $\{10, 20, 30, 40, 50\}$.

We search for the values of these hyperparameters based on 5 runs of iSIDG on the springs simulation trajectories of CRNA graphs of 15 nodes, and we find the best hyperparameters to be: (1) the number of units of the hidden layers in the encoder: 256, (2) the number of units of the hidden layers in the decoder: 256, (3) the dropout rates in the encoder: 0.0, (4) the dropout rates in the decoder: 0.0, (5) the weight for KL-divergence in the loss: 200, (6) the weight for smoothness in the loss: 50, (7) the weight for connectiveness in the loss: 20, and (8) the weight for sparsity in the loss: 20. Due to computational requirements, we do not perform the hyperparameter search on every trajectory but use this set of choices for all of the experiments. We argue that there might be other possible values, but the effect on the structural inference results is minor.

### C.13   RCSI

**Implementation.**       We   use   the   official   implementation   sent   by   the   authors. The   original   code   contains   evaluation   pipelines   to   calculate   AUROC   val-

ues. Our implementation can be found at `https://github.com/wang422003/Benchmarking-Structural-Inference-Methods-for-Interacting-Dynamical-Systems/tree/main/src/models/RCSI`. The method is implemented with PyTorch [81] with the help of Scikit-Learn [82] to calculate metrics. The AUROC values are calculated between the ground truth adjacency matrix and the `prob` variable in the algorithm.

**Computational resources.** We train RCSI with two different GPU cards depending on the number of nodes in the trajectories. For the trajectories with less than 50 nodes, we train RCSI on a single NVIDIA Tesla V100 SXM2 16G GPU card, with 768 GB RAM, and with a single Xeon Gold 6132 @ 2.6GHz CPU. For the trajectories with equal or more than 50 nodes, we train RCSI on a single NVIDIA Tesla V100 SXM2 32G GPU card, with 768 GB RAM, and with a single Xeon Gold 6132 @ 2.6GHz CPU. We show the batch size for training RCSI in Table 18. The learning rate we use is identical to the default in RCSI [106], i.e., 0.0005.

**Hyperparameters.** The hyperparameters that are being considered during implementation are (1) the number of units of the hidden layers in the encoder, (2) the number of units of the hidden layers in the decoder, (3) the dropout rates in the encoder, (4) the dropout rates in the decoder, (5) the weight for KL-divergence in the loss, (6) the weight for smoothness in the loss, (7) the weight for connectiveness in the loss, and (8) the weight for sparsity in the loss, while the rest are set the same as the default. These hyperparameters can be set from the arguments of `arg_parser`. The corresponding search spaces are:

- the number of units of the hidden layers in the encoder: $\{128, 256, 512\}$,
- the number of units of the hidden layers in the decoder: $\{128, 256, 512\}$,
- the dropout rates in the encoder: $\{0.0, 0.3, 0.5, 0.6, 0.7, 0.8\}$,
- the dropout rates in the decoder: $\{0.0, 0.3, 0.5, 0.6, 0.7, 0.8\}$,
- the weight for KL-divergence: $\{100, 200, 300, 400, 500\}$,
- the weight for smoothness: $\{20, 30, 40, 50, 60, 70\}$,
- the weight for connectiveness: $\{10, 20, 30, 40, 50\}$,
- the weight for sparsity: $\{10, 20, 30, 40, 50\}$.
- the number of neurons in each reservoir computing cell: $\{20, 40, 60, 80\}$,
- the number of reservoir computing cells: $\{1, 2, 3, 4\}$

We search for the values of these hyperparameters based on 5 runs of RCSI on the springs simulation trajectories of CRNA graphs of 15 nodes, and we find the best hyperparameters to be: (1) the number of units of the hidden layers in the encoder: 256, (2) the number of units of the hidden layers in the decoder: 256, (3) the dropout rates in the encoder: 0.0, (4) the dropout rates in the decoder: 0.0, (5) the weight for KL-divergence in the loss: 200, (6) the weight for smoothness in the loss: 50, (7) the weight for connectiveness in the loss: 20, (8) the weight for sparsity in the loss: 20, (9) the number of of neurons in each reservoir computing cell: 20, and (10) the number of reservoir computing cells: 3. Due to computational requirements, we do not perform the hyperparameter search on every trajectory but use this set of choices for all of the experiments. We argue that there might be other possible values, but the effect on the structural inference results is minor.

# D   Further benchmarking results and details

In this section, we present additional experimental results apart from those discussed in Section 5 in the main content.

## D.1   Results on all of the trajectories without noise

The average AUROC values with standard deviations of ten runs of all investigated structural inference methods are presented in Tables 19-29. The results are grouped into each table according to the type of underlying interaction graphs. In each table, the nested column headings indicate the type of simulation and system size used for trajectory generation, e.g., "Springs" and "n30" refer to the trajectories of a system of 30 nodes that are generated by the "Springs" simulation.

Table 19: AUROC values (in %) of investigated structural inference methods on BN trajectories.

| Method | Springs | | | | NetSims | | | |
|---|---|---|---|---|---|---|---|---|
| | n15 | n30 | n50 | n100 | n15 | n30 | n50 | n100 |
| ppcor | - | - | - | - | $96.12_{\pm 0.40}$ | $98.08_{\pm 0.22}$ | $98.83_{\pm 0.09}$ | $99.43_{\pm 0.01}$ |
| TIGRESS | - | - | - | - | $93.14_{\pm 0.67}$ | $96.44_{\pm 0.76}$ | $97.72_{\pm 0.26}$ | $98.72_{\pm 0.04}$ |
| ARACNe | - | - | - | - | $94.10_{\pm 0.66}$ | $96.45_{\pm 0.31}$ | $97.78_{\pm 0.19}$ | $98.83_{\pm 0.03}$ |
| CLR | - | - | - | - | $95.39_{\pm 0.48}$ | $96.72_{\pm 0.56}$ | $97.73_{\pm 0.19}$ | $98.84_{\pm 0.03}$ |
| PIDC | - | - | - | - | $88.45_{\pm 0.61}$ | $93.16_{\pm 0.69}$ | $94.28_{\pm 0.26}$ | $96.17_{\pm 0.12}$ |
| Scribe | - | - | - | - | $48.71_{\pm 1.37}$ | $62.41_{\pm 1.64}$ | $68.79_{\pm 2.53}$ | $69.36_{\pm 1.50}$ |
| dynGENIE3 | - | - | - | - | $90.70_{\pm 2.97}$ | $99.87_{\pm 0.01}$ | $99.89_{\pm 0.00}$ | $99.97_{\pm 0.00}$ |
| XGBGRN | - | - | - | - | $100.00_{\pm 0.00}$ | $100.00_{\pm 0.00}$ | $100.00_{\pm 0.00}$ | $100.00_{\pm 0.00}$ |
| NRI | $99.75_{\pm 0.00}$ | $99.57_{\pm 0.00}$ | $99.12_{\pm 0.01}$ | $97.54_{\pm 0.02}$ | $99.79_{\pm 0.00}$ | $98.73_{\pm 0.00}$ | $76.08_{\pm 0.01}$ | $75.26_{\pm 0.01}$ |
| ACD | $99.75_{\pm 0.00}$ | $99.60_{\pm 0.00}$ | $98.96_{\pm 0.01}$ | $99.57_{\pm 0.01}$ | $99.87_{\pm 0.00}$ | $98.95_{\pm 0.00}$ | $80.96_{\pm 0.01}$ | $79.88_{\pm 0.01}$ |
| MPM | $99.98_{\pm 0.00}$ | $99.95_{\pm 0.00}$ | $99.97_{\pm 0.00}$ | $98.69_{\pm 0.01}$ | $99.95_{\pm 0.00}$ | $99.56_{\pm 0.00}$ | $98.60_{\pm 0.01}$ | $79.92_{\pm 0.01}$ |
| iSIDG | $99.97_{\pm 0.00}$ | $99.94_{\pm 0.00}$ | $99.95_{\pm 0.01}$ | $98.92_{\pm 0.01}$ | $99.91_{\pm 0.00}$ | $99.62_{\pm 0.00}$ | $98.59_{\pm 0.01}$ | $76.41_{\pm 0.01}$ |
| RCSI | $99.81_{\pm 0.01}$ | $99.46_{\pm 0.01}$ | $99.50_{\pm 0.01}$ | $99.04_{\pm 0.01}$ | $99.72_{\pm 0.01}$ | $99.43_{\pm 0.00}$ | $98.60_{\pm 0.01}$ | $80.01_{\pm 0.01}$ |

Table 20: AUROC values (in %) of investigated structural inference methods on CRNA trajectories.

| Method | Springs | | | | NetSims | | | |
|---|---|---|---|---|---|---|---|---|
| | n15 | n30 | n50 | n100 | n15 | n30 | n50 | n100 |
| ppcor | - | - | - | - | $91.37_{\pm 1.10}$ | $90.35_{\pm 0.39}$ | $90.26_{\pm 0.54}$ | $89.16_{\pm 0.55}$ |
| TIGRESS | - | - | - | - | $84.40_{\pm 2.84}$ | $74.88_{\pm 0.64}$ | $69.41_{\pm 0.64}$ | $60.10_{\pm 0.46}$ |
| ARACNe | - | - | - | - | $78.11_{\pm 1.50}$ | $77.93_{\pm 1.00}$ | $77.55_{\pm 0.80}$ | $75.74_{\pm 0.89}$ |
| CLR | - | - | - | - | $86.01_{\pm 1.98}$ | $86.59_{\pm 1.06}$ | $84.24_{\pm 0.76}$ | $81.14_{\pm 1.24}$ |
| PIDC | - | - | - | - | $85.70_{\pm 3.35}$ | $75.38_{\pm 0.42}$ | $70.81_{\pm 1.99}$ | $82.74_{\pm 0.88}$ |
| Scribe | - | - | - | - | $55.19_{\pm 3.80}$ | $52.19_{\pm 0.22}$ | $50.78_{\pm 0.25}$ | $50.94_{\pm 0.74}$ |
| dynGENIE3 | - | - | - | - | $56.92_{\pm 6.83}$ | $50.32_{\pm 1.36}$ | $50.12_{\pm 0.84}$ | $50.35_{\pm 0.60}$ |
| XGBGRN | - | - | - | - | $99.60_{\pm 0.30}$ | $99.58_{\pm 0.13}$ | $97.40_{\pm 0.52}$ | $51.48_{\pm 0.22}$ |
| NRI | $83.91_{\pm 0.03}$ | $72.81_{\pm 0.05}$ | $70.73_{\pm 0.02}$ | $65.32_{\pm 0.02}$ | $49.47_{\pm 0.02}$ | $49.03_{\pm 0.03}$ | $50.06_{\pm 0.02}$ | $50.65_{\pm 0.02}$ |
| ACD | $85.90_{\pm 0.04}$ | $75.41_{\pm 0.01}$ | $69.97_{\pm 0.01}$ | $64.51_{\pm 0.02}$ | $48.26_{\pm 0.02}$ | $48.40_{\pm 0.03}$ | $51.42_{\pm 0.01}$ | $50.21_{\pm 0.02}$ |
| MPM | $85.75_{\pm 0.03}$ | $73.71_{\pm 0.01}$ | $68.25_{\pm 0.02}$ | $64.87_{\pm 0.02}$ | $49.72_{\pm 0.01}$ | $51.16_{\pm 0.04}$ | $50.06_{\pm 0.01}$ | $50.56_{\pm 0.02}$ |
| iSIDG | $87.01_{\pm 0.02}$ | $78.21_{\pm 0.05}$ | $70.72_{\pm 0.01}$ | $62.31_{\pm 0.02}$ | $51.04_{\pm 0.01}$ | $50.24_{\pm 0.04}$ | $51.26_{\pm 0.01}$ | $50.87_{\pm 0.02}$ |
| RCSI | $87.51_{\pm 0.02}$ | $78.11_{\pm 0.05}$ | $69.82_{\pm 0.02}$ | $64.80_{\pm 0.03}$ | $51.15_{\pm 0.02}$ | $50.81_{\pm 0.04}$ | $51.10_{\pm 0.02}$ | $50.00_{\pm 0.02}$ |

Table 21: AUROC values (in %) of investigated structural inference methods on FW trajectories.

| Method | Springs | | | | NetSims | | | |
|---|---|---|---|---|---|---|---|---|
| | n15 | n30 | n50 | n100 | n15 | n30 | n50 | n100 |
| ppcor | - | - | - | - | $78.21_{\pm 1.57}$ | $73.63_{\pm 1.75}$ | $72.76_{\pm 1.04}$ | $71.72_{\pm 0.15}$ |
| TIGRESS | - | - | - | - | $64.15_{\pm 1.55}$ | $58.00_{\pm 0.61}$ | $57.92_{\pm 0.84}$ | $53.97_{\pm 0.44}$ |
| ARACNe | - | - | - | - | $66.07_{\pm 4.26}$ | $65.40_{\pm 3.82}$ | $68.39_{\pm 0.24}$ | $53.18_{\pm 2.03}$ |
| CLR | - | - | - | - | $79.69_{\pm 3.33}$ | $74.20_{\pm 1.57}$ | $73.94_{\pm 1.01}$ | $44.50_{\pm 2.24}$ |
| PIDC | - | - | - | - | $78.82_{\pm 3.75}$ | $50.00_{\pm 0.00}$ | $50.00_{\pm 0.00}$ | $64.72_{\pm 1.39}$ |
| Scribe | - | - | - | - | $52.96_{\pm 2.66}$ | $54.25_{\pm 1.16}$ | $51.02_{\pm 1.59}$ | $51.73_{\pm 0.92}$ |
| dynGENIE3 | - | - | - | - | $47.98_{\pm 2.67}$ | $49.89_{\pm 1.29}$ | $49.40_{\pm 0.58}$ | $51.26_{\pm 1.07}$ |
| XGBGRN | - | - | - | - | $84.84_{\pm 1.90}$ | $73.00_{\pm 4.00}$ | $52.36_{\pm 0.35}$ | $49.11_{\pm 0.77}$ |
| NRI | $81.80_{\pm 0.01}$ | $76.75_{\pm 0.02}$ | $74.15_{\pm 0.01}$ | $71.57_{\pm 0.01}$ | $49.30_{\pm 0.03}$ | $48.50_{\pm 0.03}$ | $50.75_{\pm 0.02}$ | $47.56_{\pm 0.03}$ |
| ACD | $81.89_{\pm 0.01}$ | $76.38_{\pm 0.02}$ | $73.50_{\pm 0.01}$ | $71.12_{\pm 0.01}$ | $50.74_{\pm 0.06}$ | $50.19_{\pm 0.01}$ | $50.49_{\pm 0.03}$ | $49.82_{\pm 0.01}$ |
| MPM | $81.87_{\pm 0.02}$ | $75.97_{\pm 0.01}$ | $73.59_{\pm 0.01}$ | $71.52_{\pm 0.01}$ | $53.01_{\pm 0.08}$ | $50.66_{\pm 0.008}$ | $51.22_{\pm 0.03}$ | $53.01_{\pm 0.03}$ |
| iSIDG | $81.95_{\pm 0.01}$ | $76.75_{\pm 0.01}$ | $74.38_{\pm 0.02}$ | $72.21_{\pm 0.02}$ | $53.36_{\pm 0.03}$ | $50.78_{\pm 0.03}$ | $50.46_{\pm 0.03}$ | $51.07_{\pm 0.01}$ |
| RCSI | $81.80_{\pm 0.01}$ | $75.62_{\pm 0.02}$ | $74.51_{\pm 0.02}$ | $72.40_{\pm 0.02}$ | $53.70_{\pm 0.05}$ | $50.28_{\pm 0.03}$ | $51.61_{\pm 0.03}$ | $53.82_{\pm 0.02}$ |

## D.2 Benchmarking over robustness

In this section, we summarized the AUROC results of all methods on trajectories with noise generated with BN and NetSims simulations. The average AUROC values and corresponding standard deviations of all investigated methods are presented in Tables 30 - 32. The results are grouped by two levels of headings, i.e., the level of Gaussian noise, and the number of nodes in the graph.

Table 22: AUROC values (in %) of investigated structural inference methods on GCN trajectories.

| Method | Springs | | | | NetSims | | | |
|---|---|---|---|---|---|---|---|---|
| | n15 | n30 | n50 | n100 | n15 | n30 | n50 | n100 |
| ppcor | - | - | - | - | $96.72_{\pm 1.64}$ | $98.48_{\pm 0.35}$ | $98.55_{\pm 0.22}$ | $98.20_{\pm 0.42}$ |
| TIGRESS | - | - | - | - | $91.72_{\pm 4.28}$ | $87.90_{\pm 1.44}$ | $80.44_{\pm 1.78}$ | $78.12_{\pm 0.15}$ |
| ARACNe | - | - | - | - | $95.24_{\pm 0.00}$ | $91.15_{\pm 2.18}$ | $92.75_{\pm 1.94}$ | $94.04_{\pm 0.71}$ |
| CLR | - | - | - | - | $94.57_{\pm 2.14}$ | $97.48_{\pm 0.54}$ | $97.25_{\pm 0.76}$ | $96.40_{\pm 0.21}$ |
| PIDC | - | - | - | - | $92.75_{\pm 3.92}$ | $91.98_{\pm 0.90}$ | $92.01_{\pm 1.23}$ | $94.17_{\pm 1.25}$ |
| Scribe | - | - | - | - | $50.47_{\pm 2.55}$ | $49.31_{\pm 1.72}$ | $48.17_{\pm 2.80}$ | $49.51_{\pm 0.77}$ |
| dynGENIE3 | - | - | - | - | $46.70_{\pm 5.05}$ | $47.86_{\pm 4.04}$ | $50.46_{\pm 1.93}$ | $49.58_{\pm 1.37}$ |
| XGBGRN | - | - | - | - | $93.28_{\pm 2.47}$ | $96.71_{\pm 0.51}$ | $96.74_{\pm 0.62}$ | $94.95_{\pm 0.33}$ |
| NRI | $97.42_{\pm 0.00}$ | $93.38_{\pm 0.01}$ | $89.54_{\pm 0.02}$ | $83.78_{\pm 0.01}$ | $43.46_{\pm 0.02}$ | $52.74_{\pm 0.06}$ | $50.98_{\pm 0.02}$ | $50.34_{\pm 0.02}$ |
| ACD | $97.95_{\pm 0.01}$ | $92.62_{\pm 0.01}$ | $89.96_{\pm 0.03}$ | $90.73_{\pm 0.02}$ | $42.23_{\pm 0.03}$ | $46.12_{\pm 0.04}$ | $47.66_{\pm 0.03}$ | $49.87_{\pm 0.04}$ |
| MPM | $98.82_{\pm 0.01}$ | $92.68_{\pm 0.02}$ | $85.81_{\pm 0.03}$ | $84.98_{\pm 0.02}$ | $52.59_{\pm 0.03}$ | $66.65_{\pm 0.07}$ | $63.01_{\pm 0.07}$ | $53.07_{\pm 0.06}$ |
| iSIDG | $98.93_{\pm 0.01}$ | $93.16_{\pm 0.01}$ | $89.53_{\pm 0.01}$ | $87.60_{\pm 0.01}$ | $56.41_{\pm 0.06}$ | $52.07_{\pm 0.03}$ | $52.96_{\pm 0.03}$ | $50.78_{\pm 0.03}$ |
| RCSI | $97.66_{\pm 0.01}$ | $93.92_{\pm 0.02}$ | $88.69_{\pm 0.02}$ | $88.01_{\pm 0.02}$ | $53.31_{\pm 0.05}$ | $59.23_{\pm 0.03}$ | $54.30_{\pm 0.02}$ | $50.44_{\pm 0.02}$ |

Table 23: AUROC values (in %) of investigated structural inference methods on GRN trajectories.

| Method | Springs | | | | NetSims | | | |
|---|---|---|---|---|---|---|---|---|
| | n15 | n30 | n50 | n100 | n15 | n30 | n50 | n100 |
| ppcor | - | - | - | - | $86.12_{\pm 0.98}$ | $88.72_{\pm 1.33}$ | $89.83_{\pm 0.89}$ | $89.61_{\pm 0.93}$ |
| TIGRESS | - | - | - | - | $79.09_{\pm 1.07}$ | $85.16_{\pm 2.26}$ | $85.85_{\pm 1.96}$ | $87.41_{\pm 2.73}$ |
| ARACNe | - | - | - | - | $70.46_{\pm 3.52}$ | $70.05_{\pm 2.10}$ | $70.73_{\pm 1.90}$ | $69.48_{\pm 2.10}$ |
| CLR | - | - | - | - | $78.25_{\pm 0.49}$ | $76.48_{\pm 1.91}$ | $75.67_{\pm 1.29}$ | $73.09_{\pm 2.10}$ |
| PIDC | - | - | - | - | $57.49_{\pm 3.59}$ | $63.51_{\pm 2.69}$ | $65.95_{\pm 1.41}$ | $63.85_{\pm 2.00}$ |
| Scribe | - | - | - | - | $44.89_{\pm 7.52}$ | $47.79_{\pm 3.50}$ | $45.50_{\pm 3.03}$ | $46.15_{\pm 2.41}$ |
| dynGENIE3 | - | - | - | - | $64.23_{\pm 4.75}$ | $59.69_{\pm 6.09}$ | $54.38_{\pm 3.18}$ | $58.53_{\pm 3.94}$ |
| XGBGRN | - | - | - | - | $80.08_{\pm 3.81}$ | $83.77_{\pm 0.49}$ | $84.51_{\pm 0.43}$ | $83.47_{\pm 1.31}$ |
| NRI | $91.65_{\pm 0.01}$ | $90.45_{\pm 0.01}$ | $90.35_{\pm 0.02}$ | $88.14_{\pm 0.02}$ | $78.08_{\pm 0.03}$ | $57.01_{\pm 0.05}$ | $55.71_{\pm 0.05}$ | $58.33_{\pm 0.04}$ |
| ACD | $91.10_{\pm 0.00}$ | $88.21_{\pm 0.01}$ | $86.78_{\pm 0.01}$ | $90.07_{\pm 0.03}$ | $80.18_{\pm 0.04}$ | $69.78_{\pm 0.07}$ | $62.65_{\pm 0.02}$ | $53.99_{\pm 0.03}$ |
| MPM | $94.02_{\pm 0.01}$ | $93.25_{\pm 0.02}$ | $84.60_{\pm 0.02}$ | $85.30_{\pm 0.02}$ | $70.46_{\pm 0.04}$ | $57.36_{\pm 0.03}$ | $72.25_{\pm 0.05}$ | $66.74_{\pm 0.03}$ |
| iSIDG | $92.91_{\pm 0.01}$ | $90.06_{\pm 0.01}$ | $90.15_{\pm 0.01}$ | $87.94_{\pm 0.04}$ | $71.11_{\pm 0.04}$ | $56.25_{\pm 0.02}$ | $57.15_{\pm 0.02}$ | $62.13_{\pm 0.02}$ |
| RCSI | $93.88_{\pm 0.02}$ | $93.01_{\pm 0.02}$ | $90.35_{\pm 0.01}$ | $89.90_{\pm 0.03}$ | $77.45_{\pm 0.03}$ | $65.77_{\pm 0.03}$ | $59.93_{\pm 0.02}$ | $60.15_{\pm 0.03}$ |

Table 24: AUROC values (in %) of investigated structural inference methods on IN trajectories.

| Method | Springs | | | | NetSims | | | |
|---|---|---|---|---|---|---|---|---|
| | n15 | n30 | n50 | n100 | n15 | n30 | n50 | n100 |
| ppcor | - | - | - | - | $94.14_{\pm 0.54}$ | $96.13_{\pm 1.65}$ | $97.64_{\pm 0.11}$ | $97.61_{\pm 0.01}$ |
| TIGRESS | - | - | - | - | $94.39_{\pm 1.34}$ | $91.79_{\pm 5.82}$ | $86.31_{\pm 1.42}$ | $78.25_{\pm 0.52}$ |
| ARACNe | - | - | - | - | $85.82_{\pm 3.90}$ | $87.77_{\pm 5.36}$ | $83.05_{\pm 2.84}$ | $86.14_{\pm 0.54}$ |
| CLR | - | - | - | - | $87.17_{\pm 0.26}$ | $92.45_{\pm 2.50}$ | $89.58_{\pm 3.93}$ | $92.82_{\pm 1.03}$ |
| PIDC | - | - | - | - | $81.90_{\pm 1.92}$ | $85.16_{\pm 1.59}$ | $84.84_{\pm 2.89}$ | $89.35_{\pm 0.48}$ |
| Scribe | - | - | - | - | $54.29_{\pm 4.17}$ | $50.81_{\pm 1.34}$ | $50.68_{\pm 3.52}$ | $50.76_{\pm 0.53}$ |
| dynGENIE3 | - | - | - | - | $58.18_{\pm 4.97}$ | $70.18_{\pm 15.42}$ | $68.08_{\pm 8.25}$ | $50.22_{\pm 1.78}$ |
| XGBGRN | - | - | - | - | $99.00_{\pm 0.85}$ | $99.69_{\pm 0.07}$ | $99.90_{\pm 0.04}$ | $99.91_{\pm 0.05}$ |
| NRI | $93.09_{\pm 0.01}$ | $90.54_{\pm 0.05}$ | $88.10_{\pm 0.03}$ | $82.51_{\pm 0.03}$ | $60.47_{\pm 0.04}$ | $61.78_{\pm 0.06}$ | $56.45_{\pm 0.04}$ | $53.96_{\pm 0.04}$ |
| ACD | $93.33_{\pm 0.02}$ | $89.12_{\pm 0.05}$ | $87.69_{\pm 0.04}$ | $81.37_{\pm 0.02}$ | $68.39_{\pm 0.06}$ | $55.11_{\pm 0.08}$ | $53.88_{\pm 0.02}$ | $53.04_{\pm 0.05}$ |
| MPM | $95.61_{\pm 0.02}$ | $89.59_{\pm 0.05}$ | $86.47_{\pm 0.03}$ | $83.45_{\pm 0.03}$ | $63.83_{\pm 0.03}$ | $64.70_{\pm 0.09}$ | $54.18_{\pm 0.03}$ | $54.37_{\pm 0.04}$ |
| iSIDG | $95.37_{\pm 0.02}$ | $90.72_{\pm 0.05}$ | $87.79_{\pm 0.02}$ | $84.00_{\pm 0.02}$ | $62.18_{\pm 0.03}$ | $61.91_{\pm 0.01}$ | $56.50_{\pm 0.02}$ | $53.85_{\pm 0.02}$ |
| RCSI | $96.70_{\pm 0.01}$ | $93.46_{\pm 0.02}$ | $88.02_{\pm 0.02}$ | $83.96_{\pm 0.01}$ | $62.08_{\pm 0.04}$ | $60.05_{\pm 0.02}$ | $55.65_{\pm 0.03}$ | $52.84_{\pm 0.02}$ |

## D.3 Benchmarking over efficiency

Investigating the potential influence of trajectory lengths on the performance of structural inference methods is of significant interest. Additionally, such evaluations shed light on the data efficiency of these methods by examining the number of time steps required to yield reliable results. To explore these aspects, we conducted evaluations using trajectories generated by BN with varying numbers of time steps (lengths). The selected time step counts include $10, 20, 30, 40, 49$, with 49 representing the full-length trajectories. By comparing the average AUROC results between shorter and full-length trajectories, we computed the differences $\Delta \text{AUROC} = \text{AUROC}_{TS} - \text{AUROC}_{raw}$, where $\text{AUROC}_{TS}$ denotes the average AUROC results with shorter trajectories, and $\text{AUROC}_{raw}$ represents the average

Table 25: AUROC values (in %) of investigated structural inference methods on LN trajectories.

| Method | Springs | | | | NetSims | | | |
|---|---|---|---|---|---|---|---|---|
| | n15 | n30 | n50 | n100 | n15 | n30 | n50 | n100 |
| ppcor | - | - | - | - | $99.49_{\pm 0.56}$ | $95.04_{\pm 5.20}$ | $86.75_{\pm 1.66}$ | $79.32_{\pm 4.32}$ |
| TIGRESS | - | - | - | - | $84.15_{\pm 1.16}$ | $87.38_{\pm 3.32}$ | $92.22_{\pm 0.42}$ | $93.97_{\pm 1.96}$ |
| ARACNe | - | - | - | - | $92.33_{\pm 4.84}$ | $80.36_{\pm 5.67}$ | $71.17_{\pm 0.48}$ | $62.82_{\pm 8.36}$ |
| CLR | - | - | - | - | $97.35_{\pm 3.17}$ | $96.56_{\pm 4.87}$ | $91.04_{\pm 2.35}$ | $95.04_{\pm 0.53}$ |
| PIDC | - | - | - | - | $97.53_{\pm 1.01}$ | $82.03_{\pm 7.28}$ | $88.58_{\pm 1.69}$ | $94.18_{\pm 2.28}$ |
| Scribe | - | - | - | - | $54.22_{\pm 3.98}$ | $56.16_{\pm 3.88}$ | $52.12_{\pm 2.49}$ | $52.55_{\pm 1.62}$ |
| dynGENIE3 | - | - | - | - | $51.32_{\pm 5.21}$ | $50.12_{\pm 2.42}$ | $50.49_{\pm 1.22}$ | $67.32_{\pm 14.23}$ |
| XGBGRN | - | - | - | - | $97.21_{\pm 1.13}$ | $96.95_{\pm 2.10}$ | $96.90_{\pm 0.83}$ | $97.99_{\pm 0.93}$ |
| NRI | $97.01_{\pm 0.02}$ | $94.94_{\pm 0.00}$ | $87.10_{\pm 0.01}$ | $82.80_{\pm 0.01}$ | $56.00_{\pm 0.04}$ | $53.94_{\pm 0.02}$ | $54.36_{\pm 0.02}$ | $51.75_{\pm 0.03}$ |
| ACD | $96.99_{\pm 0.02}$ | $95.79_{\pm 0.01}$ | $87.58_{\pm 0.02}$ | $83.92_{\pm 0.02}$ | $61.94_{\pm 0.03}$ | $61.56_{\pm 0.04}$ | $53.36_{\pm 0.02}$ | $50.19_{\pm 0.02}$ |
| MPM | $97.92_{\pm 0.01}$ | $95.53_{\pm 0.02}$ | $86.92_{\pm 0.01}$ | $84.22_{\pm 0.03}$ | $52.18_{\pm 0.02}$ | $62.08_{\pm 0.05}$ | $53.44_{\pm 0.01}$ | $50.42_{\pm 0.03}$ |
| iSIDG | $97.38_{\pm 0.02}$ | $94.70_{\pm 0.02}$ | $87.44_{\pm 0.02}$ | $83.15_{\pm 0.02}$ | $59.19_{\pm 0.05}$ | $56.18_{\pm 0.03}$ | $55.73_{\pm 0.03}$ | $52.30_{\pm 0.02}$ |
| RCSI | $97.30_{\pm 0.02}$ | $94.42_{\pm 0.02}$ | $88.02_{\pm 0.02}$ | $84.26_{\pm 0.02}$ | $60.28_{\pm 0.03}$ | $60.15_{\pm 0.02}$ | $57.56_{\pm 0.03}$ | $53.48_{\pm 0.02}$ |

Table 26: AUROC values (in %) of investigated structural inference methods on MMO trajectories.

| Method | Springs | | | | NetSims | | | |
|---|---|---|---|---|---|---|---|---|
| | n15 | n30 | n50 | n100 | n15 | n30 | n50 | n100 |
| ppcor | - | - | - | - | $96.42_{\pm 0.02}$ | $98.28_{\pm 0.00}$ | $98.98_{\pm 0.00}$ | $99.49_{\pm 0.00}$ |
| TIGRESS | - | - | - | - | $99.88_{\pm 0.01}$ | $99.98_{\pm 0.00}$ | $100.00_{\pm 0.00}$ | $100.00_{\pm 0.00}$ |
| ARACNe | - | - | - | - | $89.76_{\pm 0.16}$ | $96.60_{\pm 1.51}$ | $97.09_{\pm 1.07}$ | $98.11_{\pm 0.79}$ |
| CLR | - | - | - | - | $96.43_{\pm 0.00}$ | $98.28_{\pm 0.00}$ | $98.98_{\pm 0.00}$ | $98.81_{\pm 0.37}$ |
| PIDC | - | - | - | - | $44.74_{\pm 4.70}$ | $70.03_{\pm 7.65}$ | $77.24_{\pm 1.02}$ | $75.01_{\pm 0.29}$ |
| Scribe | - | - | - | - | $69.85_{\pm 12.21}$ | $38.03_{\pm 25.86}$ | $20.70_{\pm 10.19}$ | $23.88_{\pm 15.76}$ |
| dynGENIE3 | - | - | - | - | $16.90_{\pm 2.38}$ | $23.49_{\pm 5.12}$ | $23.31_{\pm 4.03}$ | $45.89_{\pm 20.23}$ |
| XGBGRN | - | - | - | - | $59.77_{\pm 2.14}$ | $81.64_{\pm 6.68}$ | $72.13_{\pm 11.09}$ | $63.83_{\pm 6.71}$ |
| NRI | $99.62_{\pm 0.00}$ | $84.96_{\pm 0.02}$ | $77.66_{\pm 0.01}$ | $78.04_{\pm 0.02}$ | $68.34_{\pm 0.03}$ | $66.21_{\pm 0.06}$ | $57.84_{\pm 0.03}$ | $56.10_{\pm 0.01}$ |
| ACD | $99.68_{\pm 0.00}$ | $85.53_{\pm 0.02}$ | $85.53_{\pm 0.02}$ | $85.46_{\pm 0.01}$ | $71.88_{\pm 0.03}$ | $59.46_{\pm 0.06}$ | $64.14_{\pm 0.03}$ | $58.05_{\pm 0.02}$ |
| MPM | $99.83_{\pm 0.00}$ | $88.32_{\pm 0.01}$ | $87.02_{\pm 0.03}$ | $86.75_{\pm 0.02}$ | $79.34_{\pm 0.04}$ | $65.48_{\pm 0.07}$ | $54.78_{\pm 0.04}$ | $57.06_{\pm 0.02}$ |
| iSIDG | $99.84_{\pm 0.00}$ | $89.77_{\pm 0.01}$ | $87.47_{\pm 0.02}$ | $85.47_{\pm 0.01}$ | $74.58_{\pm 0.03}$ | $64.71_{\pm 0.06}$ | $56.07_{\pm 0.04}$ | $58.80_{\pm 0.01}$ |
| RCSI | $99.70_{\pm 0.01}$ | $92.73_{\pm 0.02}$ | $88.05_{\pm 0.02}$ | $85.49_{\pm 0.02}$ | $73.61_{\pm 0.04}$ | $66.08_{\pm 0.02}$ | $57.90_{\pm 0.03}$ | $58.74_{\pm 0.02}$ |

Table 27: AUROC values (in %) of investigated structural inference methods on RNLO trajectories.

| Method | Springs | | | | NetSims | | | |
|---|---|---|---|---|---|---|---|---|
| | n15 | n30 | n50 | n100 | n15 | n30 | n50 | n100 |
| ppcor | - | - | - | - | $96.36_{\pm 0.10}$ | $98.28_{\pm 0.00}$ | $98.95_{\pm 0.04}$ | $99.25_{\pm 0.38}$ |
| TIGRESS | - | - | - | - | $99.82_{\pm 0.06}$ | $99.98_{\pm 0.00}$ | $99.99_{\pm 0.00}$ | $99.99_{\pm 0.01}$ |
| ARACNe | - | - | - | - | $93.47_{\pm 2.99}$ | $95.67_{\pm 1.61}$ | $97.02_{\pm 0.86}$ | $98.03_{\pm 0.43}$ |
| CLR | - | - | - | - | $96.35_{\pm 0.12}$ | $98.28_{\pm 0.00}$ | $98.72_{\pm 0.31}$ | $98.62_{\pm 0.29}$ |
| PIDC | - | - | - | - | $56.18_{\pm 6.51}$ | $72.67_{\pm 10.76}$ | $74.36_{\pm 6.83}$ | $71.95_{\pm 2.31}$ |
| Scribe | - | - | - | - | $38.49_{\pm 1.57}$ | $47.15_{\pm 18.16}$ | $46.52_{\pm 26.84}$ | $20.23_{\pm 13.56}$ |
| dynGENIE3 | - | - | - | - | $15.96_{\pm 2.97}$ | $21.37_{\pm 8.84}$ | $27.57_{\pm 7.69}$ | $56.44_{\pm 21.63}$ |
| XGBGRN | - | - | - | - | $83.55_{\pm 8.24}$ | $81.05_{\pm 5.42}$ | $81.82_{\pm 5.07}$ | $67.30_{\pm 12.31}$ |
| NRI | $95.54_{\pm 0.02}$ | $72.53_{\pm 0.08}$ | $72.72_{\pm 0.03}$ | $75.07_{\pm 0.02}$ | $69.43_{\pm 0.04}$ | $67.70_{\pm 0.08}$ | $60.55_{\pm 0.03}$ | $62.42_{\pm 0.02}$ |
| ACD | $96.20_{\pm 0.02}$ | $93.44_{\pm 0.03}$ | $75.83_{\pm 0.02}$ | $79.14_{\pm 0.02}$ | $57.32_{\pm 0.05}$ | $53.75_{\pm 0.01}$ | $61.68_{\pm 0.05}$ | $65.45_{\pm 0.03}$ |
| MPM | $97.40_{\pm 0.01}$ | $83.70_{\pm 0.06}$ | $78.50_{\pm 0.02}$ | $79.36_{\pm 0.02}$ | $72.62_{\pm 0.03}$ | $62.34_{\pm 0.01}$ | $56.90_{\pm 0.05}$ | $60.05_{\pm 0.02}$ |
| iSIDG | $97.45_{\pm 0.01}$ | $81.60_{\pm 0.05}$ | $78.51_{\pm 0.03}$ | $79.08_{\pm 0.03}$ | $64.79_{\pm 0.05}$ | $57.10_{\pm 0.02}$ | $64.50_{\pm 0.05}$ | $66.01_{\pm 0.02}$ |
| RCSI | $97.30_{\pm 0.01}$ | $83.05_{\pm 0.03}$ | $80.43_{\pm 0.02}$ | $79.04_{\pm 0.02}$ | $69.92_{\pm 0.04}$ | $59.42_{\pm 0.03}$ | $60.99_{\pm 0.04}$ | $60.24_{\pm 0.02}$ |

AUROC results with full-length trajectories. The results are presented in Fig. 5. These findings provide insights into the impact of trajectory lengths on the performance and efficiency of structural inference methods.

The performance of the majority of the methods investigated tends to decrease as the trajectory lengths shorten, as evident in Fig. 5, where most methods show a decline in AUROC values with decreasing trajectory lengths across various graph sizes. This reduction in performance is largely due to the limited information available in shorter trajectories, which constrains the methods' capacity to accurately infer the underlying structures. Notably, ARACNe, CLR, and PIDC behave differently; they show improved performance with shorter trajectories. ARACNe and CLR experience a decline in performance due to the removal of correctly predicted edges when the number of time steps

Table 28: AUROC values (in %) of investigated structural inference methods on SN trajectories.

| Method | Springs | | | | NetSims | | | |
|---|---|---|---|---|---|---|---|---|
| | n15 | n30 | n50 | n100 | n15 | n30 | n50 | n100 |
| ppcor | - | - | - | - | $93.77_{\pm 0.59}$ | $94.17_{\pm 0.28}$ | $94.74_{\pm 0.44}$ | $94.37_{\pm 0.05}$ |
| TIGRESS | - | - | - | - | $90.20_{\pm 1.52}$ | $82.82_{\pm 0.30}$ | $78.22_{\pm 1.92}$ | $67.98_{\pm 0.57}$ |
| ARACNe | - | - | - | - | $80.80_{\pm 3.58}$ | $78.78_{\pm 3.00}$ | $80.42_{\pm 1.00}$ | $81.49_{\pm 0.32}$ |
| CLR | - | - | - | - | $85.08_{\pm 0.54}$ | $87.70_{\pm 1.11}$ | $89.81_{\pm 0.74}$ | $88.24_{\pm 0.60}$ |
| PIDC | - | - | - | - | $83.96_{\pm 2.44}$ | $84.29_{\pm 1.00}$ | $84.66_{\pm 0.70}$ | $91.76_{\pm 0.25}$ |
| Scribe | - | - | - | - | $56.52_{\pm 2.94}$ | $51.30_{\pm 0.50}$ | $50.38_{\pm 0.50}$ | $50.74_{\pm 1.01}$ |
| dynGENIE3 | - | - | - | - | $62.48_{\pm 5.44}$ | $55.74_{\pm 3.23}$ | $50.00_{\pm 1.70}$ | $50.20_{\pm 0.77}$ |
| XGBGRN | - | - | - | - | $99.83_{\pm 0.21}$ | $99.88_{\pm 0.07}$ | $99.74_{\pm 0.12}$ | $98.81_{\pm 0.12}$ |
| NRI | $93.26_{\pm 0.01}$ | $79.96_{\pm 0.02}$ | $80.40_{\pm 0.02}$ | $71.84_{\pm 0.01}$ | $58.41_{\pm 0.04}$ | $51.43_{\pm 0.01}$ | $49.57_{\pm 0.03}$ | $50.16_{\pm 0.03}$ |
| ACD | $93.47_{\pm 0.01}$ | $81.17_{\pm 0.01}$ | $79.63_{\pm 0.02}$ | $68.76_{\pm 0.02}$ | $65.24_{\pm 0.05}$ | $52.96_{\pm 0.03}$ | $49.28_{\pm 0.02}$ | $50.76_{\pm 0.01}$ |
| MPM | $92.68_{\pm 0.00}$ | $79.32_{\pm 0.01}$ | $75.90_{\pm 0.01}$ | $69.36_{\pm 0.03}$ | $67.42_{\pm 0.02}$ | $50.87_{\pm 0.01}$ | $53.12_{\pm 0.03}$ | $50.08_{\pm 0.02}$ |
| iSIDG | $93.51_{\pm 0.00}$ | $81.38_{\pm 0.01}$ | $80.80_{\pm 0.02}$ | $69.25_{\pm 0.01}$ | $66.14_{\pm 0.04}$ | $53.79_{\pm 0.03}$ | $54.83_{\pm 0.01}$ | $51.72_{\pm 0.02}$ |
| RCSI | $94.13_{\pm 0.02}$ | $82.66_{\pm 0.01}$ | $81.21_{\pm 0.01}$ | $73.42_{\pm 0.02}$ | $67.58_{\pm 0.03}$ | $55.84_{\pm 0.02}$ | $55.04_{\pm 0.02}$ | $53.24_{\pm 0.03}$ |

Table 29: AUROC values (in %) of investigated structural inference methods on VN trajectories.

| Method | Springs | | | | NetSims | | | |
|---|---|---|---|---|---|---|---|---|
| | n15 | n30 | n50 | n100 | n15 | n30 | n50 | n100 |
| ppcor | - | - | - | - | $96.68_{\pm 0.01}$ | $98.33_{\pm 0.01}$ | $99.00_{\pm 0.00}$ | $99.50_{\pm 0.00}$ |
| TIGRESS | - | - | - | - | $99.28_{\pm 0.18}$ | $99.41_{\pm 0.15}$ | $99.62_{\pm 0.09}$ | $99.84_{\pm 0.02}$ |
| ARACNe | - | - | - | - | $96.66_{\pm 0.03}$ | $97.85_{\pm 0.09}$ | $98.54_{\pm 0.01}$ | $99.08_{\pm 0.00}$ |
| CLR | - | - | - | - | $96.68_{\pm 0.00}$ | $98.34_{\pm 0.00}$ | $99.00_{\pm 0.00}$ | $99.50_{\pm 0.00}$ |
| PIDC | - | - | - | - | $76.51_{\pm 2.67}$ | $85.70_{\pm 3.99}$ | $91.80_{\pm 0.43}$ | $95.01_{\pm 0.70}$ |
| Scribe | - | - | - | - | $51.56_{\pm 5.64}$ | $52.71_{\pm 4.98}$ | $57.68_{\pm 2.56}$ | $59.50_{\pm 0.83}$ |
| dynGENIE3 | - | - | - | - | $92.81_{\pm 2.83}$ | $97.33_{\pm 1.01}$ | $97.87_{\pm 0.66}$ | $97.30_{\pm 1.26}$ |
| XGBGRN | - | - | - | - | $97.99_{\pm 0.49}$ | $98.54_{\pm 0.38}$ | $99.21_{\pm 0.12}$ | $99.59_{\pm 0.02}$ |
| NRI | $94.58_{\pm 0.01}$ | $95.12_{\pm 0.01}$ | $94.65_{\pm 0.02}$ | $89.17_{\pm 0.02}$ | $90.31_{\pm 0.01}$ | $74.64_{\pm 0.04}$ | $69.78_{\pm 0.03}$ | $68.80_{\pm 0.02}$ |
| ACD | $94.34_{\pm 0.01}$ | $93.73_{\pm 0.01}$ | $87.54_{\pm 0.03}$ | $90.49_{\pm 0.03}$ | $80.32_{\pm 0.02}$ | $65.36_{\pm 0.06}$ | $69.01_{\pm 0.03}$ | $68.72_{\pm 0.03}$ |
| MPM | $96.56_{\pm 0.01}$ | $89.71_{\pm 0.04}$ | $85.07_{\pm 0.02}$ | $84.56_{\pm 0.03}$ | $91.18_{\pm 0.01}$ | $83.37_{\pm 0.03}$ | $72.66_{\pm 0.04}$ | $70.34_{\pm 0.03}$ |
| iSIDG | $96.59_{\pm 0.02}$ | $95.66_{\pm 0.01}$ | $95.72_{\pm 0.02}$ | $85.07_{\pm 0.02}$ | $91.20_{\pm 0.02}$ | $78.08_{\pm 0.06}$ | $73.68_{\pm 0.02}$ | $68.81_{\pm 0.02}$ |
| RCSI | $97.03_{\pm 0.01}$ | $95.31_{\pm 0.01}$ | $94.48_{\pm 0.02}$ | $90.72_{\pm 0.03}$ | $91.53_{\pm 0.02}$ | $82.27_{\pm 0.04}$ | $74.08_{\pm 0.02}$ | $70.29_{\pm 0.03}$ |

Table 30: AUROC values (in %) of investigated structural inference methods on BN_NS trajectories with 1 (N1) and 2 (N2) levels of Gaussian noise.

| Method | N1 | | | | N2 | | | |
|---|---|---|---|---|---|---|---|---|
| | n15 | n30 | n50 | n100 | n15 | n30 | n50 | n100 |
| ppcor | $92.66_{\pm 0.80}$ | $97.16_{\pm 0.59}$ | $98.48_{\pm 0.19}$ | $99.30_{\pm 0.02}$ | $91.25_{\pm 0.75}$ | $96.68_{\pm 0.64}$ | $98.28_{\pm 0.22}$ | $99.21_{\pm 0.03}$ |
| TIGRESS | $93.08_{\pm 0.76}$ | $96.42_{\pm 0.67}$ | $97.59_{\pm 0.23}$ | $98.65_{\pm 0.05}$ | $93.12_{\pm 0.80}$ | $96.43_{\pm 0.62}$ | $97.55_{\pm 0.24}$ | $98.59_{\pm 0.05}$ |
| ARACNe | $84.73_{\pm 1.20}$ | $91.90_{\pm 1.00}$ | $95.84_{\pm 0.33}$ | $98.11_{\pm 0.11}$ | $84.39_{\pm 1.04}$ | $92.37_{\pm 0.98}$ | $95.73_{\pm 0.34}$ | $97.76_{\pm 0.13}$ |
| CLR | $91.46_{\pm 0.45}$ | $96.48_{\pm 0.64}$ | $97.97_{\pm 0.24}$ | $98.97_{\pm 0.03}$ | $90.88_{\pm 0.73}$ | $96.55_{\pm 0.67}$ | $98.12_{\pm 0.20}$ | $99.04_{\pm 0.03}$ |
| PIDC | $87.87_{\pm 0.64}$ | $94.54_{\pm 0.41}$ | $95.84_{\pm 0.10}$ | $96.77_{\pm 0.08}$ | $88.58_{\pm 0.66}$ | $95.02_{\pm 0.75}$ | $96.78_{\pm 0.19}$ | $97.56_{\pm 0.06}$ |
| Scribe | $47.75_{\pm 6.78}$ | $63.04_{\pm 2.33}$ | $73.37_{\pm 1.11}$ | $70.95_{\pm 1.87}$ | $46.19_{\pm 5.58}$ | $63.42_{\pm 4.19}$ | $72.37_{\pm 1.98}$ | $71.36_{\pm 1.12}$ |
| dynGENIE3 | $83.60_{\pm 3.35}$ | $90.28_{\pm 1.63}$ | $92.28_{\pm 2.10}$ | $98.00_{\pm 0.45}$ | $76.46_{\pm 0.64}$ | $88.32_{\pm 3.03}$ | $90.96_{\pm 1.39}$ | $97.93_{\pm 0.04}$ |
| XGBGRN | $93.72_{\pm 1.08}$ | $98.35_{\pm 0.21}$ | $98.63_{\pm 0.18}$ | $99.40_{\pm 0.01}$ | $86.78_{\pm 2.19}$ | $96.92_{\pm 1.00}$ | $97.94_{\pm 0.28}$ | $99.07_{\pm 0.05}$ |
| NRI | $72.98_{\pm 0.01}$ | $73.85_{\pm 0.02}$ | $74.12_{\pm 0.02}$ | $74.70_{\pm 0.02}$ | $56.76_{\pm 0.02}$ | $59.64_{\pm 0.03}$ | $62.52_{\pm 0.03}$ | $63.52_{\pm 0.02}$ |
| ACD | $65.62_{\pm 0.02}$ | $63.47_{\pm 0.01}$ | $66.69_{\pm 0.02}$ | $61.56_{\pm 0.03}$ | $62.08_{\pm 0.02}$ | $58.14_{\pm 0.03}$ | $61.73_{\pm 0.02}$ | $59.04_{\pm 0.02}$ |
| MPM | $70.23_{\pm 0.02}$ | $74.37_{\pm 0.02}$ | $75.72_{\pm 0.03}$ | $75.60_{\pm 0.03}$ | $62.83_{\pm 0.02}$ | $65.22_{\pm 0.02}$ | $66.52_{\pm 0.02}$ | $66.88_{\pm 0.03}$ |
| iSIDG | $74.33_{\pm 0.03}$ | $76.06_{\pm 0.02}$ | $76.29_{\pm 0.01}$ | $76.54_{\pm 0.03}$ | $63.40_{\pm 0.04}$ | $66.44_{\pm 0.03}$ | $67.52_{\pm 0.03}$ | $68.75_{\pm 0.02}$ |
| RCSI | $73.09_{\pm 0.03}$ | $74.50_{\pm 0.03}$ | $76.83_{\pm 0.02}$ | $76.01_{\pm 0.02}$ | $63.90_{\pm 0.02}$ | $64.72_{\pm 0.02}$ | $65.31_{\pm 0.03}$ | $66.62_{\pm 0.02}$ |

exceeds 20, affecting their AUROC scores. Conversely, PIDC benefits from shorter trajectories as it tends to infer more false positive edges with increasing time steps, often connecting node pairs that co-influence a common node.

The impact of shorter trajectories on the performance of structural inference methods can be mitigated by increasing the number of nodes in the graph. With the exception of Scribe, all methods show smaller reductions in AUROC when the graph contains more nodes, as observed in Fig. 5. Typically, shorter trajectories provide limited information, challenging the methods' ability to discern the true structure. However, larger dynamical systems with more nodes offer richer information, allowing

Table 31: AUROC values (in %) of investigated structural inference methods on BN trajectories with 3 (N3) and 4 (N4) levels of Gaussian noise.

| Method | N3 | | | | N4 | | | |
|---|---|---|---|---|---|---|---|---|
| | n15 | n30 | n50 | n100 | n15 | n30 | n50 | n100 |
| ppcor | $90.87_{\pm 0.66}$ | $96.36_{\pm 0.62}$ | $98.16_{\pm 0.19}$ | $99.15_{\pm 0.04}$ | $90.81_{\pm 0.67}$ | $96.10_{\pm 0.65}$ | $98.09_{\pm 0.19}$ | $99.09_{\pm 0.04}$ |
| TIGRESS | $93.11_{\pm 0.65}$ | $96.45_{\pm 0.62}$ | $97.59_{\pm 0.21}$ | $98.56_{\pm 0.05}$ | $93.00_{\pm 0.38}$ | $96.44_{\pm 0.60}$ | $97.64_{\pm 0.22}$ | $98.57_{\pm 0.05}$ |
| ARACNe | $88.04_{\pm 1.01}$ | $93.42_{\pm 0.85}$ | $96.02_{\pm 0.34}$ | $97.80_{\pm 0.11}$ | $89.51_{\pm 0.73}$ | $93.89_{\pm 0.79}$ | $96.22_{\pm 0.35}$ | $97.85_{\pm 0.12}$ |
| CLR | $91.22_{\pm 0.82}$ | $96.57_{\pm 0.70}$ | $98.20_{\pm 0.20}$ | $99.07_{\pm 0.03}$ | $91.40_{\pm 0.86}$ | $96.63_{\pm 0.71}$ | $98.26_{\pm 0.20}$ | $99.09_{\pm 0.03}$ |
| PIDC | $90.24_{\pm 0.56}$ | $95.17_{\pm 0.75}$ | $96.93_{\pm 0.23}$ | $97.98_{\pm 0.04}$ | $91.53_{\pm 1.11}$ | $95.17_{\pm 0.84}$ | $97.03_{\pm 0.28}$ | $98.12_{\pm 0.04}$ |
| Scribe | $51.12_{\pm 2.82}$ | $61.51_{\pm 3.27}$ | $71.40_{\pm 3.26}$ | $72.10_{\pm 0.97}$ | $48.14_{\pm 2.15}$ | $60.82_{\pm 2.68}$ | $67.96_{\pm 2.52}$ | $70.71_{\pm 1.81}$ |
| dynGENIE3 | $63.28_{\pm 2.16}$ | $80.56_{\pm 2.28}$ | $87.03_{\pm 2.73}$ | $98.04_{\pm 0.03}$ | $52.46_{\pm 0.55}$ | $73.68_{\pm 1.60}$ | $81.89_{\pm 4.03}$ | $97.77_{\pm 0.01}$ |
| XGBGRN | $86.90_{\pm 1.19}$ | $96.38_{\pm 1.00}$ | $97.55_{\pm 0.32}$ | $98.88_{\pm 0.06}$ | $85.29_{\pm 0.62}$ | $95.74_{\pm 1.21}$ | $97.37_{\pm 0.31}$ | $98.75_{\pm 0.07}$ |
| NRI | $50.67_{\pm 0.02}$ | $51.68_{\pm 0.01}$ | $54.40_{\pm 0.02}$ | $58.16_{\pm 0.02}$ | $50.91_{\pm 0.03}$ | $51.11_{\pm 0.02}$ | $51.24_{\pm 0.02}$ | $52.89_{\pm 0.03}$ |
| ACD | $50.09_{\pm 0.03}$ | $54.38_{\pm 0.02}$ | $56.42_{\pm 0.01}$ | $56.12_{\pm 0.02}$ | $51.89_{\pm 0.02}$ | $54.65_{\pm 0.02}$ | $55.73_{\pm 0.03}$ | $55.02_{\pm 0.03}$ |
| MPM | $55.29_{\pm 0.03}$ | $56.81_{\pm 0.03}$ | $57.41_{\pm 0.02}$ | $59.23_{\pm 0.02}$ | $55.85_{\pm 0.03}$ | $57.48_{\pm 0.01}$ | $59.76_{\pm 0.02}$ | $59.90_{\pm 0.02}$ |
| iSIDG | $56.73_{\pm 0.02}$ | $56.79_{\pm 0.02}$ | $57.71_{\pm 0.01}$ | $60.60_{\pm 0.03}$ | $54.59_{\pm 0.04}$ | $57.82_{\pm 0.03}$ | $58.08_{\pm 0.02}$ | $59.70_{\pm 0.02}$ |
| RCSI | $54.20_{\pm 0.02}$ | $54.72_{\pm 0.02}$ | $56.44_{\pm 0.02}$ | $59.43_{\pm 0.03}$ | $52.47_{\pm 0.03}$ | $53.02_{\pm 0.03}$ | $59.50_{\pm 0.02}$ | $58.34_{\pm 0.03}$ |

Table 32: AUROC values (in %) of investigated structural inference methods on BN trajectories with 5 (N5) levels of Gaussian noise.

| Method | N5 | | | |
|---|---|---|---|---|
| | n15 | n30 | n50 | n100 |
| ppcor | $91.11_{\pm 0.69}$ | $95.81_{\pm 0.61}$ | $97.97_{\pm 0.18}$ | $99.04_{\pm 0.05}$ |
| TIGRESS | $92.95_{\pm 0.42}$ | $96.38_{\pm 0.64}$ | $97.66_{\pm 0.18}$ | $98.57_{\pm 0.05}$ |
| ARACNe | $90.22_{\pm 0.96}$ | $94.15_{\pm 0.70}$ | $96.33_{\pm 0.35}$ | $97.90_{\pm 0.11}$ |
| CLR | $91.59_{\pm 0.90}$ | $96.65_{\pm 0.70}$ | $98.31_{\pm 0.20}$ | $99.10_{\pm 0.04}$ |
| PIDC | $91.18_{\pm 1.61}$ | $95.11_{\pm 0.95}$ | $96.90_{\pm 0.32}$ | $98.17_{\pm 0.03}$ |
| Scribe | $52.20_{\pm 6.61}$ | $58.31_{\pm 2.98}$ | $66.41_{\pm 2.87}$ | $69.35_{\pm 1.47}$ |
| dynGENIE3 | $47.84_{\pm 1.10}$ | $67.07_{\pm 2.68}$ | $74.14_{\pm 4.26}$ | $97.46_{\pm 0.03}$ |
| XGBGRN | $85.18_{\pm 0.34}$ | $95.41_{\pm 1.22}$ | $97.27_{\pm 0.28}$ | $98.70_{\pm 0.08}$ |
| NRI | $46.68_{\pm 0.03}$ | $46.70_{\pm 0.02}$ | $49.57_{\pm 0.03}$ | $49.79_{\pm 0.03}$ |
| ACD | $46.21_{\pm 0.03}$ | $46.34_{\pm 0.05}$ | $44.06_{\pm 0.02}$ | $44.41_{\pm 0.02}$ |
| MPM | $55.39_{\pm 0.05}$ | $58.87_{\pm 0.02}$ | $59.07_{\pm 0.03}$ | $60.45_{\pm 0.03}$ |
| iSIDG | $55.59_{\pm 0.03}$ | $58.82_{\pm 0.03}$ | $59.08_{\pm 0.01}$ | $60.70_{\pm 0.02}$ |
| RCSI | $52.00_{\pm 0.04}$ | $55.31_{\pm 0.02}$ | $57.43_{\pm 0.02}$ | $58.10_{\pm 0.03}$ |

the methods to compensate for the limited data and improve performance. This underscores the importance of the interplay between trajectory length and graph size in achieving reliable structural inference results.

Furthermore, ppcor, TIGRESS, and XGBGRN demonstrate remarkable resilience to shorter trajectories. As depicted in Fig. 5, these methods exhibit minimal decreases in AUROC as trajectory lengths decrease. This resilience underscores the robustness of correlation metrics and tree-based approaches when faced with shorter trajectories. Thus, for developing algorithms focused on structural inference with limited data, integrating these techniques could be a promising direction to overcome the challenges of shorter trajectories and enhance the accuracy and reliability of inferred structural connections.

### D.4 Discussion on metrics

The AUROC (Area Under the Receiver Operating Characteristic) metric has several advantages over other metrics such as F1 score, accuracy, and Hamming distance when it comes to evaluating structural inference problems, where the results are binary:

- Handling imbalanced datasets: AUROC is less sensitive to class imbalance compared to accuracy and F1 score. In imbalanced datasets where one class is dominant, such as the adjacency matrix of a sparse graph, accuracy and F1 score can be misleading due to the high accuracy achieved by simply predicting the majority class. AUROC considers the trade-off between true positive rate and false positive rate, making it more suitable for imbalanced datasets.

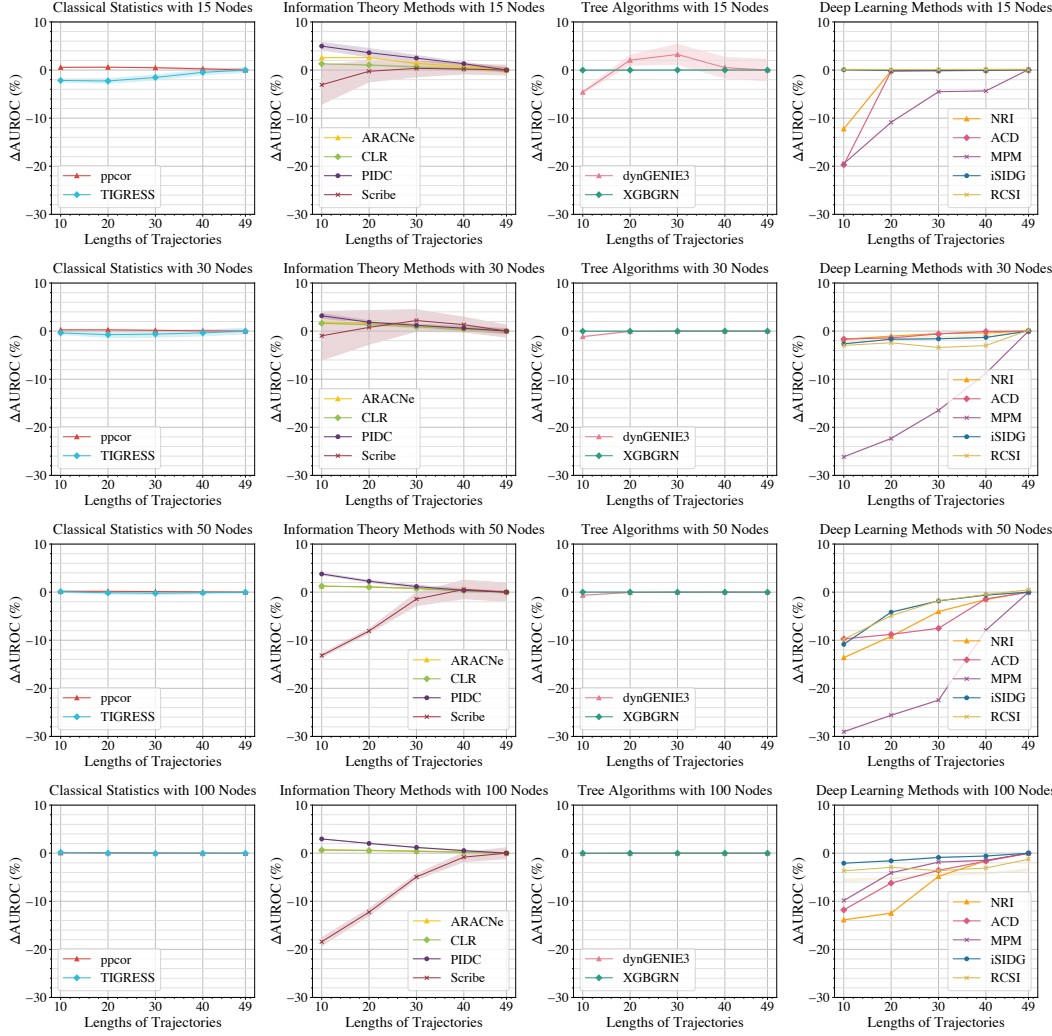

Figure 5: Performance drops (in %) of investigated structural inference methods on BN trajectories of different shorter lengths with respect to the performance on the full-length trajectories.

- Performance across different classification thresholds: AUROC considers the structural inference method's performance at various classification thresholds by plotting the ROC curve. It captures the overall discriminative power of the method across all possible threshold values, whereas F1 score, accuracy, and Hamming distance are based on a specific threshold. This makes AUROC more comprehensive in evaluating the method's performance.

- Robustness to class distribution changes: AUROC remains consistent even when the class distribution changes, for example, the underlying interaction graph may be sparse or dense. In scenarios where the class distribution in the test set differs from the training set, AUROC provides a reliable measure of the method's performance. F1 score, accuracy, and Hamming distance can be influenced by changes in class distribution, leading to biased evaluations.

- Handling probabilistic predictions: AUROC can handle probabilistic predictions and rank them accordingly, which is particularly useful when the structural inference method outputs probabilities instead of hard class labels. F1 score, accuracy, and Hamming distance require explicit thresholding, which may not be suitable for probabilistic outputs.

While F1 score, accuracy, and Hamming distance have their own strengths in specific contexts, AUROC is widely used and preferred when evaluating binary classification tasks due to its robustness, ability to handle imbalanced datasets, and comprehensive evaluation of method performance across different classification thresholds. So in this work, we benchmark all of the methods with AUROC.

### D.5 Benchmarking with EMT Dataset

In addition to the datasets generated by simulations, we also tested the investigated structural inference methods on an EMT dataset collected from the real world. The description of the dataset is in Section 4 and Appendix B.4, and the average results of ten runs are summarized in Table 33.

The EMT dataset is comparable with NetSims dataset with 30 nodes, both sharing a similar node size and unidimensionality. While the model performance on NetSims dataset with 30 nodes ranges from 51.10% to 93.59%, that on the EMT dataset ranges from 51.14% to 57.22%. Such a much lower value and narrower range depicted the difficulty of inferencing a real-world interaction graph. Furthermore, this difficulty cannot simply be explained by the data noise because all methods fail in modeling the cell dynamic, while we have shown that classical statistical methods and information theory-based methods are resistant to Gaussian noise in Table 32. The root causes of the difficulty may include data deficiency in the number of trajectories and time steps, ultra-high complexity of cell dynamics, and unreliable ground truth interaction graphs. In addition, the collected data may only capture a portion of the information from the cell dynamics, and current sequencing technology does not support snapshotting every key change during gene regulations. These uncertainties forbid our benchmarking on other real-world datasets, suggesting that benchmarking on synthetic datasets is the optimal choice for controllable and reliable experiments.

Table 33: AUROC values (in %) of investigated structural inference methods on EMT dataset.

| Method | AUROC |
|---|---|
| ppcor | $55.31_{\pm 0.00}$ |
| TIGRESS | $56.32_{\pm 0.28}$ |
| ARACNe | $57.22_{\pm 0.00}$ |
| CLR | $51.41_{\pm 0.00}$ |
| PIDC | $54.53_{\pm 0.00}$ |
| Scribe | $54.82_{\pm 0.00}$ |
| dynGENIE3 | $44.42_{\pm 0.05}$ |
| XGBGRN | $55.63_{\pm 0.74}$ |
| NRI | $52.09_{\pm 0.06}$ |
| ACD | $51.14_{\pm 0.03}$ |
| MPM | $52.43_{\pm 0.07}$ |
| iSIDG | $52.58_{\pm 0.06}$ |
| RCSI | $53.02_{\pm 0.07}$ |

## E Limitations

This study has certain limitations, which can be summarized as follows: resource limitation, trajectory generation, and the exploration of additional valid methods.

- **Resource limitation:** The computational resources available for this study include NVIDIA Tesla V100 SXM2 cards, AMD Epyc ROME 7H12 CPUs, and AWS Graviton3 processors. As a result, conducting experiments on trajectories generated with larger graphs (e.g., exceeding 100 nodes) would be infeasible or would require a significant amount of time. However, in the interest of fostering further research, we plan to make the trajectories generated by graphs with more than 100 nodes publicly available. We encourage interested researchers to leverage their own computational resources to test alternative structural inference methods on these trajectories.

- **Computational Intensity:** Fully reproducing the benchmarking results presented in this study requires over 263,400 GPU hours, highlighting the computational demands of the evaluation in this paper. We advise researchers to consider their available resources and, where necessary, focus on specific methods or datasets that align with their computational capacity.

- **Assumption:** The fundamental assumption underlying our study is that the nodes in the graph are entirely observed within the specified time frame, and the edges remain stable. However, we acknowledge the potential for nodes to be only partially observed, resulting in incomplete data. Moreover, dynamic graphs may come into play, where nodes and edges evolve over time. While this paper primarily focuses on benchmarking structural inference methods on static graphs, we

recognize the significance of exploring these methods in the context of dynamic graphs. This avenue remains a promising area for future research.

- **Need for Further Validation on Diverse Real-World Datasets:** Although we included an evaluation using the EMT single-cell dataset, there remains a need for broader validation across diverse real-world datasets. This is a notable limitation, and we are committed to addressing it in future work.

- **Trajectory generation:** This study heavily relies on synthetic data generated by synthetic static interaction graphs. While the synthetic graphs were designated based on properties observed in real-world graphs, discrepancies may still exist between them, proven by the differences in model performance between synthetic and experimental GRN networks. Furthermore, the chosen dynamical simulations are based on first-order and second-order ODEs. They may not fully capture the diverse range of dynamical systems encountered in real-world scenarios, such as those based on stochastic differential equations, and those based on quadratic dependency on locations. Future research should aim to incorporate more real-world data and explore a broader array of dynamical simulations to enhance the evaluation of the fidelity and applicability of structural inference methods.

- **Exploration of additional valid methods:** It is important to acknowledge that this study does not encompass all potentially valid methods for structural inference. Numerous methods from various fields may possess the capability to perform, or to be adapted for, the task of structural inference. Besides the methods investigated in this work, we also recognize the possibility of leveraging federated graph learning to perform structural learning, such as [48]. We select the methods for our benchmarking based on four criteria:

  - Representativeness: Our selected methods are either the latest work in its line of work or widely-used methods in its research domain. XGBGRN and RCSI are the latest work in their line of work, while ppcor, ARACNe and CLR are widely used methods in GRN inference. Although GENIE3 is also widely used in GRN inference, we have chosen its successor, the dynGENIE3 method, in our benchmark.

  - Diversity: We only choose one representative if methods have similar functional mechanisms. For example, for all of the methods based on information theory, we choose PIDC and Scribe as they use new MI estimators in their algorithms. Similarly, we choose TIGRESS because it uses feature selection instead of indirect edge elimination in GRN inference.

  - Data constraint: As most methods are domain-specific, we screen out methods with strong data assumptions or low utilization of our data input. For example, methods that only allow single time series input are screened out, such as GRNVBEM [90], SCODE [73] and SINCERITIES [79]. Besides, LinkedSOMs [54] and method in [44] were screened out because the former requires additional single-cell ATAC sequencing data on top of the gene expression level as input, and the latter restricted node interaction as Boolean functions. For similar reasons, we exclude several methods in the field of causal structural discovery, because they either require interventional data [121, 40, 114, 113] or impose strong assumptions [27, 16, 53, 13].

  - Computational constraint: We screen out methods with long computation time such as SINGE [30], PCA-PMI [118], Jump3 [51] and Bayesian network methods.

  We encourage researchers in this field to explore and evaluate other promising methods originating from diverse disciplines. Such exploration will contribute to the advancement of the field and the discovery of innovative approaches to structural inference.

By recognizing and addressing these limitations in future research endeavors, we can enhance the robustness, versatility, and effectiveness of structural inference methods, enabling their application in a wide range of real-world scenarios.

# F   Broader Impact

Structural inference methods on dynamical systems allow numerous researchers in the fields of physics, chemistry, and biology to study the interactions inside the systems. We have shown that investigated methods work well on either one-dimensional node features or multi-dimensional features, where the features are continuous variables. These results prove the wide application of the methods. Similarly to [106], while the emergence of the structural inference technology may be

extremely helpful for many, it has the potential for misuse. Potentially, structural inference methods can be extended to infer online social connections via measuring mutual information or correlations, which could erode privacy.

## G Reproducibility

All results in this benchmark paper can be easily reproduced. The DoSI dataset can be downloaded at: `https://structinfer.github.io/download/`, while the code of all evaluated methods and with our implementation can be found at `https://structinfer.github.io/`. The implementation details in Appendix C will guide the reproduction of the benchmark results.

## H Author statement

We, the authors, confirm that we bear full responsibility for any violation of rights, including but not limited to intellectual property rights, privacy, and confidentiality, that may arise in connection with the content of this paper. We declare that all data used in this study are either created by us or obtained and processed under appropriate licenses that permit their use in this research. We confirm that we have adhered to all relevant data protection and usage guidelines in the preparation of this manuscript.

