# OpenReview forum: "Benchmarking Structural Inference Methods for Interacting Dynamical Systems with Synthetic Data"
_NeurIPS.cc/2024/Datasets_and_Benchmarks_Track — NeurIPS 2024 Track Datasets and Benchmarks Poster_

### Official Review · Reviewer_DD7x · 2024-07-22
**Review of DoSI**

**Rating:** 6
**Confidence:** 4
**Correctness:** Yes.
**Clarity:** Yes.

**Review:**

See Strengths and Limitations.

**Strengths:**

1. The paper highlights structural inference in dynamical systems.
2. The paper considers various types of baselines.
3. The paper evaluates baselines from multiple perspectives.

**Additional Feedback:**

N/A.

**Documentation:**

Should add more details on dataset generation.

**Ethics:**

None.

**Limitations:**

1. Although the authors say that existing methods are tailored to different research domains with unique underlying assumptions, from my point of view, Springs or NetSim simulation also poses a strong and unique inductive bias on generated datasets.
2. Structure inference is highly challenging. To what extent can the simulated structure approximate the actual real-world structure? The data evaluation is necessary.
3. How do you determine the number of nodes and edges? What are the influences of the length of generated trajectories?

**Opportunities For Improvement:**

1. Add datasets of different simulations.
2. Add dataset quality evaluation.
3. Add more details of dataset generation.

**Relation To Prior Work:**

Yes.

**Summary And Contributions:**

This paper develops datasets for structural inference using Springs and Netsims simulation. They evaluate 13 structural inference methods on accuracy, scalability, robustness, and sensitivity to graph properties. The authors find that t classical statistical methods excel in scalability, information theory-based methods in robustness, and deep learning methods in handling complex multi-dimensional features.

---

> ### Author Rebuttal · Authors · 2024-08-16
>
> First of all, we would like to thank Reviewer DD7x for the comments. Here are our answers to the questions and concerns raised by the reviewer:
>
> > **[Opportunities For Improvement 1]** Add datasets of different simulations. AND **[Limitations 1]** Although the authors say that existing methods are tailored to different research domains with unique underlying assumptions, from my point of view, Springs or NetSim simulation also poses a strong and unique inductive bias on generated datasets.
>
> Thank you for your comments. We have added another simulation dataset in Appendix D.5, based on the dynamics of charged particles, which complements the existing simulations of Springs and NetSims datasets that are governed by second-order and first-order ordinary differential equations (ODEs) respectively.
>
> The charged particles simulation introduces a system characterized by quadratic dependencies, a prevalent type of real-world dynamical system not covered by the existing datasets. The dynamics are modeled by the following ODE, which describes the forces acting on each particle due to electrostatic interactions:
>
> $$
> m_ i\cdot x''_ i(t) = \sum_ {j\in\mathcal{N}_ i}C\cdot \text{sign}(q_ i\cdot q_ j)\cdot\frac{1}{\|x_ i(t)-x_ j(t)\|^2},
> $$
> Here, $m_i$ represents the mass of node $i$, assumed to be $1$ for simplicity. $\mathcal{N}_i$ refers to the set of neighboring nodes with connections to node $i$. In this simulation, it represents all nodes in the system. The equation is integrated to compute $x'_i(t)$, and subsequently, $x_i(t)$ is determined for each time step. These calculated values of $x'_i(t)$ and $x_i(t)$ collectively constitute the 4D node features at each time point. Initially, the positions are drawn from a Gaussian distribution $\mathcal{N}(0, 0.5)$, while the initial velocities, represented as 2D vectors, are randomly generated with a norm of $0.5$.
>
> We have reanalyzed this dataset, with results depicted in Table 30 and Figure 1 of the attached PDF. Figure 1 summarizes the average AUROC values across various node counts, comparing these with results from the Springs dataset. Although all methods manage to infer the structure of the underlying interaction graphs, their performance slightly decreases due to the added complexity of distinguishing between two types of interactions in the Charged Particles model. This decrease also correlates with the node count, highlighting the challenges larger graphs pose to structural inference methods.
>
> Furthermore, we validated these methods on the real-world single-cell dataset EMT, detailed in Appendix D.6.
>
> In preparation for the camera-ready version of our paper, we plan to consolidate the results from the Charged Particles simulations into Figure 3 in the main body of the paper and transition the discussions from Appendix D.6 to the main text, leveraging the additional page we are granted.
>
> > **[Opportunities For Improvement 2]** Add dataset quality evaluation.
>
> Thank you for your suggestion. To evaluate the quality of our proposed DoSI dataset, we adapted the metrics from Gong et al., 2023, which were originally designed for image and text data, to suit the characteristics of our dataset. We assessed the dataset using the following adapted metrics: (1) file completeness, (2) trajectory completeness, (3) adjacency matrix completeness, and (4) label accuracy.
>
> **(1) File Completeness:**
> This metric evaluates the presence of expected data files in the dataset. Traditionally calculated as:
>
> $$X_ {FC} = \frac{\sum^{N1}_ {i=1} a_i + \sum^{N2}_ {i=1} b_i}{2 \times \max(N1, N2)},$$
>
> Where $N1$: Number of trajectory files, $N2$: Number of adjacency matrix files.
>
> $a_i$: Traverse the data folder, and for the $i$th data record, check whether the annotation file corresponding to the data record exists in the annotation folder. If it exists, $a_i = 0$, otherwise, $a_i = 1$.
>
> $b_i$: Traverse the annotation folder, and for the $i$th annotation file, check whether the data record file corresponding to the annotation file exists in the data folder.
>
> As each subdataset’s trajectory file and corresponding adjacency matrix file are stored together, we simplified the metric to:
>
> $$X_{FC} = \frac{\sum^{N}_{i=1} a_i }{2 \times N},$$
> Where $N$: Number of trajectory files. $a_i$: Traverse the folder, and for the $i$th trajectory, check whether the adjacency matrix file corresponding to the trajectory exists in the annotation folder. If it exists, $a_i = 0$, otherwise, $a_i = 1$.
>
> **(2) Trajectory Completeness:**
> This metric measures the presence of expected features in each trajectory:
>
> $$X_{TC} = \frac{\sum^T_{t=1}c_t}{T} $$
> where $T$ is the expected length of the trajectory and $c_t = 0$ if the feature at time $t$ exists, $c_t = 1$ if not.
>
> **(3) Adjacency Matrix Completeness:**
> This metric checks for missing values within each adjacency matrix:
>
> $$X_ {AC} = \frac{\sum^{N}_ {i=1}\sum^{N}_ {j=1}d_ {ij}}{N^2}$$
> where $N$ refers to the number of the nodes, $d_{ij}$: check whether the value representing the connectivity from node $i$ to $j$ exists or not in the adjacency matrix (simply whether the values exist or not). If it exists, $d_{ij} = 0$, otherwise, $d_{ij} = 1$.
>
> **(4) Label Accuracy:**
> This metric compares the adjacency matrix used in the dynamics simulation with the matrix obtained from the dataset: $X_{LC} = 0$ if there is no mismatch; otherwise, $X_{LC} = 1$.
>
> We have documented the outcomes of these evaluations in Tables 1 - 3 of the attached PDF. The results indicate that all datasets are free from errors across these four metrics. We will incorporate these findings into the revised manuscript.
>
>
> ---- To be continued on the Part 2 of Rebuttal ---

---

> > ### Author Rebuttal · Authors · 2024-08-16
> >
> > ---- Part 2 of Rebuttal ---
> >
> > > **[Opportunities For Improvement 3]** Add more details of dataset generation.
> >
> > Thank you for your request for more detailed information on dataset generation. We have comprehensively documented the dataset generation process in Appendix B.1 - B.2, which includes specifics on how the graphs for each category were generated and a detailed description of the dynamical simulations. Due to page constraints, we provided only a brief overview in the main body of the paper. However, we plan to expand this section in the camera-ready version, given an additional page allowance, and will also include a direct link to the appendices for detailed information.
> >
> > > **[Limitations 2]** To what extent can the simulated structure approximate the actual real-world structure?
> >
> > Thank you for your question. We have thoroughly examined the properties of each simulated structure and documented the findings in Tables 2 through 12 in the Appendix. In addition to manual inspections, we programmatically verified each simulated structure against the categories of real-world graphs, ensuring that the feature dimensions align with those outlined in Table 1 of Appendix B.1.
> >
> > The results of this assessment are detailed in the attached PDF above in the last rebuttal chat, specifically in Tables 1 to 3 under the column labeled 'Prop.' We use a tick mark to indicate whether the properties of a simulated structure satisfy the criteria set forth in Table 1 of Appendix B.1. As indicated in these tables, the properties of all our graphs successfully meet the specified requirements, suggesting a reasonable approximation of actual real-world structures.
> >
> >
> > > **[Limitations 3.1]** How do you determine the number of nodes and edges?
> >
> > The number of nodes is predefined based on prior benchmarks in structural inference research, such as those used in NRI (Kipf et al., 2019) and ACD (Löwe et al., 2022), which typically involve no more than 10 nodes. To assess the scalability of these methods with larger graphs, we selected node counts of $\\{15, 30, 50, 100, 150, 200, 250\\}$.
> >
> > The number of edges for each graph is guided by the specific graph properties we aim to replicate from real-world networks. This aspect is governed by the graph generation process detailed in Appendix B.1, where we ensure that the properties of the generated graphs, such as edge density and connectivity patterns, fall within the ranges specified in Table 1 in our original submission.
> >
> > > **[Limitations 3.2]** What are the influences of the length of generated trajectories?
> >
> > Thank you for your question regarding the impact of trajectory length on the performance of structural inference methods. We have extensively analyzed this in our benchmarks using different lengths of BN_NS trajectories, with detailed results presented in Appendix D.3 and illustrated in Figure 5.
> >
> > Our findings indicate that most methods experience a decline in AUROC values as trajectory lengths decrease, primarily due to the reduced information available, which limits the methods’ ability to accurately infer underlying structures. Interestingly, methods like ARACNe, CLR, and PIDC respond differently; they exhibit improved performance with shorter trajectories. Specifically, ARACNe and CLR see a drop in performance when trajectory lengths exceed 20 time steps due to the loss of correctly predicted edges, impacting their AUROC scores. In contrast, PIDC benefits from shorter trajectories as it tends to generate more false positives with longer trajectories, often incorrectly connecting nodes that influence a common node.
> >
> > We also observed that the negative impact of shorter trajectories can be mitigated in larger graphs. Except for Scribe, all methods show smaller reductions in AUROC values in larger networks, as more nodes provide richer contextual information that compensates for the brevity of data (Figure 5). This highlights the crucial interplay between trajectory length and graph size in structural inference.
> >
> > Moreover, methods such as ppcor, TIGRESS, and XGBGRN demonstrate exceptional resilience to shorter trajectories, showing minimal performance decreases. Their robustness suggests that incorporating correlation metrics and tree-based approaches could be advantageous for algorithm development in scenarios with limited data.
> >
> > To facilitate easier access to these insights, we will enhance the visibility of the discussion on trajectory length effects by providing a direct link from the main body of the paper to the relevant sections of the appendix.
> >
> > We hope our answers addressed the concerns correctly.

---

> > > ### Comment · Reviewer_DD7x · 2024-09-01
> > >
> > > Thanks for your response. Most of my concerns are addressed. Thus I will raise my score to 6. However, I am still concerned about the consistency between the simulated structure and real-world dynamics since most parameters in Table 1 are missing.

---

> > > > ### Author Response · Authors · 2024-09-01
> > > >
> > > > Dear Reviewer DD7x,
> > > >
> > > > Thank you for your thoughtful review and for considering our responses. We appreciate your decision to raise your score to 6.
> > > >
> > > > Regarding your concern about the missing parameters in Table 1, we would like to clarify that the missing values are derived from the limitations of the literature we referenced, which provides the collection of real-world datasets. These gaps reflect the challenges in obtaining complete information from real-world sources, rather than inconsistencies in our simulation. We understand the importance of this issue and are committed to improving the accuracy and completeness of our datasets in future work.
> > > >
> > > > Thank you again for your valuable feedback.
> > > >
> > > > Best regards,
> > > > Authors

---

> ### Author Response · Authors · 2024-08-31
>
> Dear Reviewer DD7x,
>
> We would like to reiterate our gratitude for taking the time to provide detailed feedback on our paper.
>
> We are rapidly approaching the end of the discussion period. As such, we would greatly appreciate your prompt attention to our rebuttal.
>
> Thanks,
> Authors

---

### Official Review · Reviewer_hQrd · 2024-07-25
**The paper presents a comprehensive evaluation framework for structural inference methods applied to dynamical systems.**

**Rating:** 9
**Confidence:** 4

**Review:**

The paper presents a high-quality and methodical approach to benchmarking structural inference methods. Its robust methodology and well-documented empirical results provide strong evidence of its findings. The paper is clearly written and well-structured, with thorough explanations of the methods, datasets, and experimental setups. Detailed tables and figures enhance the reader's understanding. This novel benchmarking framework addresses a significant gap in the standardization and comparison of structural inference methods across diverse domains. Its comprehensive evaluation framework is highly significant, offering valuable guidance to researchers in selecting and developing structural inference methods for various applications.


Pros and Cons

Pros

- The study evaluates a wide range of methods, offering a broad perspective on their performance across different datasets.
- The Dataset for Structural Inference (DoSI) is a valuable contribution, providing a versatile and comprehensive resource for future research.
- The evaluation criteria cover multiple aspects, including accuracy, scalability, robustness, and sensitivity, offering a holistic view of each method's performance.

Cons

- The benchmarking process is computationally intensive, requiring significant resources and expertise， which makes it hard for broader audience to understand and reproduce.
- While the study includes a real-life biological dataset, further validation on diverse real-world datasets could strengthen the findings. But it is mentioned in the limitation that the aquisition of such dataset is currently hard.

**Strengths:**

The primary strength of the submission is its comprehensive approach to benchmarking structural inference methods, which is highly relevant to the broader research community. The development of the DoSI dataset is particularly noteworthy, providing a valuable resource for future studies. The quality of the research is high, with thorough documentation and rigorous evaluation methods. The ethical and social implications are positive, as the study aims to advance scientific understanding and methodological innovation in structural inference.

**Additional Feedback:**

Overall, the paper makes a valuable contribution to the field by providing a comprehensive and rigorous benchmarking framework for structural inference methods. Future research could focus on dynamic graph models and broader validation to enhance the applicability and impact of the findings. The development of the DoSI dataset is a significant achievement, offering a valuable resource for the research community. But my suggestion to the authors is, the research community really needs a reliable real-world data, and it would be invaluable to work on it.

**Clarity:**

The claims made in the submission are well-supported by the data and methods presented. The study's design and execution are sound, with appropriate evaluation metrics and thorough analysis.

**Correctness:**

The claims made in the submission are well-supported by the data and methods presented. The study's design and execution are sound, with appropriate evaluation metrics and thorough analysis.

**Documentation:**

The documentation of the dataset and methods is thorough, with a website and github, providing sufficient detail for reproducibility. The paper includes information on data collection, organization, and intended uses, supporting ethical and responsible use.

**Ethics:**

There are no significant ethical concerns with the submission.

**Limitations:**

The authors have adequately addressed the limitations of their work in the appendix, acknowledging the computational resources required and the static nature of the graphs used. They suggest future research directions to explore dynamic systems and broader datasets, demonstrating a proactive approach to potential limitations.

**Opportunities For Improvement:**

The limitations of the work include the computational intensity of the benchmarking process and the need for further validation on diverse real-world datasets. Addressing these limitations could enhance the study's generalizability and applicability. But the first step would be design and collect reliable real-world data for structural inference. I do believe this would be the future work.

**Relation To Prior Work:**

The paper provides a clear discussion of how this work differs from and builds upon previous contributions in the section of related work. It situates the benchmarking framework within the context of existing structural inference methods and highlights its novel contributions.

**Summary And Contributions:**

The paper presents a comprehensive evaluation framework for structural inference methods applied to dynamical systems. The authors develop a versatile dataset called the Dataset for Structural Inference (DoSI), which includes various interaction graphs and dynamical functions. They benchmark 13 structural inference methods across synthetic and real-life datasets, focusing on accuracy, scalability, robustness, and sensitivity to graph properties. Key findings indicate that deep learning methods excel with multi-dimensional data, while classical statistical and information theory-based approaches show notable accuracy and robustness. This benchmark aids researchers in selecting suitable methods and stimulates further methodological innovation.

---

> ### Author Rebuttal · Authors · 2024-08-16
>
> We greatly appreciate Reviewer hQrd's motivating review and inspiring questions. Below, we address the concerns raised:
>
> > **[Cons 1]** The benchmarking process is computationally intensive, requiring significant resources and expertise, which makes it hard for broader audience to understand and reproduce.
>
> Thank you for highlighting the computational intensity of the benchmarking process. We acknowledge that the substantial GPU requirements, totaling over 263,400 hours, may pose challenges for reproduction and broader understanding. To facilitate easier access and replication, we have provided a link to our GitHub repository, which includes the complete implementation details, along with a thorough documentation of the selected hyperparameters in Appendix C. We hope these resources will mitigate the difficulties associated with understanding and reproducing our work.
>
> > **[Cons 2]** While the study includes a real-life biological dataset, further validation on diverse real-world datasets could strengthen the findings. But it is mentioned in the limitation that the aquisition of such dataset is currently hard.
>
> Many thanks for the comment. Having worked extensively in this field, we understand that acquiring reliable datasets for structural inference poses significant challenges. Ideal datasets would feature trajectories with minimal error rates, such as low missing rates and comprehensive observations, alongside dependable ground-truth structures. These criteria are essential for accurately assessing the performance of structural inference methods through direct comparisons with the ground truth.
>
> However, obtaining such high-quality real-world data involves considerable expense and time, and despite our diligent efforts, this process is ongoing. We have also explored numerous publicly available datasets across various disciplines, but thus far, only the EMT dataset has met our criteria for providing a satisfying reliable basis for assessment. We are committed to continuing our search for applicable real-world data to further enrich and validate our research in this critical area.
>
> > **[Opportunities For Improvement]** The limitations of the work include the computational intensity of the benchmarking process and the need for further validation on diverse real-world datasets. Addressing these limitations could enhance the study's generalizability and applicability. But the first step would be design and collect reliable real-world data for structural inference. I do believe this would be the future work.
>
> We would like to express our gratitude to the reviewer once again for the thoughtful and motivating comments. We will incorporate the following points into Appendix E - Limitations, presented as new bullet points:
>
> Computational Intensity: As detailed earlier, fully reproducing the benchmarking results necessitates over 263,400 GPU hours. This requirement underscores the computational demands of our approach. We advise interested researchers and readers to be cognizant of this aspect and, where feasible, to possibly focus on specific methods or datasets that align with their resource availability.
>
> Need for Further Validation on Diverse Real-World Datasets: While we have undertaken evaluations using the EMT single-cell dataset, the necessity for further validation across a broader range of real-world datasets persists. We recognize this as a significant limitation and are committed to addressing it in our future work.
>
> We are actively working on the collection of real-world data and aim to make it publicly available in the near future.
>
> > **[Feedback 1]** Future research could focus on dynamic graph models and broader validation to enhance the applicability and impact of the findings.
>
> Thank you for the suggestion. We wholeheartedly agree with your observation. Dynamic graphs are prevalent across numerous domains, and validating structural inference methods on such graphs could greatly enhance their applicability and impact. We will include this important aspect in our future research outlook.
>
> > **[Feedback 2]** But my suggestion to the authors is, the research community really needs a reliable real-world data, and it would be invaluable to work on it.
>
> Thank you for your valuable suggestion. We fully recognize the critical need for reliable real-world data within the research community and are committed to addressing this gap. We will prioritize efforts to gather and utilize such data in our ongoing and future projects.
>
> We hope our answers addressed the concerns and questions proposed by the reviewer correctly.

---

> > ### Comment · Reviewer_hQrd · 2024-08-31
> >
> > Thanks to the author's response, it has addressed most of my concerns and hence, I choose to raise my score

---

> > > ### Author Response · Authors · 2024-08-31
> > >
> > > Dear Reviewer hQrd,
> > >
> > > Thank you for taking the time to consider our responses. We are pleased to hear that we were able to address most of your concerns. We appreciate your decision to raise your score and are grateful for your constructive feedback, which has helped us improve our work.
> > >
> > > Best regards,
> > > Authors

---

### Official Review · Reviewer_3UMV · 2024-07-26

**Rating:** 7
**Confidence:** 4
**Correctness:** Correct.
**Clarity:** Good.

**Review:**

Please see my detailed reviews below.

**Strengths:**

- Important research problem. Inferring network structures based on time series data is important in many applications.

- A versatile benchmark that covers both traditional and deep learning-based approaches and evaluates effectiveness, scalability, and robustness.

- Detailed results analysis provided in both the main text and supplementary materials.

**Additional Feedback:**

Please see my reviews above.

**Documentation:**

Good.

**Ethics:**

None.

**Limitations:**

None.

**Opportunities For Improvement:**

- Benchmark results on real-world datasets are critical for evaluating different approaches. The provided results in Appendix D.6 seem unconvincing, as all methods perform poorly. Is it due to the low quality of this dataset?

- Some explanations of results require further clarification. For example, it is not clear that deep-learning-based approaches perform well on multi-dimensional datasets.

**Relation To Prior Work:**

Yes.

**Summary And Contributions:**

This paper aims to build a benchmark for identifying the topological structures of complex dynamical systems through robust structural inference methods. The authors benchmarked 13 structural inference methods across different disciplines using synthetic data that mimic real-world scenarios, including two types of dynamics and 11 interaction graph models, along with a biological dataset. They evaluated these methods for accuracy, scalability, robustness, and sensitivity to graph properties. The study found that deep learning methods are particularly effective with multi-dimensional data, while classical statistical and information theory-based approaches exhibit notable accuracy and robustness. The correlation between performance and graph properties, such as average shortest path length, was also examined. This comprehensive benchmark aims to guide researchers in selecting appropriate methods for their needs and to encourage further innovation in structural inference methodologies.

---

> ### Author Rebuttal · Authors · 2024-08-16
>
> We greatly appreciate Reviewer 3UMV's insightful review and thoughtful questions. Below, we address the concerns raised:
>
> >  **[Opportunities For Improvement 1]** Benchmark results on real-world datasets are critical for evaluating different approaches. The provided results in Appendix D.6 seem unconvincing, as all methods perform poorly. Is it due to the low quality of this dataset?
>
> Thank you for your observations regarding the performance of methods on the real-world EMT dataset outlined in Appendix D.6. Structural inference, inherently reliant on the quality of observed data, poses challenges when the available real-world datasets include technical noise and confounding factors—common issues that can indeed affect outcomes. Despite these limitations, we selected the single-cell EMT dataset due to its robust pre-processing pipelines and the extensive study it has received within the field of single-cell transcriptomics. This dataset offers a relatively dense reference network, facilitating a more meaningful evaluation compared to other disciplines. It's important to note that gene regulatory network (GRN) inference, closely related to structural inference, uses similar benchmarking methods. Our analysis shows that methods developed for GRN inference perform comparably to those designed for broader structural inference tasks, reinforcing our decision to employ the EMT dataset for demonstrating practical applicability.
>
> > **[Opportunities For Improvement 2]** Some explanations of results require further clarification. For example, it is not clear that deep-learning-based approaches perform well on multi-dimensional datasets.
>
> Thank you for prompting further clarification on this point. Our evaluation shows that deep-learning-based methods like NRI, ACD, MPM ISIDG, and RCSI exhibit notably better performance on datasets with multidimensional features. Specifically, these methods achieved AUROC scores above 90% on the Springs dataset, which are 4-dimensional features, compared to approximately 60% on the NetSims dataset, where each node at every timestep is described by a single-dimensional feature (Figure 2, original submission). Additional evaluations using the Charged Particles dataset, which also includes 4-dimensional features but incorporates different dynamic transition functions compared to Springs, support this observation. Results detailed in Table 30 and Figure 2 (in Appendix D.5) consistently demonstrate that deep-learning-based approaches excel with richer, multidimensional input data. This trend underscores the effectiveness of these methods in contexts where complex feature interactions are present.
>
> We hope our answers addressed the concerns correctly.

---

### Author Rebuttal · Authors · 2024-08-16

## General Rebuttal to All Reviewers

Dear Reviewers,

We sincerely appreciate the thoughtful and comprehensive feedback provided by each of you—Reviewers 3UMV, hQrd, and DD7x. Your insights and suggestions have been invaluable in refining our manuscript and will undoubtedly strengthen the final version.

Reviewer 3UMV highlighted the importance of our work in inferring network structures and appreciated our versatile benchmark, which covers both traditional and deep learning-based approaches. We are grateful for your positive assessment and the good rating, and we have taken your advice into account by adding more real-world datasets in the future work, as suggested in your review.

Reviewer hQrd commended the methodical approach and robust methodology of our study. Your recognition of the Dataset for Structural Inference (DoSI) as a significant contribution motivates us further. The detailed feedback on our evaluation criteria and the comprehensive approach to benchmarking has been instrumental in highlighting the strength of our research.

Reviewer DD7x noted the importance of structural inference in dynamical systems and acknowledged the breadth of our evaluation across multiple perspectives. We appreciate your constructive suggestions on expanding our dataset types and enhancing dataset quality evaluations. We have addressed these points by incorporating additional details on dataset generation and evaluating dataset quality more rigorously.

We are grateful for the points raised by you and have addressed each of your concerns and suggestions specifically in the corresponding sections of our rebuttal.
In response to common themes in your feedback, we have expanded our discussions on computational intensity, dataset diversity, and the implications of graph size and trajectory length on structural inference accuracy. These changes are reflected in the revised manuscript, with specific emphasis placed on improving clarity and expanding our discussions to address the scalability and robustness of the evaluated methods more comprehensively.

We have also ensured that all additional materials and detailed results in the appendix are now more accessible, with clearer references to these in the main body of the paper, as per your suggestions. This will aid in better understanding and replicating our research.

Once again, we thank you for your rigorous evaluations and constructive criticisms. Your insights have not only validated the significance of our work but have also guided us in making it more robust and accessible to a wider audience. We look forward to your final recommendations and are eager to contribute this work to the community.

Warm regards,

Authors

---

### Author Response · Authors · 2024-08-25
**Reminder: Discussion Period Closing Soon**

Dear Reviewers,

Thank you once again for taking the time to review our paper. As the discussion period is nearing its end, we would like to make this comment. We wanted to kindly remind you that we have posted a detailed response addressing the questions and comments raised by each of you. If you have any further concerns or questions, please don’t hesitate to reach out.

Best regards,

The Authors

---

### Decision · Program_Chairs · 2024-09-26

**Decision:**

Accept (Poster)

**Comment:**

The paper introduces a new benchmark for structural inferences on interacting dynamical systems. This is a very nice paper on a significant problem since interacting dynamical systems present a quite challenging setting for which standardized benchmarks are lacking.

 Reviewers appreciate the contributions and are positive towards the submission. The main points of criticism are regarding computational intensity, as well as whether the presented methods perform too poorly. Upon close inspection, considering that this is a benchmark paper, it would be expected that better literature review would be made. As it stands, several more recent publications than then benchmarked methods have been published in NeurIPS, ICML, etc, that should be included, given that they study specifically interacting dynamical systems and show improvements over existing SoTA methods and the limitations of their data settings. Examples can be found in the following papers, the benchmarks they propose (pose estimation over time, charged particles with latent electrical fields) and the papers they cite in their comparisons (including SDS, RedSDS):

https://github.com/mkofinas/aether
https://github.com/mkofinas/locs
https://github.com/yongtuoliu/Graph-Switching-Dynamical-Systems
https://github.com/yongtuoliu/Amortized-Equation-Discovery-in-Hybrid-Dynamical-Systems
https://github.com/abdulfatir/REDSDS
https://github.com/slinderman/recurrent-slds

Considering the important of having standardized benchmarks in interactive dynamical systems, I recommend the paper to be accepted, conditional on addressing the above points. Specifically, regarding computational complexity, the authors should consider creating a mini version of the dataset, by selecting carefully representative samples that would correlate well with the whole datasets. Also, the authors should include the more competitive baselines above, and their data settings, as well as to explain the differences with the proposed benchmark, to ensure the submission is in line with the latest state-of-the-art.